# Tectonic evolution of the Indio Hills segment of the San Andreas fault in southern California, southwestern USA

Jean-Baptiste P. Koehl[1,2,3,4], Steffen G. Bergh[2,3], Arthur G. Sylvester[5]

1) Centre for Earth Evolution and Dynamics, (CEED), University of Oslo, N-0315 Oslo, Norway.

2) Department of Geosciences, UiT The Arctic University of Norway in Tromsø, N-9037 Tromsø, Norway.

3) Research Center for Arctic Petroleum Exploration (ARCEx), UiT The Arctic University of Norway.

4) CAGE – Centre for Arctic Gas Hydrate, Environment and Climate, UiT The Arctic University of Norway.

5) Department of Earth Science, University of California, Santa Barbara, USA.

**Correspondence:** jeanbaptiste.koehl@gmail.com

## Abstract

Transpressional uplift domains of invertedPliocene–Pleistocene basin fill along the San Andreas fault zone in Coachella Valley, southern California, are characterized by fault linkage and segmentation and deformation partitioning. The Indio Hills wedge-shaped uplift block is located in between two boundary fault strands, the Indio Hills fault to the northeast and the main San Andreas fault to the southwest, which merge to the southeast. Uplift commenced about or later than 0.76 million years ago and involved progressive fold and faulting stages caused by a change from distributed strain to partly partitioned right-slip and reverse/thrust displacement on the bounding faults when approaching the fault junction. Major fold structures in the study area include oblique, right-stepping, partly overturned *en echelon* macro-folds that tighten and bend into parallelism with the Indio Hills fault to the east and become more open towards the main San Andreas fault to the west, indicating an early and close relationship of the macro-folds with the Indio Hills fault and a late initiation of the main San Andreas fault. Sets of strike-slip to reverse step-over and right- and left-lateral cross faults and conjugate kink bands affect the entire uplifted area, and locally offset the *en echelon* macro-folds. Comparison with the Mecca Hills and Durmid Hills uplifts farther southeast in Coachella Valley reveals notable similarities, but also differences in fault architectures, spatial and temporal evolution, and deformation mechanisms.

## Introduction

This paper describes and evaluates structural patterns of the Indio Hills uplift in the northwestern part of Coachella Valley along the San Andreas Fault Zone (SAFZ) in California, southwestern USA (Fig. 1), where the fold–fault architecture, evolution, and

partitioning of deformation compared to Mecca Hills and Durmid Hills are not well
understood (e.g., Keller et al., 1982, Dibblee and Minch, 2008). The main goal of this study is
to analyze internal macro- and meso-scale folds and related faults and to outline the kinematic
evolution in relation to major SAFZ-related fault strands in the area (Fig. 1: Keller et al.,
1982; Guest et al., 2007). These include the Indio Hills fault in the northeast (Allen, 1957;
Tyley, 1974), and the main San Andreas fault along the southwest flank of the Indio Hills (we
refrain from using the name "Indio strand" ascribed to this fault by Gold et al., 2015 to avoid
confusion with the Indio Hills fault) and of the Mecca Hills, and along the northeast flank of
the Durmid Hills (Janecke et al., 2018; Fig. 1). The progressive tectonic evolution model for
the Indio Hills uplift is then compared and correlated with other major uplifts and SAFZ-
related fault strands along strike in the Mecca Hills and Durmid Hills (Sylvester and Smith,
1987; McNabb et al., 2017; Janecke et al., 2018; Bergh et al., 2019). We also discuss briefly
the potential northwestward continuation of the Indio Hills fault into the East California Shear
Zone and its role as possible transfer fault (Dokka and Travis, 1990a, 1990b; Thatcher et al.,
2016). The variable fault and fold architectures and associated ongoing seismic activity in
these uplift areas underline the need for persistent along-strike studies of the SAFZ to
characterize the fundamental geometry, resolve the kinematic development, and correlate
regionally major fault strands (cf. Janecke et al., 2018). Such studies are essential to explain
the observed lateral variations in fold and fault architectures and to resolve mechanisms of
transpression, fault linkage, and areal segmentation in continental transform settings.

**Geological setting**
The Coachella Valley segment of the SAFZ in southern California is expressed
asmultiple, right-lateral fault strands, which uplifted blocks in the Indio Hills, Mecca Hills,
and Durmid Hills (Fig. 1; Sylvester, 1988). These domains comprise thick successions of
Pliocene–Pleistocene sedimentary strata uplifted and deformed in Pleistocene–Holocene time
due to oblique convergence of the Pacific and North American plates and movement along the
SAFZ and related faults (e.g., Atwater and Stock, 1998; Spotila et al., 2007; Dorsey et al.,
2011). Recent structural studies in the Mecca Hills (McNabb et al., 2017; Bergh et al., 2019),
and Durmid Hills at the southern termination of the SAFZ (Janecke et al., 2018), show that
individual fault strands are linked, and that the deformation splits into abruptly changing fold
and fault geometries (Fuis et al., 2012, 2017). These recent works call for further
characterization of the understudied Indio Hills segment in order to compare its structural
development with other uplifted features along a major transform plate boundary fault zone.
Below we summarize local and regional fault nomenclature, distribution, and fault movement
history (Table 1) throughout the greater Coachella Valley region (Fig. 1), the stratigraphy of
the Indio Hills area, and previous structural work in the main Indio Hills, Mecca Hills, and
Durmid Hills uplift areas.

*Regional faults*

The southeastern Indio Hills are a WNW–ESE–trending tectonic uplift situated in a

small restraining bend northeast of the main San Andreas fault (Figs. 1 and 2 and Supplement
S1). The studied uplift is located along strike about 25–50 kilometers northwest of the Mecca
Hills and Durmid Hills, and to the southeast of the major left bend in the SAFZ trace near San
Gorgonio Pass (Matti et al., 1985, 1992; Matti and Morton, 1993; Dair and Cooke, 2009).

The main faults in the southeastern Indio Hills include the Indio Hills fault in the

northeast (Allen, 1957; Tyley, 1974), and the main San Andreas fault in the southwest.
Regionally, the Indio Hills fault possibly merges with the Landers Mojave Line and the
eastern California shear zone in the north (Dokka and Travis, 1990a, 1990b; Nur et al., 1993a,
1993b; Thatcher et al., 2016). The Landers Mojave Line is believed to be the locus of several
recent earthquakes aligned along a NNW–SSE-trending axis, including the 1992 Joshua Tree
earthquake (Fig. 1b; Nur et al., 1993a, 1993b). These earthquakes were tentatively ascribed to
movement along a through-going NNW–SSE-striking fault, possibly the west-dipping,
Quaternary West Deception Canyon fault (Sieh et al., 1993; Rymer, 2000). This fault is
thought to crosscut the E–W- to ENE–WSW-striking, left-lateral, Holocene Pinto Mountain
fault, which merges with the main strand of the San Andreas fault in the west at the
intersection of the right-lateral Mission Creek and Mill Creek strands (Allen, 1957; Bryant,
2000; Kendrick et al., 2015; Blisniuk et al., 2021). The former is thought to correspond to the
continuation of the main San Andreas fault to the northwest (Gold et al., 2015) and may have
accommodated ca. 89 km of right slip in the past 4 million years, whereas the latter
accommodated about 8 km right slip at 0.5–0.1 Ma and is offset ca. 1 km by the Pinto
Mountain fault (Kendrick et al., 2015).

The main San Andreas fault continues to the southeast where it bounds the Mecca

Hills to the southwest, whereas the Painted Canyon fault, a previous (late Miocene?–)
Pliocene southwest-dipping normal fault reactivated as a right-lateral-reverse oblique-slip
fault in the Pleistocene–present-day bounds the Mecca basin to the northeast (Sylvester and
Smith, 1987; McNabb et al., 2017; Bergh et al., 2019). Farther southeast, the main San
Andreas fault proceeds along the northeast flank of the Durmid Hills opposite the Pleistocene

(ca. 1 Ma), right-lateral East Shoreline fault (Babcock, 1969, 1974; Bürgmann, 1991; Janecke et al., 2018). There, the main San Andreas fault merges with the Brawley seismic zone (Lin et al., 2007; Hauksson et al., 2012; Lin, 2013) and, together with the right-lateral San Jacinto fault zone, they merge into the right-lateral Imperial fault (Rockwell et al., 2011). In the north, the main San Andreas fault splays into the Banning strand and the Mission Creek fault in the northwestern part of the Indio Hills (Keller et al., 1982; Gold et al., 2015). The Banning strand is much younger than the Mission Creek fault and may have accommodated approximately 3 km of right slip in the past 0.1 million years (Kendrick et al., 2015).

Northwest and west of the Coachella Valley, Miocene–Pleistocene sedimentary strata are structurally bounded by the San Bernardino and San Jacinto fault strands of the SAFZ (Bilham and Williams, 1985; Matti et al., 1985; Morton and Matti, 1993; Spotila et al., 2007). To the southwest, Miocene–Pleistocene strata are bounded by the West Salton Detachment fault (Dorsey et al., 2011). The San Jacinto fault is typically believed to have slipped ca. 25 km right-laterally in the past 1.5 million years (Matti and Morton, 1993; Kendrick et al., 2015), whereas the West Salton Detachment fault is a low-angle normal fault that accumulated ca. 8–10 km of normal-oblique movement starting in the mid Miocene and is related to the opening of the Gulf of California (Prante et al., 2014 and references therein).

### Stratigraphy of the Indio Hills and adjacent areas

The Indio Hills uplift is an inverted Pliocene–Pleistocene sedimentary basin lying upon Mesozoic granitic basement rocks, which we regard as an analog to the inverted Mecca basin farther southeast (Keller et al., 1982; Damte, 1997; McNabb et al., 2017; Bergh et al., 2019). In the Mecca basin, alluvial, fluvial and lacustrine deposits of the Mecca and Palm Spring formations are truncated unconformably by the mid to upper Pleistocene Ocotillo Formation (Dibblee, 1954; Sylvester and Smith, 1976, 1987; Boley et al., 1994; Rymer, 1994; Sheridan et al., 1994; Sheridan and Weldon, 1994; Winker and Kidwell, 1996; Kirby et al., 2007; McNabb et al., 2017; Table 1). Similar uplifted strata at Durmid Hills (Fig. 1) belong to the Pliocene–Pleistocene Borrego Formation, and are overlain by mid to upper Pleistocene deposits of the Brawley and Ocotillo formations (Dibblee, 1997; Herzig et al., 1988; Lutz et al., 2006; Kirby et al., 2007; Dibblee and Minch, 2008).

Leuco-granitic rocks crop out near gently SW-dipping conglomerates along the northeastern flank of the Indio Hills, near the trace of the Indio Hills fault (Fig. 2). Despite proximity of the conglomerates with segmented granite outcrops, the contact itself is not exposed. The conglomerates are the lowermost stratigraphic unit exposed in the Indio Hills

and are characterized by a succession of meter-thick beds of very coarse, poorly sorted blocks
of gneissic and granitic rocks more than a meter in size. Previous mapping in the area
(Dibblee, 1954; Lancaster et al., 2012) considered the conglomerates as stratigraphic
equivalents to the mid to upper Pliocene Mecca Formation in the Mecca Hills (Sylvester and
Smith, 1987; McNabb et al., 2017; Bergh et al., 2019) and that at least part of the clasts are
from the leuco-granitic rocks, which must correspond to basement rocks of the inverted Indio
Hills basin. Up-section toward the southwest the conglomerate gradually is succeeded by
coarse-grained sandstone, which defines the transition from the Mecca Formation to the lower
Palm Spring Formation.
The Palm Spring Formation in the Indio Hills consists of moderately- to well-
consolidated alluvial fan deposits (Dibblee and Minch, 2008), with some interbedded gypsum
layers and red-colored calcareous mudstone, as in the Mecca Hills (Sylvester and Smith,
1987). The main rock types include beds of light-colored, medium- to coarse-grained
sandstone, gray–brown silty sandstone, and dark biotite-rich mudstone. The southwestwards
increase in silt–clay toward the main San Andreas fault (also recorded in the Mecca Hills;
Bergh et al., 2019) may indicate a gradual transition from the lower to the upper member of
the Palm Spring Formation.
By contrast, the transition between the lower and upper members of the Palm Spring
Formation in the Mecca Hills is marked by two angular unconformities that signal further
steps in uplift and inversion of the Mecca basin (Table 1; McNabb et al., 2017; Bergh et al.,

2019).

Ages of 3.0-2.3 Ma (latest Pliocene–early Pleistocene) and 2.6–0.76 Ma (earliest
Pleistocene to earliest late Pleistocene), were obtained respectively for the lower and upper
member of the Palm Spring Formation in the Mecca Hills based on reversed magnetic polarity
data (Chang et al., 1987; Boley et al., 1994; McNabb, 2013; McNabb et al., 2017; Table 1).
We infer a similar age range for the Palm Spring Formation in the southern Indio Hills.
In contrast to other uplift areas in Coachella Valley, the Ocotillo Formation has not
been mapped in the Indio Hills in the present study. However, based on the occurrence of the
Bishop Ash at the northwestern edge of the study area and on the occurrence of the volcanic
deposit within the uppermost Palm Spring Formation or at the base of the overlying Ocotillo
Formation in the Mecca Hills, it is likely that the Ocotillo Formation is present just northwest
of the area mapped (Fig. 2). In addition, it is deposited on the flank northeast of the Indio
Hills fault, and southwest of the main San Andreas fault (Figs. 1 and 2), indicating that this
unit was either not deposited or eroded in the area that recorded the most uplift in Indio Hills.
Additional dating constraints on transpressional uplift in the Coachella Valley include
tephrochonology of the 0.765 million year old Bishop Ash layer (Sarna-Wojcicki et al., 2000;
Zeeden et al., 2014; Table 1). This volcanic deposit is found within the upper member of the
Palm Spring Formation (which is unconformably overlain by the Ocotillo Formation) in the
hanging wall of the Painted Canyon fault away from the fault, and within the base of the
Ocotillo Formation in the hanging wall of the Painted Canyon fault near the fault (Ocotillo
and uppermost Palm Spring formations interfingering near the fault) and in the footwall of the
fault (McNabb et al., 2017; Bergh et al. 2019). The unconformable contact between the Palm
Spring and Ocotillo formations away from the Painted Canyon fault towards the southwest
and their interfingering relationship near the fault suggest that uplift had already initiated prior
to deposition of the Ocotillo Formation (i.e., before 0.76 Ma, in the mid Pleistocene), possibly
during the formation of the lower unconformity between the lower and upper members of the
Palm Spring Formation (McNabb et al., 2017; Table 1). Complementarily, the involvement of
the Bishop Ash in deformation suggest that deformation continued past 0.76 Ma (in the late
Pleistocene).

***Major tectonic uplifts in the Coachella Valley***
*Indio Hills*
The southeastern end of the Indio Hills is an uplifted domain of deformed strata of the
Mecca and Palm Spring formations situated in between the main San Andreas and Indio Hills
fault (Fig. 2). The main San Andreasfault corresponds to a major oblique strike-slip fault
segment at the eastern end of San Gorgonio Pass (Matti et al., 1985; Morton et al, 1987). It is
easily traced to Indio Hills (Figs. 1 and 2) since its main trace provides preferential pathways
for ground water flow and growth of wild palm trees along strike.
The Indio Hills fault was mapped north of the study area (Dibblee and Minch, 2008)
extending into the Landers–Mojave Line (Nur et al., 1993a, 1993b), a NNW–SSE-striking
right-lateral fault system extending hundreds of kilometers northward from the southeastern
Indio Hills into the East California Shear Zone and related fault segments such as the Calico
and Camp Rock faults (Fig. 1; Dokka et al., 1990a; Nur et al. 1993b). The Indio Hills fault
may correspond to a major fault splay of the SAFZ (Dokka and Travis, 1990a, 1990b;
Thatcher et al., 2016). Southeast of the Indio Hills, however, the geometry of the Indio Hills
fault remains elusive, and the fault either dies out or merges with structures like the main San
Andreas fault, the Skeleton Canyon fault, and/or the Painted Canyon fault in the Mecca Hills
(Fig.1).
The transpressional character of the Indio Hills uplift was suggested by Sylvester and
Smith (1987). However, detailed structural analyses documenting this hypothesis for the uplift
as a whole have not been conducted. Gold et al. (2015) explore tectonogeomorphic evidence
for dextral-oblique uplift and Keller et al. (1982) and Blisniuk et al. (2021) focus on landscape
evolution near the intersection of the Banning strand and Mission Creek fault (northwest of
the study area), which merge into the main San Andreas fault (Fig. 1). In addition to
investigating soil profiles, offset drainage systems, and recent (a few thousand years old)
displacement along the SAFZ, Keller et al. (1982) called attention to a strong dominance of
gently plunging and upright macro-folds in bedrock strata along the Mission Creek fault and
at the southeastern end of the Banning strand where these faults merge. Their study showed
that bends and steps along the main fault traces were consistently located near brittle fault
segments and zones of uplift. The study also showed that drainage systems were offset
recently (at ca. 0.03–0.02 Ma) and indicate relatively high slip rates along the Mission Creek
fault in the order of 23–35 cm.y$^{-1}$, i.e., comparable to the more recent c. 23 cm.y$^{-1}$ estimate by
Blisniuk et al. (2021).
*Mecca Hills*
Farther south, the Mecca Hills uplift was previously defined as a classic flower-
structure (Sylvester and Smith, 1976, 1987; Sylvester, 1988), in which all folds and faults
formed synchronously and merged at depth. Recent analyses (Bergh et al., 2014, 2019)
indicate that a modified flower-like structure, consisting of a steep SAFZ fault core zone to
the southwest, a surrounding approximately one–two kilometers wide damage zone expressed
by *en echelon* folds and faults oblique to the SAFZ (including left-slip cross faults), steeply
plunging folds, and SAFZ-parallel fold and thrust belt features (including right- and left-slip
and oblique-reverse faults) formed in kinematic succession. In addition to the steep (shallow)
SAFZ (Fuis et al., 2012, 2017), two other, major NW–SE-striking faults occur in the Mecca
Hills (Fig. 1). One is the Skeleton Canyon fault, which initiated as a steep SAFZ-parallel
strike-slip fault and was reactivated as a reverse and thrust fault dipping gently northeastwards
in the late kinematic stages (Sylvester and Smith, 1976, 1979, 1987; Bergh et al., 2019). The
other is the Painted Canyon fault, which is a former Miocene–Pliocene basin-bounding
normal fault (McNabb et al., 2017) and is now reactivated as a NE-directed thrust fault with
dip to the southwest (Bergh et al., 2019; Table 1). The polyphase evolution and reactivation of
internal oblique, step-over faults, and SAFZ-parallel faults, were explained by a series of
successive–overlapping events involving a change from distributed, locally partitioned, into
fully partitioned strain in a changing, oblique-plate convergence regime (Bergh et al., 2019).

*Durmid Hills*

The Durmid Hills are an elongate ridge that parallels the main strand of the SAFZ at the south edge of the Salton Sea in Imperial Valley (Fig. 1). Farther south, this deformation zone and the SAFZ project towards the Brawley seismic zone, an oblique, transtensional rift area with particularly high seismicity (Lin et al., 2007; Hauksson et al., 2012; Lin, 2013). The main San Andreas fault (mSAF) is located on the northeast side of the Durmid Hills and has been thoroughly studied (Dibblee, 1954, 1997; Babcock, 1969, 1974; Bilham and Williams, 1985; Bürgmann, 1991; Sylvester et al., 1993; Lindsey and Fialko, 2013; Janecke et al., 2018). The rocks southwest of the mSAF consist of highly folded Pliocene–Pleistocene deposits (Babcock, 1974; Bürgmann, 1991; Markowski, 2016; Janecke et al., 2018) bounded to the southwest by the subsidiary East Shoreline Fault strand of the SAFZ. Northeast of the mSAF, the formations are much less deformed (Janecke et al., 2018). The overall structure (Fig. 1) resembles a right-lateral strike-slip duplex (Sylvester, 1988), but the geometry is not fully consistent with a duplex model due to abundant left-lateral cross faults and internal block rotations. Instead, the Durmid Hills structure was interpreted as a ladder structure (Janecke et al., 2018), as defined by Davis (1999) and Schulz and Balasko (2003), where overlapping, E–W- to NW–SE-striking step-over faults rotated along multiple connecting cross faults. The one–three kilometers wide Durmid ladder structure consists of multiple internal, clockwise-rotating blocks bounded by major *en echelon* folds and right- and left-lateral cross faults in between the right-slip mSAF and Eastern Shoreline fault strand, indicating a complex termination of the SAFZ around the Brawley Seismic Zone to the southeast (Fig.1).

**Methods and data**

In our investigation of the Indio Hills, we used high-resolution Google Earth DEM images and aerial photographs (© Google Earth 2011) as a basis for detailed field and structural analyses (Fig. 2). We mapped and analyzed individual macro- and meso-scale folds and associated faults in Miocene–Pliocene strata both in the field and via imagery analysis. Key horizons of light-colored quartz sandstone and carbonate rocks in the Palm Spring Formation provide structural markers, notably for restoring bed offsets and fault–fold geometries and kinematics. We address crosscutting relations of the main San Andreas and Indio Hills faults and nearby fold structures. Structural orientation data are obtained from meso-scale folds and faults and are integrated between the areal segments to link a prevalent pattern of deformation into a wider structural architecture (Fig. 2).

**Results**

*Structural overview of the Indio Hills*

The study area comprises three major, SAFZ-oblique, asymmetric, E–W-trending, moderately west-plunging fold systems having multiple smaller-scale parasitic folds (Fig. 2). The main folds affect most of the Palm Spring Formation in a zone approximately two kilometers wide between the-main San Andreas and Indio Hills faults (Fig. 2). The northeastern flank of the Indio Hills is structurally different by consisting of a sub-horizontal, NW–SE-trending, open, upright anticline, which trends parallel to the Indio Hills fault (Fig. 2). Similarly, close to the main San Andreas fault, tilted strata of the Palm Spring Formation are folded into a tight, steeply plunging shear fold (folds involving shearing along a plane that is parallel to subparallel to the fold's axial plane; Groshong, 1975; Meere et al., 2013; Fig. 2). At smaller scale, several subsidiary reverse faults and mostly right-slip, step-over faults having orientations both parallel with (E–W to NW–SE) and perpendicular (NNE–SSW) to the bounding faults exist within the macro-folded domain. Most of these faults truncate individual SAFZ-oblique folds.

*SAFZ-oblique macro-folds*

SAFZ-oblique macro-folds are consistently asymmetric and mostly south-verging, and their axial surfaces are arcuate and right-stepping (Fig. 2). Fold geometries change from open and nearly upright near the main San Andreas fault, to kink/chevron styles in the middle part, to very tight (isoclinal) and overturned fold styles adjacent to the Indio Hills fault (Fig. 3a–c and Supplement S2a–c). These changes in geometry correspond to a change in obliquity of the fold axial surface trace from approximately 60–70° to less than 20° with the Indio Hills fault (Fig. 2). All three macro-folds have axial trends that bend and partly merge into parallelism with the Indio Hills fault. In contrast, moderate to steeply WSW-dipping strata of the Palm Spring Formation are obliquely truncated by the main San Andreas fault. Tighter fold hinges are mapped in the central macro-fold and on the back-limb (stretched long limb in an overturned fold) of the Z-shaped, southeastern macro-fold (Fig. 2). These folds were not observed northeast of the Indio Hills fault, nor southwest of the main San Andreas fault.

*Northwestern and central macro-folds*

The northwestern and central macro-folds define two major, compound and arcuate fold systems that affect the entire Palm Spring Formation between the main San Andreas and Indio Hills faults (Fig. 3a–b). They consist of eight subsidiary Z- and S-shaped, south-verging

anticline-syncline pairs, and show fold axes plunging variably but mostly about 30° to the west (Fig. 2). At large scale, both folds tighten northeastward and display clockwise bend of axial traces from ENE–WSW near the main San Andreas fault, to E–W and NW–SE as they approach the Indio Hills fault (Fig. 2 and 3c). Fold hinges in the west are typically symmetric, concentric, and open (Supplement S3a–b), and become gradually tighter and dominantly Z-shaped kink folds eastward (Supplement S3c). The folds transform into tight, isoclinal, and inverted geometries (Supplement S3d–e) when approaching the central macro-fold back-limb (Fig. 3b), and they potentially merge with the SAFZ-parallel anticline less than 200 meters from the Indio Hills fault (Fig. 2). From southwest to northeast, the central macro-fold hinge zone displays a corresponding change in geometry , i.e., from symmetric, to kink/chevron, and to isoclinal overturned styles (Supplement S4a–b), until the folds of the central macro-fold flank the back-limb of the southeastern macro-fold (Supplement S4c–d). Bedding surfaces on the fore-limb (the shortened, inverted limb indicating the direction of tectonic transport in an overturned fold) of the central macro-fold dip steeply or are inverted, whereas strata on the back-limb mostly dip gently to the north or northwest, i.e., at a high angle to the bounding faults, and gradually change to northward dip when approaching the Indio Hills fault (Fig. 3c).

Another feature of the central macro-fold is that it is offset by a system of both layer-parallel and bed-truncating faults (Fig. 3b). Strata east of the fault system are affected by a large shear fold having thickened hinges and thinned limbs. The next fold to the north-northeast changes from open to tight, overturned, and locally isoclinal (Supplement S4a–c), and merges with the inverted, NE-dipping back-limb of the southeastern macro-fold (Fig. 3c). Notably, the consistent eastward tightening of fold hinges occurs within the lower stratigraphic parts of the Palm Spring Formation, whereas conglomerates of the underlying Mecca Formation are only weakly folded (see section about the southeastern macro-fold). Furthermore, beds in tighter folds (especially in relatively weak clayish–silty dark mudstone layers) are commonly accompanied by disharmonic folds and internal structural disconformities. By contrast, more rigid, and thicker sandstone beds are more commonly fractured.

*Southeastern macro-fold*

The southeastern macro-fold is expressed as a kilometer-wide, Z-shaped, open to tight, south-verging syncline-anticline pair showing moderately west-plunging axes and steeply north-dipping axial surfaces (Fig. 3c). Most of the Palm Spring Formation strata on the back-limb trend parallel to the Indio Hills fault and dip about 50–70° to the north, whereas strata in

the hinge and fore-limb dip about 40–70° to the west/southwest (Fig. 3c). Combined with a
relatively narrow hinge zone, these attitudes define the southeastern macro-fold as a chevron
type. The axial trend of the syncline-anticline pair is at a low angle (< 20°) to the Indio Hills
fault but bends into a NE–SW trend westward with a much higher (oblique) angle to the main
San Andreas fault, which cuts off the fore-limb strata (Fig. 2). The southeastern macro-fold is
very tight in the north and east and has several smaller-scale, tight to isoclinal, strongly
attenuated folds on the main back-limb that merge from the central macro-fold, thus
indicating increasing strain intensity northeastward (see discussion). In contrast to the tightly
folded beds of the Palm Spring Formation, bedding surfaces in conglomerates of the
underlying Mecca Formation are only weakly folded northeastward and becomes part of the
open SAFZ-parallel anticline close to the Indio Hills fault.
A macro-folded siltstone layer of the lower Palm Spring Formation more than 200
meters southwest of the Indio Hills fault (Fig. 4a) contains centimeter-scale, upright (sub-
horizontal) and disharmonic folds having E–W trend and western plunge (Fig. 4b). These
intra-layer folded strata are cut by low-angle reverse faults yielding a NE-directed sense-of-
shear. The upright geometry and the sub-horizontal fold axes (about 5° plunge) of these intra-
bed minor folds differ from the SAFZ-oblique folds but resemble those of the macro-scale,
SAFZ-parallel NW–SE-trending anticline near the Indio Hills fault. These disharmonic folds
are interpreted as intra-detachment folds (see discussion).

*SAFZ-parallel macro-folds*
About 100–200 meters southwest of the trace of the Indio Hills fault, the
conglomerates of the Mecca Formation are folded into a major open anticline, whose axis is
parallel to slightly oblique (< 20°) to the Indio Hills fault. This macro-fold is traceable
northwestward to where the Indio Hills fault bends northward (Fig. 1). The southwestern limb
of the fold marks the transition from the Mecca Formation conglomerate with the overlying
Palm Spring Formation on the back-limb of the southeastern and central macro-folds (Fig. 2
and Supplement S4c). The conglomerate beds are thicker, nearly unconsolidated, and much
less internally deformed than the strata of the Palm Spring Formation. The major anticline
displays an open, symmetric, partly box-shaped, NW–SE-trending, upright geometry with 2–
3° plunge of the fold axis to the northwest. Outcrops on the SW-dipping limb of the anticline
(Fig. 3c) are cut by a SW-dipping reverse fault system that is sub-parallel to the Indio Hills
fault (Supplement S5a). These reverse faults may be linked with the reverse fault in folded
strata of the Palm Spring Formation on the southeastern macro-fold back-limb described

above (Fig. 4). The upright geometry and sub-horizontal NW–SE-trending axes of related small-scale folds in a mudstone layer (Fig. 4) resembles that of the SAFZ-parallel anticline.

A couple of major synclines showing axial traces parallel to the main San Andreas fault are also well displayed on DEM images (Fig. 5 and Supplement S6). These folds affect WSW-dipping strata of the Palm Spring Formation on the broadened western part of the northwest and central macro-folds. Fold geometries are tight and asymmetric, with wavelengths less than 200 meters, and presumably steep NW-plunging axes. The local appearance and sheared geometry of these folds contrast both with the broad SAFZ-oblique folds near the main San Andreas fault, and with that of the upright, SAFZ-parallel anticline near the Indio Hills fault.

### *Major and minor faults*

. Fold-related brittle faults exist both in granitic basement and in sedimentary rocks of the Mecca and Palm Spring formations in the study area. Such faults display narrow damage zones less than one meter wide and are geometrically either related to SAFZ-oblique or SAFZ-parallel macro- and meso-scale folds, or are orthogonal to the SAFZ and related faults. With exception of the main San Andreas and Indio Hills faults, brittle faults are generally difficult to trace laterally but, where preserved, they display centimeter- to meter-scale strike-slip and/or reverse dip-slip displacement. Large-scale fault orientations and kinematics in sedimentary rocks are more variable than in basement rocks, but strike commonly WNW–ESE to N–S and show moderate–steep dips to the northeast (Fig. 2). Subsidiary meso-scale faults include high-angle SW- and SE-dipping strike-slip faults, and low-angle SW-dipping thrust faults. We describe the Indio Hills and main San Andreas faults, strike-slip faults, and thrust faults in sedimentary strata, and fractures in basement rocks northeast of the Indio Hills fault.

*Indio Hills and main San Andreas faults*

Along the Indio Hills fault, poor exposures make it difficult to measure fault strike and dip directly , but DEM images suggest a rectilinear geometry in map view relative to the uplifted sedimentary strata to the southwest (Fig. 2). The fault strikes mainly NW–SE and is subparallel to the northeastern flank of the Indio Hills. Farther southeast, it probably merges with the main San Andreas fault (Fig. 1; Tyler, 1974). In the southeastern part of the study area (Fig. 2), the Indio Hills fault is most likely located between an outcrop of basement granite and the first outcrops of overlying strata of the Palm Spring Formation. The granite

there is highly fractured and cut by vein and joint networks (see description below), as may be
expected in the damage zone of a major brittle fault.
Like the Indio Hills Fault, fault-plane dip and strike of the main San Andreas fault
must be inferred indirectly. The main San Andreas fault in the study area strikes WNW–ESE
and is sub-vertical based on its consistent rectilinear surficial trace, and because it truncates
both back- and fore-limb strata on most of the SAFZ-oblique macro-folds (Fig. 2). Thus, the
main San Andreas fault does not seem to have had major impact on the initial geometry and
development of the macro-folds in the Indio Hills. However, notable exceptions include
displacement by the main San Andreas fault of the two shear folds on the southern flank of
the macro-folds (Fig. 5), and a consistent anticlockwise bend of most axial traces of the
macro-folds (Fig. 2).
*Strike-slip faults in folded sedimentary strata*
One major brittle fault set striking NW–SE and dipping steeply to the northeast has
developed on the central macro-fold (Figs. 3b and 6). The faults splay out from a bedding-
parallel core zone subparallel to steeply SW-dipping mudstone–siltstone layers on the
southern limb of the central macro-fold, and then proceed to truncate NW-dipping
sedimentary strata and offset the hinge of a macro-fold by c. 70 meters right-laterally before
dying out (Supplement S7a–b). The fault damage zone is traceable for more than one
kilometer along strike as a right-slip fault which displaces the hinge of a major, tight,
asymmetric, shear-like (similar style) fold (Fig. 6 and Supplement S8). The shear-folded
sedimentary strata bend clockwise toward the main fault, thus supporting dominant right-
lateral slip (Fig. 6). Minor faults branch out from the fault core zone and either die out in the
macro-fold hinge, and/or persist as bedding-parallel faults for some distance on the southern
limb of the macro-fold (Fig. 6).
At smaller scale, the folded and tilted strata of the Palm Spring Formation are
commonly truncated by sets of steep NW–SE-striking right-lateral and NNE–SSW-striking
left-lateral faults, displaying meter- to centimeter-scale offsets (Supplement S7b–d). These
minor faults generally dip steeply to the northeast to east-northeast, i.e., opposite to most
bedding surfaces, which dip southwest (Fig. 3b), and, in places, develop reddish fault gouge
along strike. Furthermore, these minor faults typically cut sandstone beds and flatten, and/or
die out within, mudstone beds, which restricts their lateral extent to a few decimeters–meters.
Kinematic indicators, such as offset of bedding surfaces and fold axial surfaces, yield mostly
right-slip displacements, in places with minor reverse components. In some localities, on fold
limbs within thick and competent sandstone beds, such minor right- and left-slip faults appear
to form conjugate sets (Supplement S7b and d) that may have developed simultaneously. In
addition, NNE–SSW-striking, ESE-dipping faults and/or semi-brittle kink bands sub-
orthogonal to the SAFZ are well displayed in the southeastern macro-fold (Fig. 3c and
Supplement S7e) and cut bedding surfaces at high angles with left-slip displacement,
therefore potentially representing cross faults between segments/splays of the SAFZ system.
*Reverse and thrust faults in folded sedimentary strata*

Reverse and thrust faults are common and traceable on the back-limb of the central

and southeastern macro-folds near the SAFZ-parallel anticline and the Indio Hills fault, but
not recorded in areas close to the main San Andreas fault. Reverse faults strike mainly NW–
SE and dip gently to the southwest, although subsidiary gently NE-dipping faults exist. An
example is the low-angle reverse fault that propagates out-of-the syncline on the southeastern
macro-fold (Fig. 4) and yields a NE-directed sense-of-shear. This thrust fault may continue
westward into the central macro-fold (Fig. 3b), where reverse offset of SW-dipping strata of
the Palm Spring Formation constrains vertical displacement from about 10–15 meters
(Supplement S5a), though offset is only of a few centimeters in the southeast (Fig. 4). This
fault system has a listric geometry, and internal splay faults die out in thick silt- to mud-stone
layers. The low-angle faults seem to develop almost consistently near major fold hinge zones
and propagate northeastward as out-of-the syncline thrusts (Fig. 4 and Supplement S5a).

In sandstone beds on the north-dipping limb of the major syncline, minor-scale thrust

faults, offset asymmetric fold hinges (Supplement S7c) and yield down-to-the-north (normal)
sense of shear if the strata are rotated to a horizontal position (Supplement S9). An opposite
effect is apparent for a conjugate set of minor normal faults in a small-scale graben structure
on the steep, north-dipping layer, which defines a set of reverse faults when rotating the
sedimentary strata to horizontal (Supplements S7d and S9).
*Fractures and faults in basement rocks north of the Indio Hills fault*

Basement-rock exposures in the Indio Hills are limited to a single, approximately 50-

m long chain of outcrops located in the southeasternmost part of the study area (Fig. 2). These
outcrops of massive granite are heavily fractured with mostly steep to sub-vertical sets that
strike dominantly NE–SW to ENE–WSW and subsidiary NW–SE to NNW–SSE (see
stereoplot in Fig. 2). Kinematic indicators are generally lacking, but in highly fractured areas,
centimeter-thick lenses of unconsolidated reddish gouge are present, comparable to fault
rocks observed in Palm Spring Formation sedimentary rocks and corresponding to similar
small-scale strike-slip and reverse faults in the basement granite. The fault sets in granitic
basement rocks trend parallel to fault sets in sedimentary strata southeast of the Indio Hills

fault (see stereoplots in Figure 2) and are therefore suggested to have formed due to similarly oriented stress.

**Discussion**

*Structural evolution of SAFZ-oblique folds*

We mapped and analyzed three macro-scale fold systems between the Indio Hills and main San Andreas faults. In map view (Fig. 2), the folds are right-stepping, and each fold set is increasingly asymmetric (Z-shaped) and sigmoidal towards the Indio Hills fault in the northeast. Based on these properties, we interpret the fold sets as modified SAFZ-oblique *en echelon* macro-folds. Various investigators (Babcock, 1974; Miller, 1998; Titus et al., 2007; Janecke et al., 2018; Bergh et al., 2019) describe similar fold geometries in sedimentary strata from many other segments of the SAFZ and are interpreted as structures formed by right-lateral displacement between two major fault strands. However, the present fold-orientation data in the Indio Hills (Fig. 2) do not correspond with a uniform simple shear model in between two active strike slip faults because the long axis of the strain ellipse is not consistently about 45° to the shear zone as expected (Sanderson and Marchini, 1984; Sylvester, 1988). Instead, fold geometries vary both across and along strike, e.g., axial surface traces of dying-out macro-fold hinges are at high obliquity angles (> 50–65°) in the southwest, whereas they are at much lower angles (< 20–30°) and merge with sigmoidal-shaped patterns against the Indio Hills fault (Fig. 2). Thus, we propose that the SAFZ-oblique macro-folds in Indio Hills rather evolved from a single boundary fault (Indio Hills fault) being progressively more active through time. For example, a model in which the folds initially splayed out from an early active Indio Hills fault through right-lateral distributed displacement (compare with Titus et al., 2007) is consistent with fold hinges extending outward south of the Indio Hills fault and dying out (broadening) away from the fault in a several kilometer-wide damage zone (Fig. 2). Fold propagation outward from the Indio Hills fault is supported by the increased structural complexity of the fold geometries towards the Indio Hills fault. Furthermore, the initial upright, *en echelon* folding clearly occurred after deposition of the entire Palm Spring Formation because of the involvement in folding of the Bishop Ash and of adjacent strata possibly of the Ocotillo Formation (i.e., maximum age of 0.76 Ma – earliest late Pleistocene; Fig. 2 and Table 1). Should the whole Ocotillo Formation be folded in the Indio Hills, the maximum age constraints could be narrowed to < 0.6–0.5 Ma based on magnetostratigraphic ages for the upper part of the Ocotillo Formation (Kirby et al.,

2007). By contrast, the main San Andreas fault truncates both limbs of the open-style, *en*
*echelon* folds (Fig. 2), which therefore indicates younger deformation along this fault.
The moderate–steep westward plunge of all three macro-folds (≥ 30°), however,
shows that the presumed initial horizontal fold hinges rotated into a steeper plunge. Such
steepening may be due to, e.g., progressive shortening above a deep-seated fault, a hidden
splay of the Indio Hills fault, or to an evolving stage of distributed shortening (folding)
adjacent to the master strike-slip faults (e.g., Bergh et al., 2019), with gradually changing
stress–strain orientation through time, and/or due to structural tilting in the hanging wall of
the Indio Hills fault. This kind of fold reworking favors a situation where the northwestern
and central macro-folds were pushed up and sideways (right-laterally), following the
topography and geometry of an evolving transpressional uplift wedge (i.e., a contractional
uplift formed synchronously with successively with simple shear transpression to balance
internal forces in a crustal-scale critical taper; Dahlen, 1990). The corresponding eastward-
tightening, enhanced shear folding, and recurrent SW-directed overturned geometries of the
central macro-fold on the back-limb of the southeastern macro-fold near the Indio Hills fault
(Fig. 3b) support this idea.
We propose a progressive model that changes from distributed (*en echelon* folding) to
partly partitioned, i.e., pure shear (shortening) plus simple shear (strike-slip) deformation
(Fig. 7), as inferred for other parts of the SAFZ, e.g., in the Mecca Hills (Bergh et al., 2019).
In this model, the tight to isoclinal fold geometries to the northeast (Fig. 3b) may account for
progressively more intense shortening near the Indio Hills fault, whereas coeval strike-slip
faulting affected the already folded and steeply dipping strata of the lower Palm Spring
Formation (Fig. 6). This model would favor shortening strain to have evolved synchronously
with renewed strike-slip shearing adjacent to the Indio Hills fault, and/or on a blind fault
below the contact between the Mecca Formation and overlying Palm Spring Formation,
because the Mecca Formation is much less deformed (Fig. 3c). Alternatively, the more mildly
deformed character of the Mecca Formation conglomerate may arise from its homogeneity,
which contrasts with alternating successions of mudstone–siltstone and sandstone of the Palm
Spring Formation prone to accommodating large amounts of deformation and to strain
partitioning. Regardless, such reshaping of *en echelon* folds is supported by analog modelling
(McClay et al., 2004; Leever et al., 2011a, 2011b) suggesting that partly partitioned strain
may lead to a narrowing of fold systems near a major strike-slip fault (i.e., Indio Hills fault),
whereas widening away from the fault indicates still ongoing distributed deformation. Partly
partitioned deformation is supported by the tight to isoclinal and consistent Z-like geometry of
smaller-scale folds present on the back-limb of the central and southeastern macro-folds (Fig.
3b–c), indicating that they are all parasitic folds and related to the same partly partitioned
shear-folding event. Where S- and Z-like fold geometries are present, these minor folds may
have formed by buckling in an early stage of *en echelon* folding. An alternative interpretation
is that the tight, reshaped parasitic folds are temporally linked to the SAFZ-parallel macro-
fold south of the Indio Hills fault (Fig. 3c; see next section).

*Structural evolution of SAFZ-parallel folds*
The SAFZ-parallel anticline differs significantly in geometry from the *en echelon*
macro-folds and associated parasitic folds by having an upright and symmetric geometry <
20° oblique to the Indio Hills fault. Thus, it resembles that of a fault-propagation fold in a
more advanced partitioned transpressional segment of the SAFZ (e.g., Titus et al., 2007;
Bergh et al., 2019). We suggest that this fold formed by dominant NE–SW-oriented
horizontal shortening, i.e., at high obliquity to the main Indio Hills fault (near-orthogonal pure
shear), and/or as a fault-related fold above a buried, major reverse (SW-dipping) oblique-slip
splay of the Indio Hills fault at depth (e.g., Suppe and Medwedeff, 1990). The timing might
be after the tight reworking of *en echelon* folds in the late Pleistocene, i.e., comparable to
other settings (e.g., western Svalbard; Bergh et al., 1997; Braathen et al, 1999). The idea of a
late-stage, highly oblique pure-shear overprint onto the macro-folds is supported by small-
scale upright folds located within the tight *en echelon* syncline on the back-limb of the
modified central macro-fold system (Fig. 4). The NW–SE trend, upright style, and negligible
plunge of the fold axes indicate that these folds may be superimposed on the steeper plunging
and reshaped *en echelon* folds, and/or that they formed in progression to an increased
component of NE–SW shortening on the Indio Hills fault. Nonetheless, these folds may have
formed simultaneously with the *en echelon* macro-folds in the (earliest?) late Pleistocene
(Table 1) due to uncertain (not fully understood) crosscutting relationships.
Progressive NE–SW-oriented contraction may have triggered formation of the upright
SAFZ-parallel anticline adjacent to the Indio Hills fault (Fig. 2 and 3c). The fault then acted
as a SW-dipping thrust fault with top-NE displacement. The oblique shortening then led to a
certain amount of uplift near the Indio Hills fault, and possibly also accomplished the
overturning of folds on the northeastern back-limb of the central and southeastern macro-fold.
A similar mode of advanced partitioned shortening was proposed for SAFZ-parallel fold
structures in central and southern California (Mount and Suppe, 1987; Titus et al., 2007;
Bergh et al., 2019). Our results are supported by stress orientation data acquired by Hardebeck
and Hauksson (1999) along a NE–SW-trending profile across the Indio Hills. They recorded
an abrupt change in the maximum horizontal stress direction from about 40° oblique to the
SAFZ around the main San Andreas fault, to about 70° oblique (i.e., sub-orthogonal) farther
northeast, near the Indio Hills fault, which supports the change in attitude and shape of macro-
fold geometries that we have outlined. Shortening and strike-slip partitioning, however, would
require synchronous right slip on another major fault strand, e.g., the main San Andreas fault,
a hypothesis that is supported by the recorded late-stage (i.e., late Pleistocene) shear folding
there (Fig. 5).

***Fold and fault interaction, evolution, and relative timing***

In this section we use the geometry and kinematics of folds and faults in the southern

Indio Hills to reconstruct the tectonic history of the area, not only of the inverted late
Cenozoic basin but also about strike-slip and dip-slip faults that bound the basin. Essential
tectonic events include (1) extensional normal faulting along the Indio Hills fault in the mid-
Miocene–Pliocene (ca. 15–3.0 Ma), (2) reactivation of the Indio Hills fault as a right-lateral to
oblique-reverse fault in the (earliest?) late Pleistocene to present-day ($< 0.76$ Ma), and (3)
right-lateral movement along the main San Andreas fault in the late Pleistocene to present-day
($< 0.76$ Ma; Table 1).

Prior to inversion and uplift of the Indio Hills, the Indio Hills fault most likely acted as

a SW-dipping, extensional, basin-bounding normal fault. Indications of an early-stage episode
of extension are shown by micro-fault grabens in steeply dipping layers (Supplements S5d
and S6), by the deposition and preservation of sedimentary strata of the Palm Spring and
Mecca formations southwest of the Indio Hills, whereas they were eroded or never deposited
northeast of the fault, and by fining upwards of the stratigraphic units from conglomerates in
the Mecca Formation to coarse-grained sandstone in the lower parts of the Palm Spring
Formation. In addition, the flat geometry of micro thrust faults (e.g., Supplements S5b–c)
suggests that they were rotated during macro-folding. Restoration of all micro faults in their
initial position prior to macro-folding shows that some of these faults exhibit normal
kinematics with associated syn-tectonic growth strata (Supplements S5d and S9).
Alternatively, the Indio Hills fault dips northeast and uplifted the granitic basement rocks in
the hanging wall to the northeast, followed by erosion of the overlying Mecca, Palm Spring
and Ocotillo formations there (Fig. 1). We favor a basin geometry and formation similar to
that of the Mecca Hills, where down-SW slip along the Painted Canyon fault was inferred in
the (Miocene?–) Pliocene (McNabb et al., 2017), and of the transtensional Ridge Basin
though having opposite vergence (Crowell, 1982; Ehman et al., 2000) with a steep, SW-
dipping normal fault that was progressively reactivated as an oblique-slip reverse/thrust fault
during basin inversion. Formation of the Indio Hills fault as a normal fault probably occurred
in mid-Miocene times during extension related to the opening of the Gulf of California (Stock
and Hodges, 1989; Stock and Lee, 1994) as proposed for the Salton Trough (Dorsey et al.,
2011 and references therein).
Right-lateral to right-lateral-reverse movement along the Indio Hills fault that led to
the formation of the SAFZ-oblique *en echelon* macro-folds also supports a steeply dipping
character for the precursory Indio Hills fault. The change to a right-lateral-reverse fault is
further supported by the presence of both meso-scale strike-slip and thrust faults having
similar NW–SE strikes (Fig. 4, and Supplements S4c and S5a). The increased reverse (and
decreasing right-lateral) component of faulting may have triggered rotation of the *en echelon*
macro-fold axes to a steeper plunge, reshaped the open asymmetric folds into tight overturned
folds, and caused gentle buckling of strata in the nearby SAFZ-parallel anticline. Hence, the
Indio Hills fault ultimately functioned as an oblique-slip thrust oblique to the convergent plate
boundary in the late Pleistocene, which is supported by oblique maximum horizontal stress
near the Indio Hills fault (c. 70°; Hardebeck and Hauksson, 1999), while the main San
Andreas fault simultaneously accommodated right slip during this period.
By contrast, the last episode of movement along the main San Andreas fault clearly
postdates *en echelon* folding, from its truncating attitude (Fig. 2). In addition, the
anticlockwise bending of the axial traces into an ENE–WSW trend towards the southwest
suggests that a distributed component of off-fault deformation affected the area around the
main San Andreas fault in its early kinematic stages in the late Pleistocene. The refolding of
the southwest limb of the central macro-fold near the main San Andreas fault (Fig. 5) also
favors a late-stage activation of this fault in the late Pleistocene (i.e., after the initial
transpressional slip events along the Indio Hills fault in the – earliest? – late Pleistocene).
Possibly as a consequence of a longer period of activity, and as suggested by relatively higher
topographic relief and more intensely folded sedimentary strata in the vicinity of and along
the Indio Hills fault than along the main San Andreas fault, the former probably
accommodated significantly larger amounts of uplift than the latter. This implies a southwest-
tilted geometry for the Indio Hills uplift.
Minor faults in the Indio Hills provide additional input to resolve the spatial, temporal
and kinematic relations between macro-fold and fault interaction. We analyzed minor fault-
related folds (Supplement S5c), which, in their current position on steep north-dipping beds,
define down-to-the north displacement. However, when rotating the sedimentary strata to
horizontal (Supplement S9), the fault-related folds define a low-angle fold-and-thrust system.
These geometric relationships suggest that the minor folds and faults (other than right-slip
faults) pre-date (or were coeval with) the SAFZ-oblique macro-folding event, and that they
formed initially as internal fractures due to N–S-oriented shortening when the sedimentary
strata were still horizontal. This implies that some partitioning (e.g., SAFZ-parallel small-
scale thrust faults) occurred simultaneously with distributed deformation (e.g., SAFZ-oblique
*en echelon* macro-folds).

Further, our field data suggest that minor right-slip faults evolved synchronously and

parallel with the E–W-trending *en echelon* fold limbs, propagating through rheologically
weaker mudstone beds that flowed plastically and acted as slip surfaces during distributed
deformation. Later or simultaneously, these faults escaped from the mudstone beds and
propagated as NW–SE-striking right-slip faults adjacent to tightened shear folds during partly
partitioned deformation, and finally ended up with truncation of the SAFZ-oblique folds (Fig.
6 and Supplement S7a–c).

The presence of out-of-the syncline reverse/thrust faults relative to the reshaped and

tightened SAFZ-oblique macro-folds (Fig. 4 and Supplement S5a and d), where SW-dipping
thrust faults formed (sub-) parallel to the Indio Hills fault, and the related upright anticline
(Fig. 3c) suggest successive distributed and partly partitioned strain in the study area. The
proximity and superimposed nature of reverse/thrust faults relative to the reshaped *en echelon*
folds suggest that they utilized modified fold hinges and steeply tilted limbs as preexisting
zones of weakness. Despite the uncertainty around the crosscutting relationship between the
SAFZ-parallel anticline and the SAFZ-oblique *en echelon* macro-folds, the low-angle thrust
and intra-detachment folds in the southeastern macro-fold (Fig. 4) indicate that such thrust
detachments may have already formed during (early?) distributed deformation, i.e., that
distributed and partitioned deformation occurred simultaneously and/or progressively (see
phases 1 and 2 in Table 1).

The conjugate WNW–ESE- to NNW–SSE-striking right-slip and NNE–SSW-striking

left-slip faults and kink band features truncate strata on both macro-fold limbs (Fig. 3b–c)
with an acute angle perpendicular to the macro-folded and tilted Palm Spring Formation strata
(e.g., Supplement S7e). Thus, they formed together with or after the *en echelon* macro-folding
($< 0.76$ Ma).

*Tectonic model*

In this section we use detailed structural analysis of folds and faults in the southeastern Indio Hills to outline the structural history of the tectonic uplift itself, evaluate it in terms of what is known about strain budgets within the southern San Andreas fault system, link it to nearby structures (Eastern California shear zone and Landers Mojave Line), and integrate the local structural history into a structural synthesis for the southern San Andreas Fault zone in the past 4 Myr.

Our field and structural data support inversion and uplift of the Indio Hills involving progressive or stepwise stages of folding and faulting, incorporating a switch from distributed to partly partitioned transpression (Fig. 7). Prior to inversion in late Pleistocene time, the Indio Hills fault may have been a steep, SW-dipping normal fault that downthrew (Miocene?–) Pliocene sedimentary strata against granitic basement rocks in its footwall to the northeast. These basement rocks were partly eroded in the footwall of the fault. In the hanging wall of the fault, strata of the Mecca Formation were deposited in the Pliocene, most likely at 3.7–3.0 Ma, and the lower and upper members of the Palm Spring Formation respectively at 3.0–2.3 Ma and 2.6–0.76 Ma, as suggested from paleomagnetic studies in the Mecca Hills (Chang et al., 1987; Boley et al., 1994; McNabb et al., 2017).

Early inversion involved distributed transpressional strain triggered by right-lateral slip along the Indio Hills fault (Fig. 7a). Three macro-scale, upright *en echelon* folds and associated parasitic folds formed in loosely consolidated sedimentary rocks of the Mecca and Palm Spring formations after the latter was deposited (< 0.76 Ma), i.e., probably in earliest late Pleistocene time (Table 1). The fold set evolved oblique to the main strand of the SAFZ and formed a right-stepping pattern of E–W-oriented axial surfaces that trend at a high angle (45°) to the bounding Indio Hills fault due to uniform simple shear (e.g., Sanderson and Marchini, 1984; Sylvester, 1988). This is notably observed in the less deformed southwestern part of the study area (Fig. 2) near the main San Andreas fault, where the macro-folds still display their initial non-plunging geometries. Bed-internal minor fold and fault systems in weak mudstone beds (Fig. 4 and Supplement S5a) may have formed parallel to the E–W-trending *en echelon* fold traces, either as thrust detachments due to oblique N–S shortening when strata were horizontal, and/or as strike-slip faults on the fold limbs. In addition, minor (bed-internal) SAFZ-parallel thrusts and folds formed prior to or together with the *en echelon* macro-folds (Supplements S4b–c and S9a–b), thus suggesting minor strain partitioning.

Further deformation in the late Pleistocene led to gradual change from mostly distributed with minor partitioned deformation to partly partitioned shortening and right-

lateral faulting and folding (Fig. 7b), probably since the Indio Hills fault started to
accommodate an increasing amount of reverse slip, thus acting as an oblique-slip right-lateral-
reverse fault, and where the main San Andreas fault did not yet play a major role. The main
result was tightening of the macro-folds toward the Indio Hills fault and clockwise rotation of
fold axes to a steeper westerly plunge due to increased shear folding, whereas *en echelon*
upright folding continued in the southwest (Fig. 7b). Increased shortening and shearing
reshaped the macro-folds and their back-limb folds to tight, isoclinal, and partly overturned
folds with consistent Z-style and sigmoidal axial-surface traces near the Indio Hills fault (Fig.
7b). The sigmoidal pattern of the WNW–ESE-trending *en echelon* macro-folds formed at a
much lower angle with the Indio Hills fault (< 20–30°) than farther southwest (60–70$^0$).
Furthermore, the incremental component of lateral strain is recorded as progressively
crosscutting NW–SE-striking, strike-slip shear faults terminating with local truncation of the
central macro-fold (see Fig. 7c and section below).Uplift of the Indio Hills in the late
Pleistocene (because the earliest late Pleistocene 0.765 Ma Bishop Ash is involved in folding;
Sarna-Wojcicki et al., 2000; Zeeden et al., 2014) was marked by a gradual switch to more
kinematically evolved transpressional strain partitioning, where the dominant shortening
component was accommodated by right-lateral-oblique, top-NE thrusting along the Indio
Hills fault and major strike-slip movement along the main San Andreas fault (Fig. 7c and
phase 3 in Table 1). NE-directed oblique thrusting on the Indio Hills fault and related minor,
reverse, out-of-the syncline faults led to uplift, which resulted in formation of a major
anticline parallel to the Indio Hills fault in sediments of the Mecca Formation (see anticline
closest to Indio Hills fault in Fig. 3c and 7c). With increasing partitioning, slip parallel to the
convergent plate boundary was accommodated by right slip along the linear main San
Andreas fault, where sub-vertical folds formed locally, and presumed antithetic conjugate
kink band sets of right- and left-slip cross faults affected the entire uplifted area.

We favor a progressive evolution from distributed to partly partitioned deformation as

presented in Fig. 7a–c, although overlapping and synchronous formation of various structures
may have occurred (overlapping of phases 1 and 2 in Table 1), at least locally (except for the
late-stage main San Andreas fault and related shear folds; phase 3 in Table 1). The
overlapping and synchronous formation of structures is based on uncertainties in our field
data, e.g., variable cross-cutting relations of early, bedding-parallel strike-slip and thrust faults
and *en echelon* macro-folds (Figs. 4 and 6, and Supplements S5c–d and S7), and from the
spatial variations in the direction of maximum horizontal stress across the Indio Hills at
present, from 40° oblique to the boundary faults near the main San Andreas fault to 70°
oblique near the Indio Hills fault (Hardebeck and Hauksson, 1999).
Our observations of mostly lateral movement along the main San Andreas fault (i.e.,
southeastern continuation of the Mission Creek fault) and the proposed late Pleistocene to
present-day age for deformation in the southeastern Indio Hills are consistent with work by
Keller et al. (1982). A major difference between the northwestern and southeastern Indio Hills
is the relatively tighter macro-folding over a narrower area and more intense character of
deformation in between the two bounding faults in the southeastern Indio Hills (Figs. 2 and 3;
Keller et al., 1982; Lancaster et al., 2012).
The right-lateral-reverse character of the Indio Hills Fault and its role in our kinematic
model for basin inversion in the southern Indio Hills are further supported by the relationship
of the Indio Hills fault with the Eastern California shear zone, which merge together north of
the study area where the Indio Hills fault bends into a NNW–SSE strike along the Landers–
Mojave Line (Dokka and Travis, 1990a, 1990b; Nur et al., 1993a, 1993b; Thatcher et al.,
2016). Recent activity along the Landers–Mojave Line recorded as six–seven earthquakes
with M > 5 between 1947 and 1999 (Fig. 1; Nur et al., 1993a, 1993b; Du and Aydin, 1996;
Spinler et al., 2010) indicates that a through-going NNW–SSE-striking fault crosscuts the
Pinto Mountain fault (Nur et al., 1993a, 1993b; Rymer, 2000). Notably, the 1992 Joshua Tree
earthquake occurred along the NNW–SSE-striking, west-dipping West Deception Canyon
fault (Rymer, 2000 and references therein), which merges with the (probably southwest-
dipping) Indio Hills fault in the south (see figure 1 in Rymer, 2000). Therefore, we propose
that the Indio Hills fault, may be one of several faults to transfer displacement from
unsuitably oriented, NW–SE-striking right-slip faults in the north, such as the Calico and
Camp Rock faults, to the main SAFZ strand in the south (Fig. 1).
Farther southeast along strike, the Indio Hills and main San Andreas faults merge
along a dextral freeway junction, i.e., a junction of three dextral fault branches (sensu Platt
and Passchier, 2016 and Passchier and Platt, 2017), which may have enhanced wedge-shaped
transpressional uplift of the Indio Hills after the late formation of the main San Andreas fault
in the late Pleistocene (Fig. 8a–c and Table 1). However, anticlockwise rotation of the Indio
Hills block and related structures in map view as predicted in a dextral freeway junction (Platt
and Passchier, 2016; Passchier and Platt, 2017) was not recorded by our field data (except
along the main San Andreas fault due to localized right-slip along the fault; cf. sub-vertical
shear fold in Fig. 5). This may be due in part to the late formation of the main San Andreas
fault (< 0.76 Ma, i.e., late Pleistocene), i.e., clockwise rotation (in map view) of the fold and
fault structures due to right-lateral slip along the Indio Hills fault, and to the oblique-slip
character of the Indio Hills fault. Thus, the dextral freeway junction in the Indio Hills may be
more of a transitional nature. Instead of major anticlockwise rotation of the Indio Hills block
in map view, the accretion of material toward the fault junction due to right slip along the
main San Andreas fault is probably partly accommodated by the dominant vertical slip
component along the Indio Hills fault, leading to further uplift near the junction (i.e.,
clockwise rotation in cross section).

*Regional comparison and implications*
The proposed progressive tectonic model for the Indio Hills uplift has wide
implications when compared and correlated with other fault strands of the SAFZ bounding
uplifted domains along strike in the Coachella and Imperial valleys (Fig. 8a–c), and in
explaining lateral variations in fault architectures, kinematic evolution and timing,
deformation mechanisms and areal segmentation (Sylvester and Smith 1987; McNabb et al.,
2017; Janecke et al., 2018; Bergh et al., 2019). Here we compare and contrast the structural
evolution of the southeastern Indio Hills with that of nearby tectonic uplifts (Mecca Hills and
Durmid Hills).
*Comparison with the Mecca Hills*
Previous studies of SAFZ-related uplifts between the Indio Hills and Durmid Hills in
Coachella Valley suggest that the Indio Hills and main San Andreas faults link up in the
southeasternmost Indio Hills and proceed as the main San Andreas fault in the Mecca Hills
(Fig. 8c) which then, together with the subsidiary Skeleton Canyon and Painted Canyon
faults, bounds a much wider flower-like uplift area than in the Indio Hills (Fig. 8c; Sylvester
and Smith, 1976, 1987; Sylvester, 1988; McNabb et al., 2017; Bergh et al., 2019). In contrast
to the Indio Hills fault, however, the main San Andreas fault in Mecca Hills has an
anastomosing geometry with thick (10–500 m), red-stained fault gouge. Regardless, we
consider these faults to be correlative and infer the lack of fault gouge along the Indio Hills
fault to be due to more localized strain on the Indio Hills fault than on the SAFZ in Mecca
Hills. This is supported by a more rectilinear geometry and lack of fold–fault linkage in Indio
Hills, which may have allowed initial lubrication of the fault surface in basement rocks with
high contrasting rheology (e.g., Di Toro et al., 2011; Fagereng and Beall, 2021), and which
hampered fluid circulation and extensive cataclasis. Another possible explanation may be the
presence of coarse-grained deposits of the Mecca Formation, which may have

partitioned/decoupled deformation along the Indio Hills fault from that in overlying Palm Spring sedimentary strata.

Both the Indio Hills and Mecca Hills uplift areas are bounded to the northeast by a presumed Miocene–Pliocene, SW-dipping normal fault (Fig. 8a), which later acted as major SAFZ-parallel oblique-reverse faults, and which significantly contributed to the uplift of these areas in (late) Pleistocene time (Sylvester and Smith, 1976, 1987; McNabb et al., 2017; Bergh et al., 2019). In the Mecca Hills (Fig. 8c), the Painted Canyon fault is flanked in the hanging-wall to the southwest by a basement-cored, macro-fold (Mecca anticline), which is similar to the upright anticline that parallels the Indio Hills fault and adjacent minor thrust faults (Figure 2 & Figure *3*c and Supplement S5a). Similar folds appear adjacent to the Hidden Springs–Grotto Hills fault (Sheridan et al., 1994; Nicholson et al., 2010), a NW–SE-striking, now reverse splay fault of the main SAFZ between the Mecca Hills and Durmid Hills (Fig. 8c). It is, however, unlikely that these marginal faults link up directly along strike. Rather, they merge or splay with the SAFZ and SAFZ-oblique faults.

The inversion and main uplift history of the Mecca Hills segment of the SAFZ (Bergh et al., 2019) initiated with right-lateral slip on a steep SAFZ, from where SAFZ-oblique *en echelon* folds and dominantly right-slip faults splayed out in a one–two kilometers wide damage zone on either side of the SAFZ (Fig. 8a). The subsidiary Skeleton Canyon fault initiated as a steep right-lateral and SAFZ-parallel strike-slip fault along a small restraining bend (Fig. 8b). Successive lateral shearing reshaped the *en echelon* folds into steeply plunging folds with axial traces parallel to the SAFZ. The final kinematic stage generated SW-verging fold and thrust structures parallel to the SAFZ (Fig. 8c), which truncated the *en echelon* folds and the NE-dipping Skeleton Canyon fault. The resulting wedge-like flower structure thus records a polyphase kinematic evolution from distributed, through locally partitioned, to fully partitioned strain (Bergh et al., 2019).

Based on the geometric similarities, we consider that the *en echelon* macro-folds in both Indio Hills and Mecca Hills formed coevally, but not on the same regional right-lateral fault strand (Fig. 8a). In both areas, the *en echelon* folds and faults are strongly reworked and tightened into sigmoidal shapes where they merge with the Indio Hills and Skeleton Canyon faults respectively (Fig. 8b; Bergh et al., 2019), and SAFZ-parallel thrust faults formed early (i.e., prior to macro-folding) both in the Indio Hills (Supplement S5c–d) and in the Mecca Hills (Rymer, 1994), thus supporting continuous, partly partitioned strain field in both areas. Strain partitioning caused major uplift of the Mecca Hills block along the Skeleton Canyon, Painted Canyon, and Hidden Springs–Grotto Hills faults (Fig. 8c), all acting as SAFZ-parallel

oblique-slip thrust faults (Sheridan et al., 1994; Bergh et al., 2019). The partitioned right-slip
component was partly transferred to the main San Andreas fault in Indio Hills, and/or to an
unknown hidden fault southwest of the SAFZ (e.g., in Mecca Hills; Hernandez Flores, 2015;
Fuis et al., 2017), possibly the Eastern Shoreline fault (Janecke et al., 2018).

Based on paleomagnetic and structural field studies, uplift of the SAFZ-related Mecca

basin started at ca. 2.6–0.76 Ma (i.e., earliest to mid Pleistocene) with partial and local erosion
of the Palm Spring Formation (see lower and upper unconformities in McNabb et al., 2017)
and culminated after 0.76 Ma (see unconformity between the uppermost Palm Spring
Formation and base of the Ocotillo Formation southwest of the Painted Canyon fault in
McNabb et al., 2017), i.e., after deposition of the whole Palm Spring Formation (McNabb et
al., 2017; Janecke et al., 2018). Uplift is still ongoing at present (Fattaruso et al., 2014;
Janecke et al., 2018). Fault activity and tectonic uplift of the Mecca Hills therefore most likely
initiated earlier (earliest Pleistocene) than in the Indio Hills (earliest late Pleistocene; Table
1), where the transition from the lower to the upper member of the Palm Spring Formation is
gradual and does not show any major unconformity.
*Comparison with Durmid Hills*

The Durmid ladder structure along the southern 30 kilometers of the SAFZ in Imperial

Valley defines a similar but oppositely merging, one–three kilometers wide wedge-shaped
uplift as in Indio Hills, bounded by the right-lateral and reverse Eastern Shoreline fault to the
southwest and the main SAFZ to the northeast (Fig. 8c; Janecke et al., 2018). Internally, the
ladder structure comprises *en echelon* folds (Babcock, 1974; Bürgmann, 1991) that merge in a
sigmoidal pattern with the main SAF, and subsidiary sets of conjugate SAFZ-parallel right-
lateral and SAFZ-oblique E–W-striking, left-slip cross faults, which accommodated clockwise
rotation of internal blocks (Janecke et al., 2018). The *en echelon* folds formed at a comparable
time, i.e., < 0.76 Ma in the Indio Hills and at ca. 0.5 Ma in the Durmid Hills (Table 1). By
assuming a northwest continuation of the main SAFZ with the SAFZ in Mecca Hills, the
Eastern Shoreline fault has no exposed correlative fault in the Mecca Hills and Indio Hills
(Fig. 8c; Damte, 1997; Bergh et al., 2019). Nevertheless, the Eastern Shoreline fault may
continue at depth southwest of the main San Andreas fault (Janecke et al., 2018).

A significant difference between the Indio Hills–Mecca Hills and the Durmid Hills,

however, is the large number of cross faults in the Durmid ladder structure. Such faults are
interpreted as early-stage (ca. 1 Ma – early/mid Pleistocene), NE–SW-striking, left-lateral,
faults (Fig. 8a), which were rotated clockwise by progressive right-lateral motion into
sigmoidal parallelism with the SAFZ and Eastern Shoreline fault (Fig. 8b–c; Janecke et al.
2018). In contrast, cross faults in Indio Hills are much less common and, where present,
probably formed late, but prior to the main San Andreas fault (i.e., in the earliest or middle
part of the late Pleistocene). Thus, in the Indio Hills, there is no evidence of clockwise
rotation of early-stage cross faults as in the Durmid Hills, but rather clockwise rotation of fold
axial traces is common, which may be a first step in the formation of ladder-like fault blocks
(e.g., Davis, 1999; Schultz and Balasko, 2003).

A major outcome of the comparison with Durmid Hills is that the wedge-shaped uplift

block between the Indio Hills and main San Andreas faults may represent a failed uplift
and/or the early stage of formation of a ladder structure. This idea is supported by presence of
similar master faults and structures with comparable kinematics in both the Indio Hills and
Durmid Hills, including oblique *en echelon* macro-folds, strike-slip faults acting as step-over
faults, and reverse faults. Younger, non-rotated, conjugate cross faults exist in the Indio Hills
but not in the Durmid Hills where such faults are more evolved features due to larger strain
and more advanced stage of ladder structure formation. From these observations, one should
expect to find ladder structures operating at different evolution stages among the many, yet
unexplored uplifts in Coachella Valley.

**Conclusions**
1) The Indio Hills fault likely initiated as a SW-dipping, basement-seated normal fault
during the opening of the Gulf of California in the mid Miocene, and was later
inverted as a right-lateral reverse, oblique-slip fault in the (earliest?–) late Pleistocene
due to transpression along the convergent plate boundary, whereas the main San
Andreas fault initiated probably as a dominantly right-slip fault during the later stages
of uplift in the late Pleistocene.
2) The Indio Hills segment of the SAFZ in Coachella Valley, southern California evolved
as a wedge-shaped uplift block between two major SAFZ-related fault strands, the
Indio Hills and main San Andreas faults, which merge in a dextral freeway junction of
a transitional nature to the southeast.

3) Transpressive deformation triggered uplift and inversion of the Indio Hills through a
progressive change from distributed *en echelon* folding to partly partitioned right-slip
thrusting. We favor a progressive rather than stepwise model in which the main uplift
was related to late shortening at the freeway junction where the Indio Hills and main
San Andreas faults merge.

4) The Indio Hills fault is a splay fault of the SAFZ that merges to the north with the Landers–Mojave Line and contributes to transfer slip from unsuitably oriented faults of the Eastern California shear zone to the main San Andreas fault in the southeast.

5) A significant difference of the Indio Hills with the Durmid Hills is that left-lateral step-over and cross faults in the Durmid Hills rotated subparallel with the mSAF, whereas in Indio Hills, all cross faults are oblique with the SAFZ and, thus, may reflect an earlier stage of a still evolving ladder structure.

6) The initiation of right-lateral to right-lateral-reverse slip along major SAFZ-parallel faults and the main San Andreas fault in the Coachella Valley is younger towards the northwest (Pliocene in the Durmid Hills, early Pleistocene in the Mecca Hills and late Pleistocene in the Indio Hills). The onset of uplift, however, appears to be coeval in all tectonic uplifts (late to latest) Pleistocene.

**Data availability**

The structural dataset and field photographs used in the present study are available on DataverseNO (Open Access repository) at https://doi.org/10.18710/TM18UZ. DEM images are from Google Earth (© Google Earth 2011).

**Authors contribution**

All authors contributed to collect structural measurements in the Indio Hills. JBPK wrote the first draft of the manuscript and designed half the figures and supplements (workload: 35%). Prof. SGB made major revision to the initial draft and designed half the figures and supplements (workload: 35%). Prof. AGS also revised the manuscript and provided major input about the local geology (workload: 30%).

**Competing interests**

The authors declare that they have no known competing interests.

**Acknowledgments**

The staff at the University of California–Santa Barbara and San Diego State University provided great hospitality during Steffen Bergh's sabbatical leaves in 2011–2012 and 2016–2017 while working with the San Andreas fault. We thank all the persons from these institutions that were involved in this project. The authors thank Prof. Emeritus Arild Andresen (University of Oslo) and Prof. Holger Stunitz (UiT) for helpful comments and Jack

Brown (San Diego State University) for fieldwork collaboration. Prof. Susanne Janecke (Utah State University) and Dr. Miles Kenney (Kenney Geoscience) provided fruitful discussion. We thank the reviewers (Dr. Jonathan Matti and an anonymous reviewer) for their extended helpful comments on the manuscript.

**Financial support**

The present study is part of the CEED (Centre for Earth Evolution and Dynamics) and ARCEx projects (Research Centre for Arctic Petroleum Exploration), which are funded by grants from UiT The Arctic University of Norway in Tromsø and the Research Council of Norway (grant numbers 223272 and 228107) together with eight academic and six industry partners.

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

**Figures**

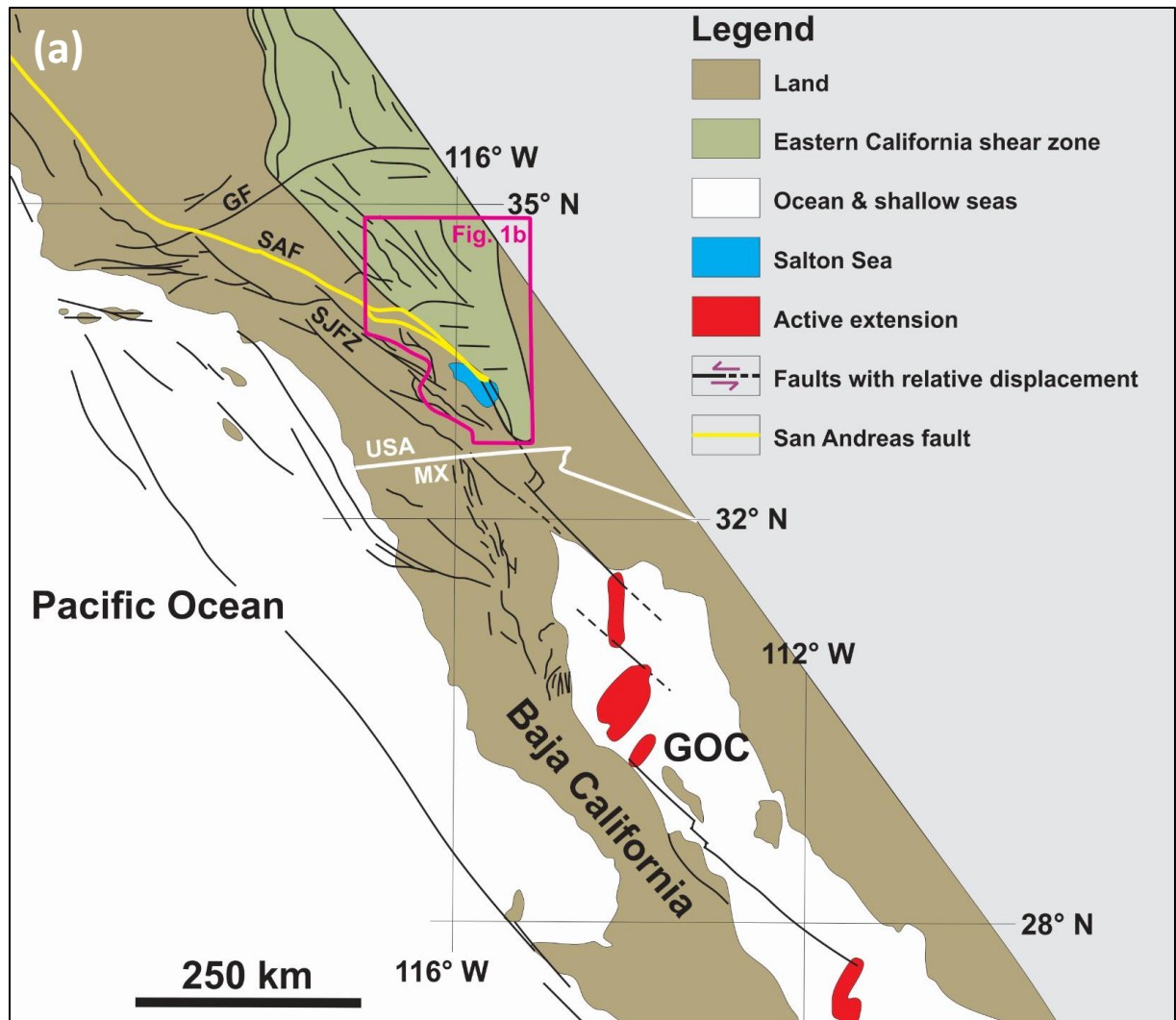


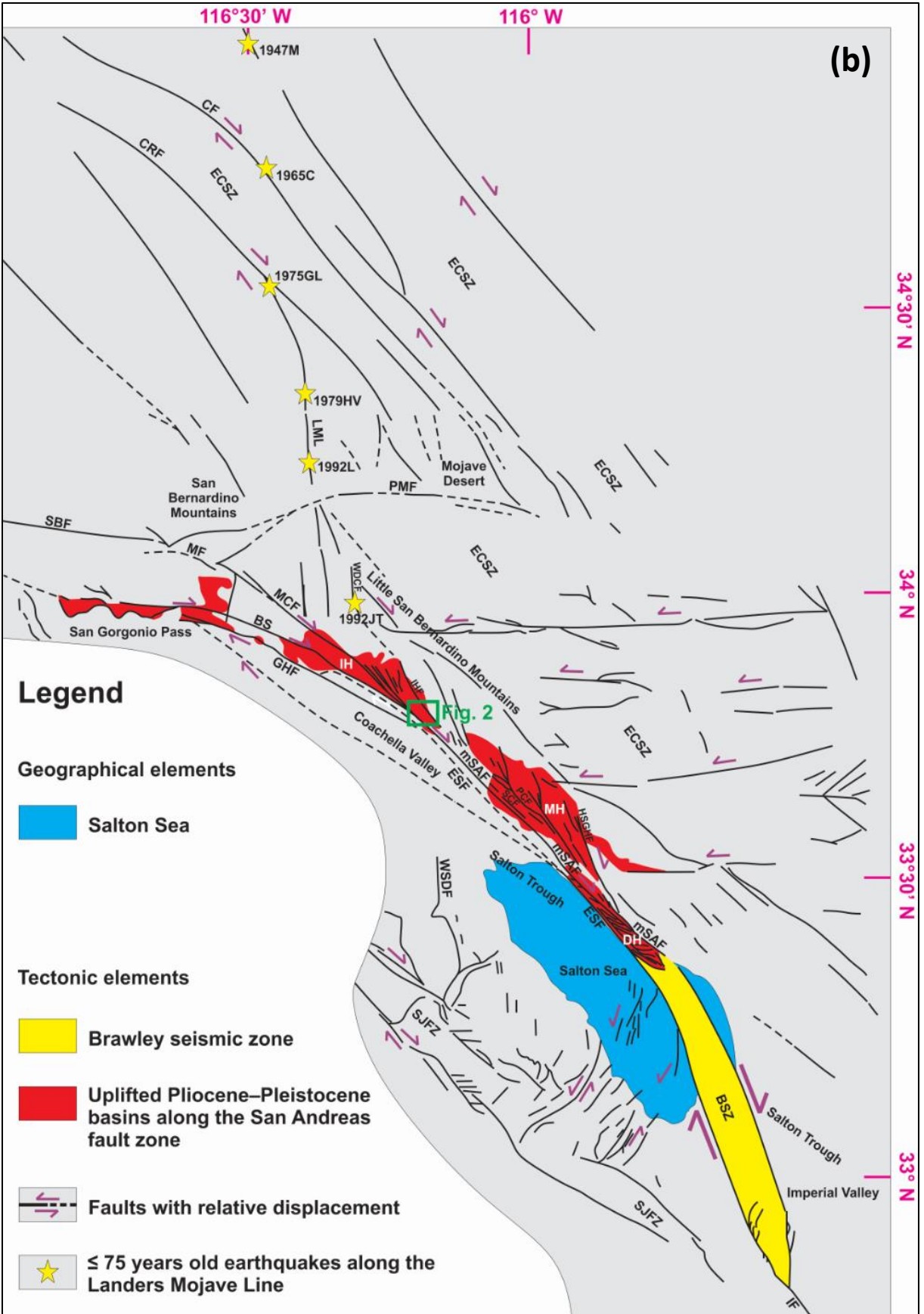


Figure 1: (a) Map of the main geological features of southern California, Baja California and the Gulf of California. The location of (b) is shown as a fuchsia polygon. Modified after Janecke et al. (2018). (b) Simplified geological map of the Coachella Valley and Salton Trough, southern California, showing the three main transpressional uplift areas along the SAFZ: the Indio Hills (IH), Mecca Hills (MH), and Durmid Hills (DH). Note the link of the SAFZ with the Brawley seismic zone to the south. The study area is shown in a green rectangle. Recent earthquakes (≤ 75 years) along the Landers–Mojave Line (LML) are shown as yellow stars with associated year of occurrence. Faults are drawn after Rymer (2000), Guest et al. (2007), Janecke et al. (2018), and Bergh et al. (2019). Earthquakes after Nur et al. (1993a, 1993b). Abbreviations: 1947M: 1947 Manix earthquake; 1965C: 1965 Calico earthquake; 1975GL: 1975 Galway Lake earthquake; 1979HV: 1979 Homestead Valley earthquake; 1992JT: 1992 Joshua Tree earthquake; 1992L: 1992 Landers earthquake; BS: Banning strand; BSZ: Brawley seismic zone; CF: Calico fault; CRF: Camp Rock fault; DH: Durmid Hills uplift; ECSZ: East California Shear Zone; ESF: Eastern Shoreline fault; GF: Garlock fault; GHF: Garnet Hill fault; GOC: Gulf of California; HSGHF: Hidden Springs–Grotto Hills fault; IF: Imperial fault; IH: Indio Hills uplift; IHF: Indio Hills fault; LML: Landers Mojave Line; MCF: Mission Creek fault; MF: Mill Creek fault; MH: Mecca Hills uplift; mSAF: main San Andreas fault; PCF: Painted Canyon fault; PMF: Pinto Mountain fault; SBF: San Bernardino fault; SCF: Skeleton Canyon fault; SJFZ: San Jacinto fault zone; WDCF: West Deception Canyon fault; WSDF: West Salton detachment fault.

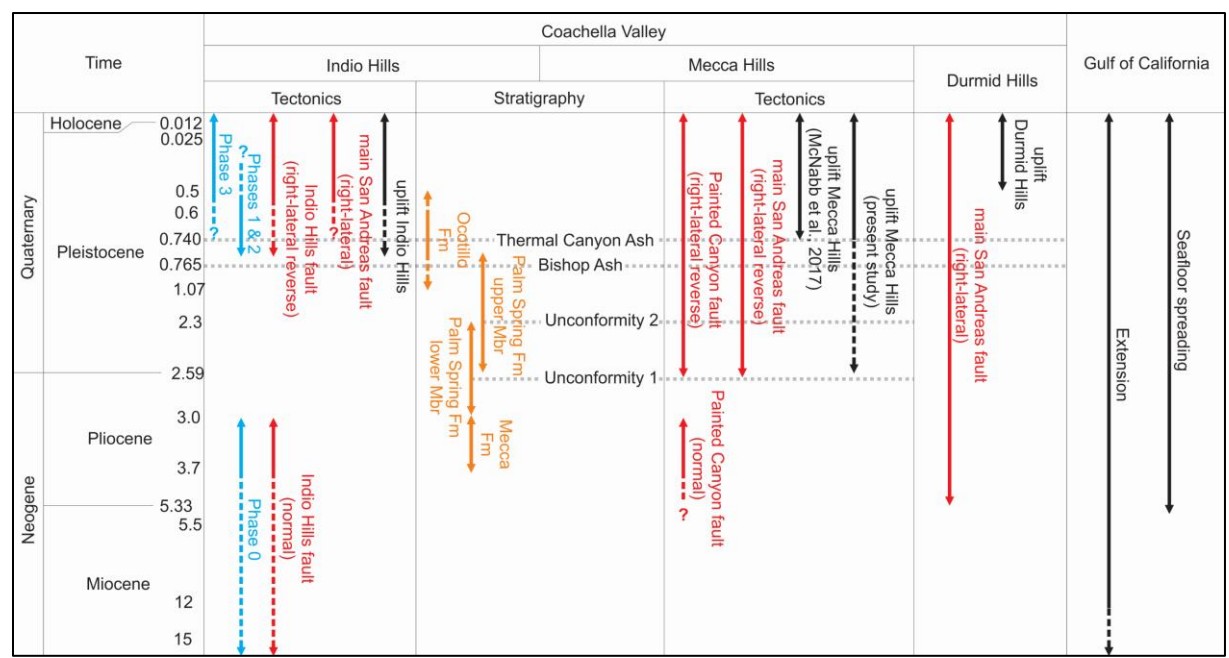


**Table 1: Summary of the timing of the main events in the Coachella Valley and Gulf of**
**California. Note the presumed timing phases (1-3) of fold-faulting and uplift events in**
**the Indio Hills (this work). The stratigraphy is common to the Mecca Hills and Indio**
**Hills, although some features are only observed in one area (e.g., unconformities 1 and 2**
**in the Mecca Hills but not in the Indio Hills).**

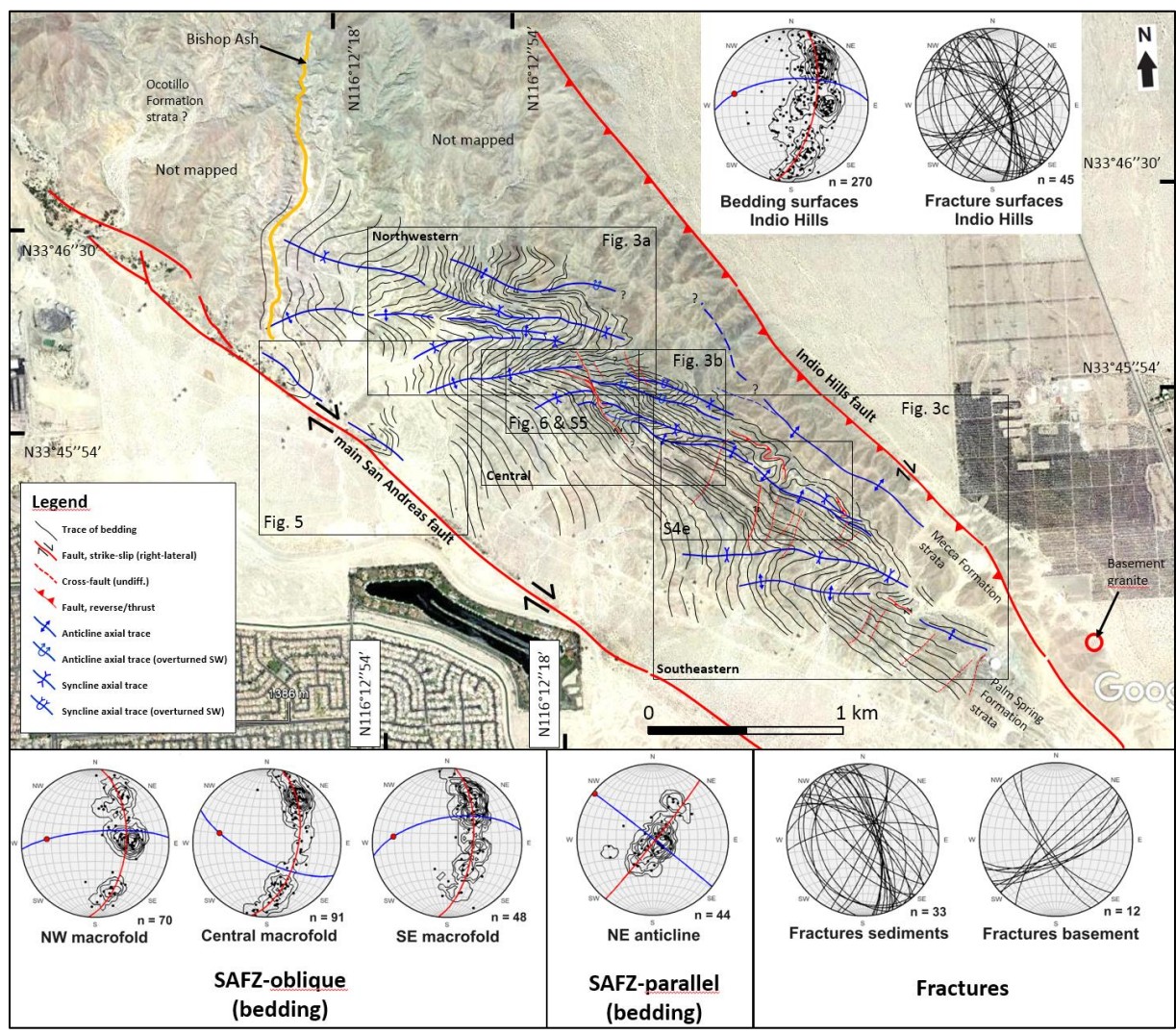

**1246 Figure 2: Interpreted DEM image in the southeastern part of the Indio Hills uplift area.**

**1247 Three main SAFZ-oblique macro-folds (northwestern, central, southeastern) are**

**1248 mapped in between the bounding Indio Hills and main San Andreas faults, whereas one**

**1249 SAFZ-parallel anticline is present close to the Indio Hills fault. More detailed figures are**

**1250 numbered and framed. Structural datasets are plotted in lower hemisphere Schmidt**

**1251 stereonets via the Orient software (Vollmer, 2015). Bedding surfaces are shown as pole to**

**1252 plane with frequency contour lines, with average πS great circle (red great circles), fold**

**1253 axial surface (blue great circles) and fold axis (red dots). Brittle fractures in sedimentary**

**1254 strata and basement rocks are plotted as great circles. Source: Google Earth historical**

**1255 imagery 09-2011. Uninterpreted version of the image available as Supplement S1. ©**

**1256 Google Earth 2011.**


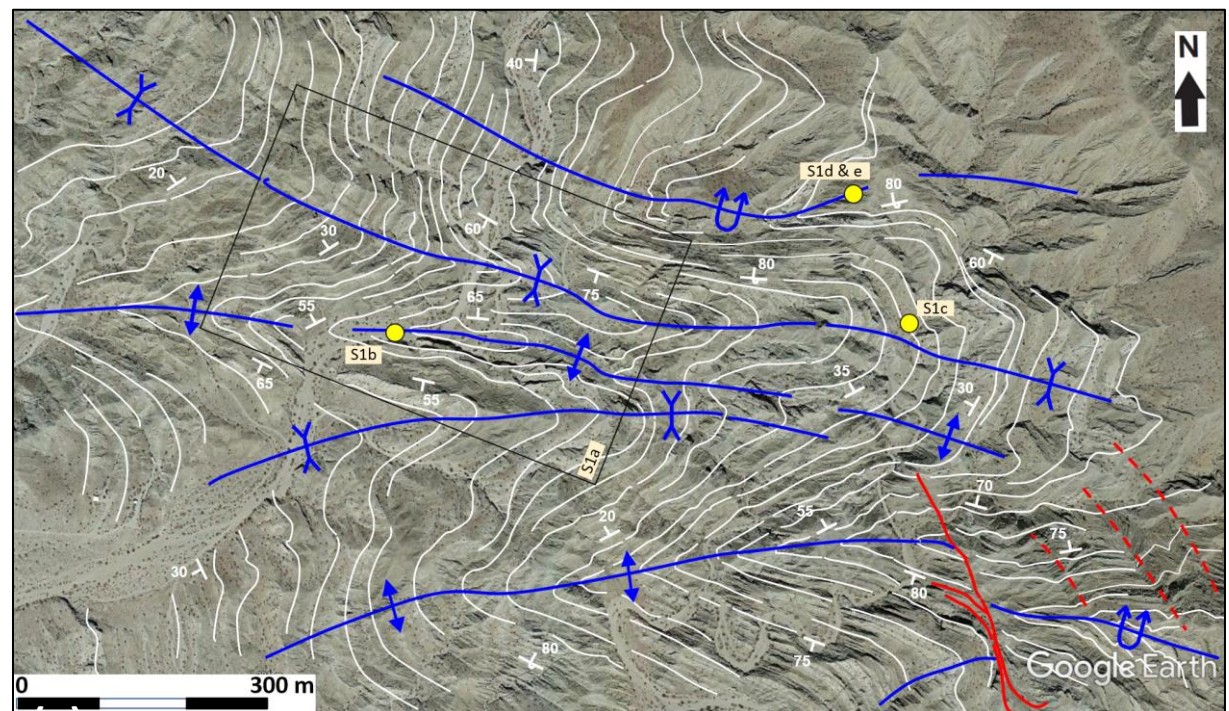


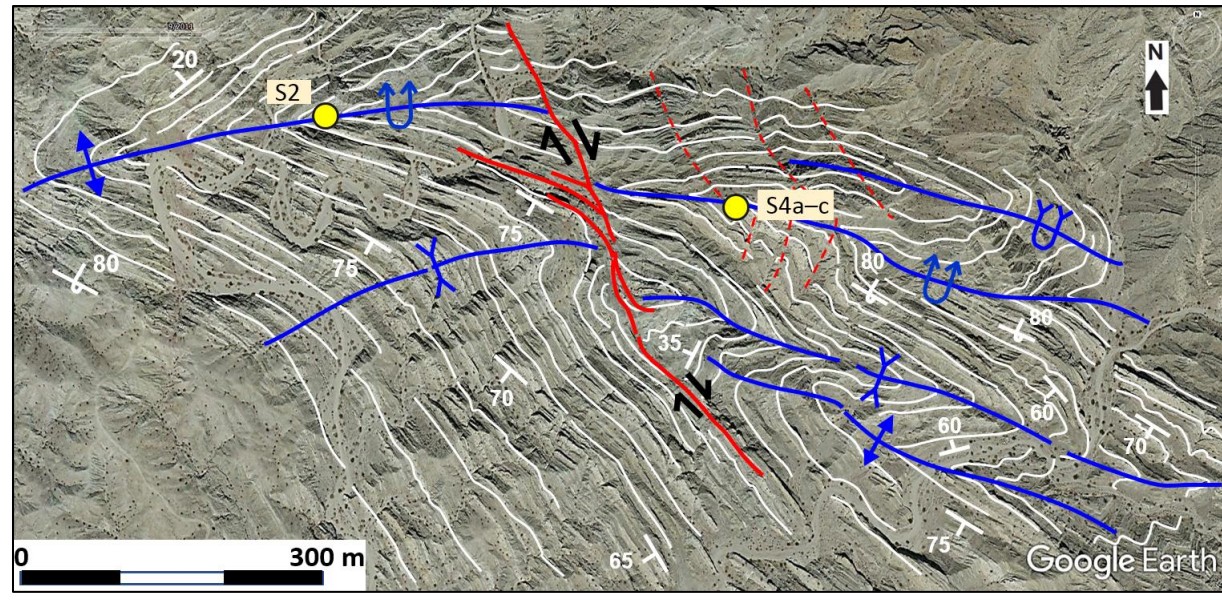


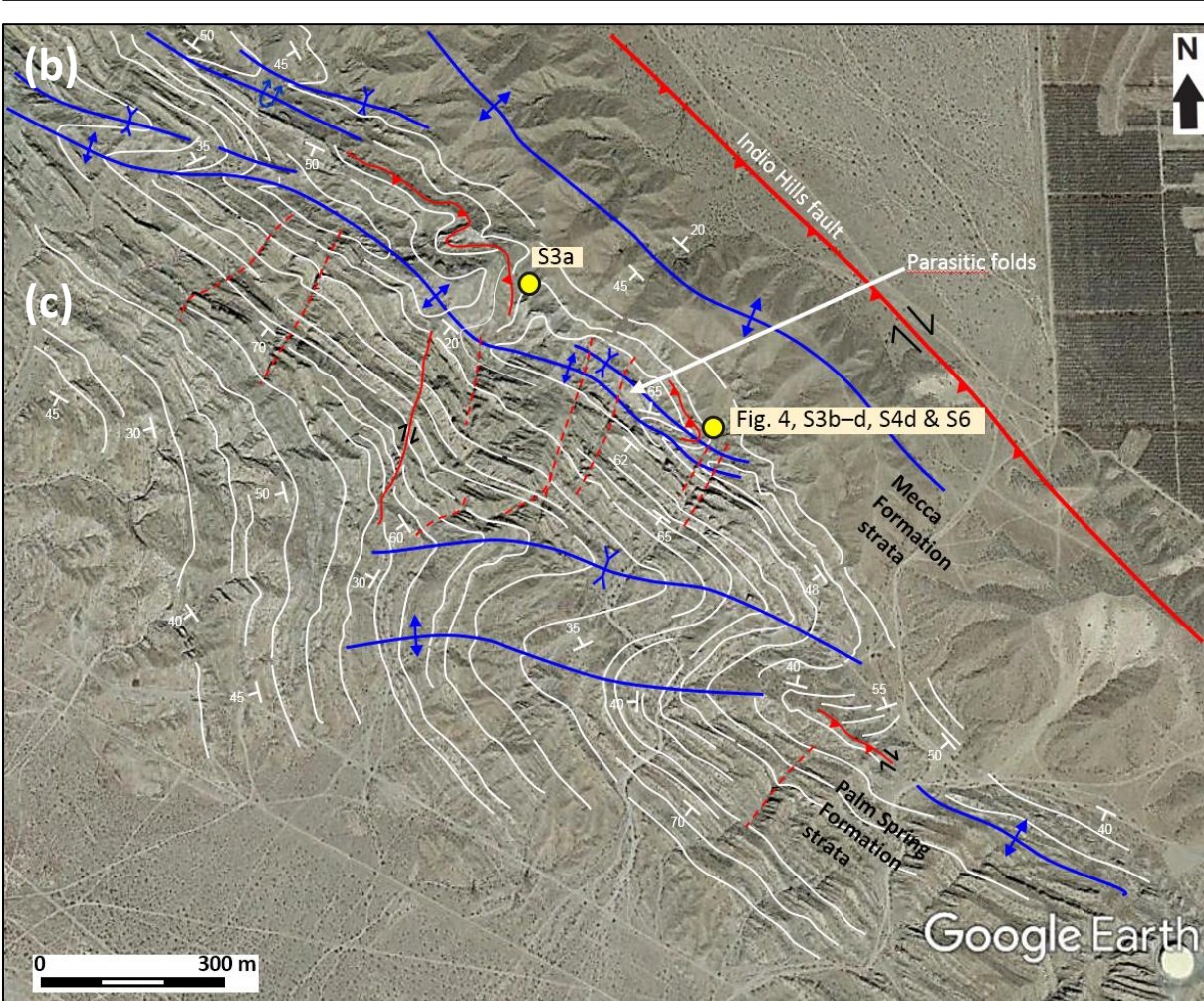


**Figure 3: Detailed structural maps showing the architecture and outline of anticline-**
**syncline pairs, traces of bedding and strike and dip orientation, axial surface traces, and**
**fold-related faults in (a) the northwestern, (b) central, and (c) southeastern macro-folds.**
**Note tighter and consistently asymmetric (Z-shaped) geometries of the macro-folds to**
the east, whereas folds to the west are more open and symmetric. Traces and orientation
of bedding show a back-limb composed of attenuated shear folds merging from the
central macro-fold in the north, whereas the fore-limb is much shorter and more
regularly folded. The yellow dots show the location of field photographs. See fig. 2 for
legend and location. Uninterpreted version of the images available as Supplement S2a–c.
© Google Earth 2011.

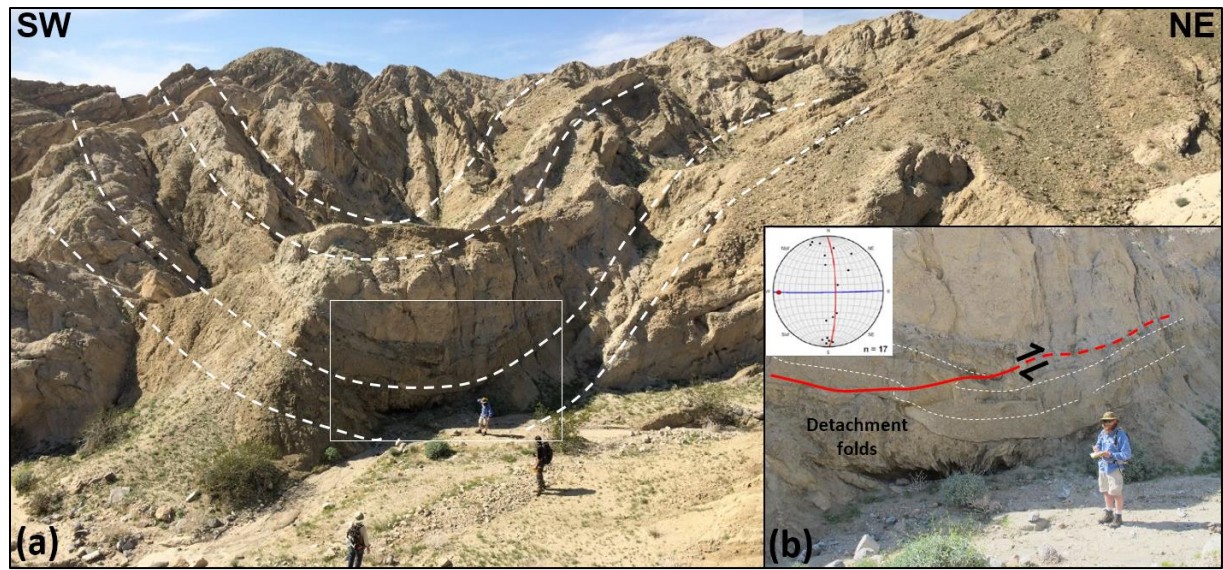

**Figure 4: Meso-scale folds and related faults on the back-limb of the southeastern macro-fold. See location in fig. 3c. (a) Syncline in upper Palm Spring Formation units adjacent to the SAFZ-parallel macro-fold near the Indio Hills fault. (b) Close-up view of the synclinal fold hinge in (a), where a meter thick sandstone bed is slightly offset by a minor, low-angle thrust fault (red line) with NE-directed sense-of-shear. The minor thrust faults die out in the overlying sandstone bed. The mudstone bed below acts as a décollement layer with internal, plastically folded lamination, including disharmonic, intra-detachment folds. Structural orientation data of minor, centimeter-scale fold limbs in the décollement zone are plotted in a lower hemisphere Schmidt stereonet, indicating E–W-trending fold axes and a sub-horizontal axial surface (average great circle in red and fold axis as a red dot).**

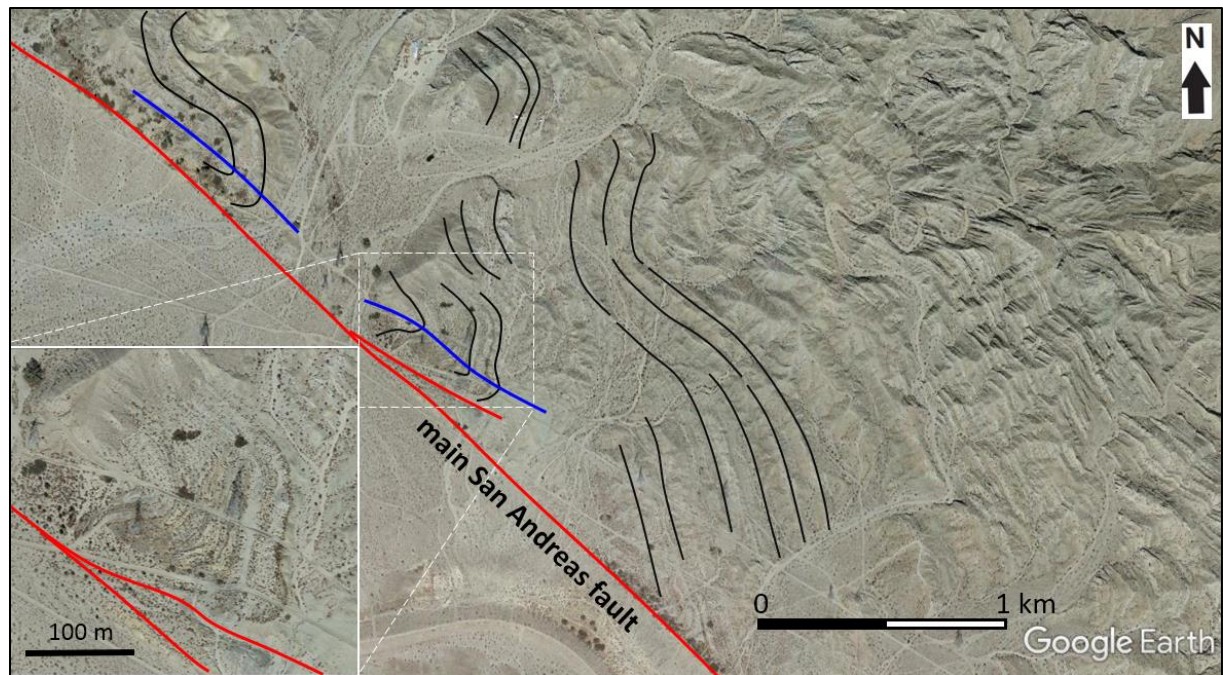

1283

**Figure 5: Interpreted SAFZ-parallel macro-folds (synclines) adjacent to the main San**

**Andreas fault, which affect the southern limb of earlier (*en echelon*) macro-folded and**

**tilted strata of the Palm Spring Formation. Note shear fold geometry in inset map with a**

**thickened hinge zone and thinned limb to the south, steeply plunging axis, and axial**

**trace parallel to the main San Andreas fault. See fig. 2 for location. Uninterpreted**

**version of the image available as Supplement S4. © Google Earth 2011.**

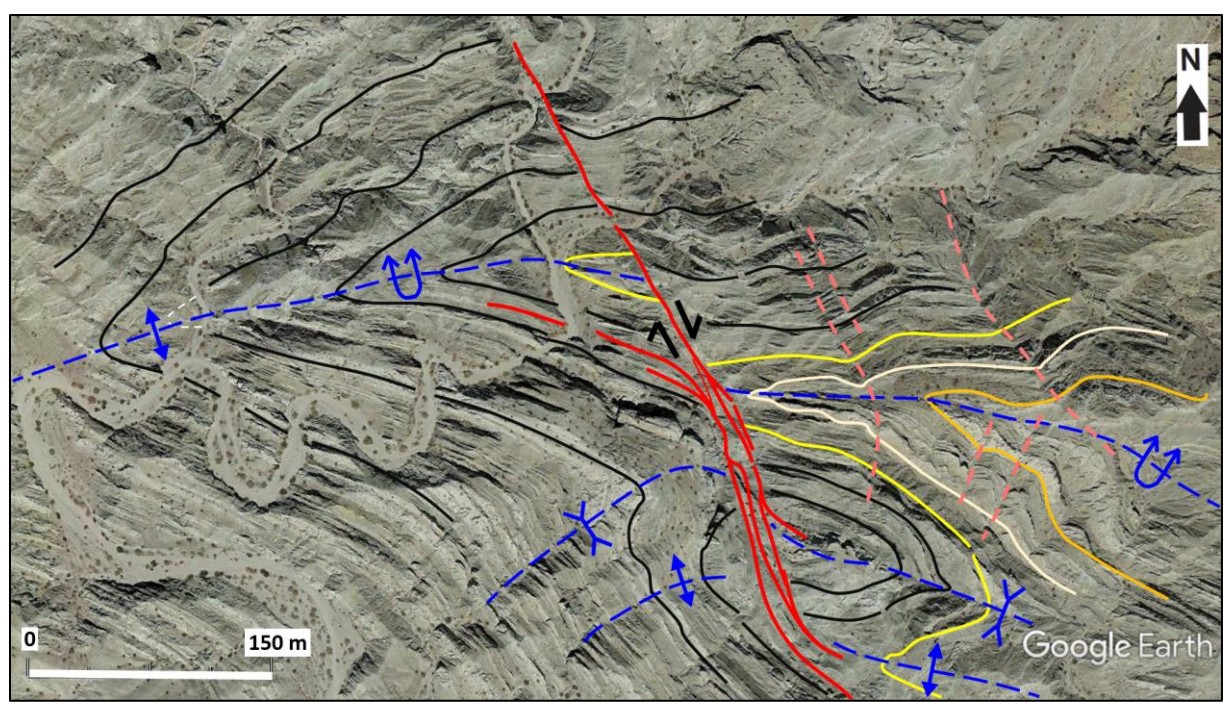

**Figure 6: Interpreted satellite image of the central macro-fold showing right-lateral offset of the entire fold hinge/axial surface (upper left dashed blue line) by a NNW–SSE-trending, NE-dipping strike-slip fault (red lines). Note that the fault merges out from a layer in the southern limb of the macro-fold (black lines) and continues as a right-lateral fault. Offset geological markers include thick sandstone beds (yellow, white, light brown lines) and the fold axial surfaces of a second syncline fold farther south (lower right, dashed blue lines). Note that the syncline axial trace dies out to the southwest, and that kink bands acting as cross faults crop out in the eastern part of image (dashed pink lines). Uninterpreted version of the image available as Supplement S8. © Google Earth 2011.**

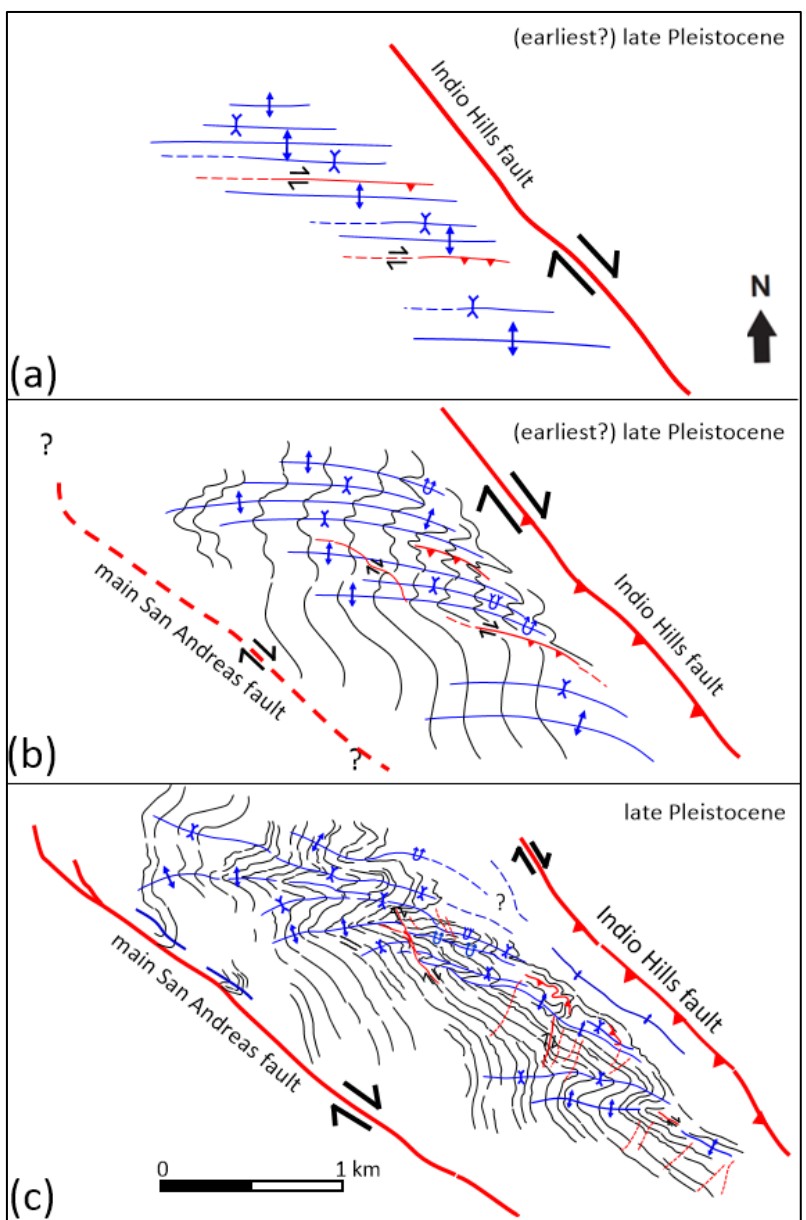

**Figure 7: Model illustrating the progressive uplift/inversion history of the Indio Hills presuming a narrow time interval between formation of all structures in the area, except for the main San Andreas fault and associated folds. (a) Early distributed transpressional strain and formation of three major, *en echelon* oriented macro-folds, several subsidiary parasitic anticline-syncline fold pairs, and bed-parallel strike-slip and reverse (décollement) faults initiating at a high angle (c. 45°) to the Indio Hills fault. (b) Incremental partly partitioned transpression when the Indio Hills fault started to accommodate oblique-reverse movement forcing previous horizontal *en echelon* macro-folds and parasitic folds to tighten, overturn, and rotate into steeper westward plunges. Note also sigmoidal rotation of axial traces on the back-limbs of the macro-folds to low angle (< 20–30°) with the Indio Hills fault. (c) Late-stage advanced strain partitioning**

1313 **with dominant shortening component on the oblique-reverse Indio Hills fault, and right-**

1314 **lateral slip on the main San Andreas Fault. Notice the formation of the anticline parallel**

1315 **to the Indio Hills fault, subsidiary fold-internal strike-slip faults, and conjugate cross**

1316 **faults and kink bands that overprinted the macro-folds. Legend as in fig. 2.**

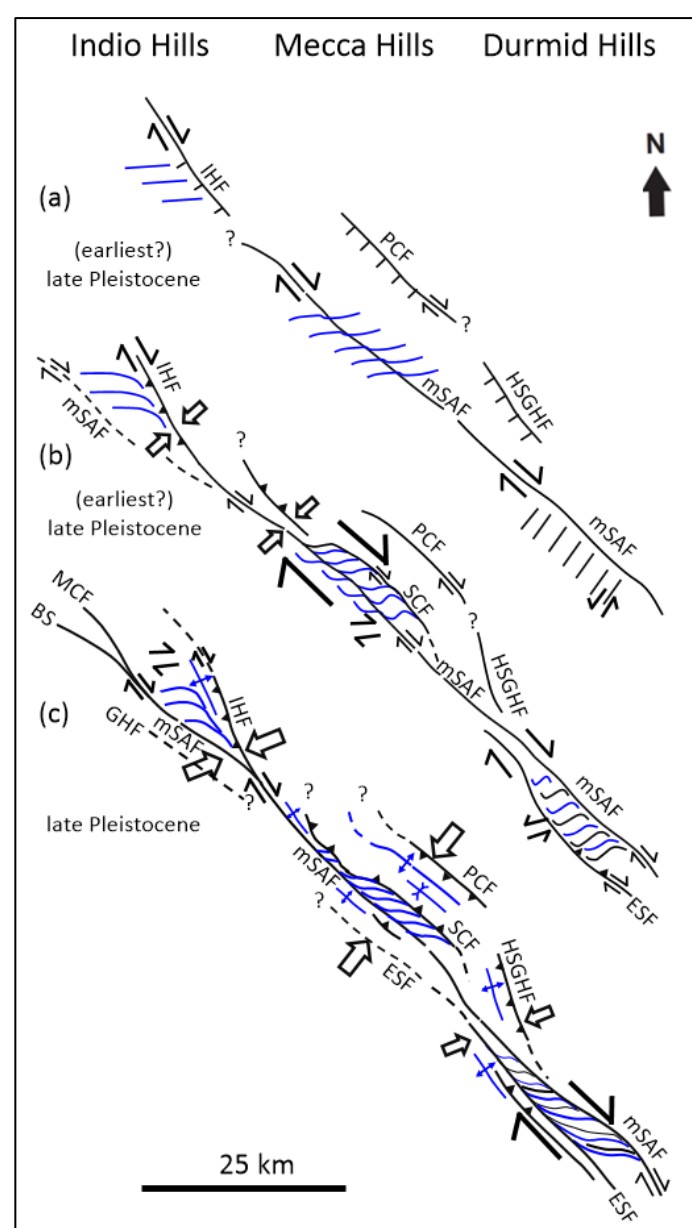

1317

**Figure 8: Kinematic evolution, timing, and along-strike correlation of the Indio Hills, Mecca Hills, and Durmid Hills uplift domains and bounding master faults in the Coachella valley, southern California. We present a progressive kinematic evolution from (a) distributed, through (b) partly partitioned, to (c) advanced partitioned strain events. See text for further explanation. Black lines are faults (full or stippled). Blue lines are fold axial traces. Wide arrows indicate main shortening direction, half-arrows lateral (strike-slip) shearing. Abbreviations: BS: Banning strand; ESF: Eastern Shoreline fault; GHF: Garnet Hill fault; HSGHF: Hidden Springs–Grotto Hills fault; IHF: Indio Hills fault; mSAF: main San Andreas fault; MCF: Mission Creek fault; PCF: Painted Canyon fault; SCF: Skeleton Canyon fault.**