# Peer review of "Tectonic evolution of the Indio Hills segment of the San"

_Solid Earth, 2022_

## Referee Report (RR1)

**General Comments**

This is my second review of the paper by Koehl et al. In general, it is readily apparent that the authors took ample time to take the reviewers' comments into consideration and to make adjustments as seen fit. This updated draft reads much better, and helps the reader to understand the importance of the work. I especially like the detailed Discussion section, notably the regional comparisons. Some changes are pretty sizable: e.g., the names of faults, and hence their structural significance, have changed between versions (what was termed the Banning fault is now recognized as the main SAFZ, it seems). I assume these changes reflect issues brought up by the other reviewer, and trust the new interpretations are sound.

There are a couple instances where the authors talk of a convergent plate boundary along the San Andreas fault. I think this is an error, and they may be referring to local contraction along the transform plate boundary?

The authors took care to address my biggest concerns. This includes making an updated Figure 1 (which looks great, by the way), and to add coordinates to the GoogleEarth image on Figure 2. However, the authors refrained from adding coordinates to all maps/images in Figs. 3, 5, and 6. I don't feel strongly that Figs. 3, 5, and 6 all need coordinates, since they are shown clearly on Fig. 2, but if it were my paper I would certainly add coordinates, north arrows, scales, etc., to all map figures. However, this is not an issue that warrants rejecting the manuscript, and I think it is okay to leave Figs. 3, 5, and 6 without coordinates if the authors choose to.

One major issue I brought up was partly addressed, but still appears. I still worry about how features are mapped on Figure 2, and then subsequently represented on Figs. 3, 5, and 6. One major issue I had with the original figures was that geologic features (faults, fold axes, etc.) appeared short and discontinuous on Figs. 3, 5, and 6, whereas on Figure 2 it was apparent these features were continuous. Some instances of this mistake still persists. I feel strongly that the geology should be represented accurately, and if strike/trend lengths are continuous across and past the bounds of the figure area, then those features strike/trend lengths should go all the way to the ends of the image, not be cut short to fit within the bounds of the figure. In geologic mapping, we do not stop mapping features because they get close to the end of the map, we keep the lines going to hit the edges of the map if that is what the geology is on the ground. I suggest the authors take a careful look at all interpreted images and make sure that geologic features are mapped correctly.

Despite these comments, I feel that the paper has had a lot of great work put into it since my first read through it. Given the careful review by the authors of our reviewer comments (including some pretty ample suggestions from the other reviewer), and that the work is timely, interesting, and a good contribution, I find that the paper is suitable for publication with minor revisions.

Good luck, and thanks for a good read. Cheers.

**Specific Comments –**

Line 14 – "…southern California (USA),…"

Line 28 – "…southeast along strike…" (add "along strike")

Line 29 – I feel that a closing sentence is warranted to pull the reader back into why this work is important. E.g., "Our work allows for better understanding of along-strike complexity and fault zone structure of a major transform plate boundary fault."

Lines 40-42 – This parentheses section may be better suited in the Geologic Setting section?

Lines 47-48 – As noted in my original revisions, I believe shear zone should be decapitalized in Eastern California shear zone. Most recent work do not capitalize it. However, if you choose to use it make sure you are consistent.

Lines 52-54 – This is a great addition to the paper; brings the reader back to why this work is important at a broader scale.

Line 83 – Be consistent. Eastern California shear zone; eastern California shear zone (either way, I think shear zone should be decapitalized).

Line 85 – is axis an appropriate word here? Could it be omitted and just use trending?

Lines 97-101 – This sentence is pretty dense. Could break it up into two.

Lines 143-145 – Should there be a reference at the end of this sentence, or is this your observation?

Lines 154-157 – Could probably merge this single-sentence paragraph with the previous paragraph.

Line 249 – suggest decapitalizing "fault" in all named faults

Lines 249 and 258 – This is a problem from the original manuscript that persists into the present manuscript. Is it "East Shoreline fault" or "Eastern Shoreline fault"? Either way, fault should not be capitalized (as it is in Line 249), and you need to check the entire manuscript so that all names are the same (East or Eastern).

Line 279 – omit dash

Line 331 – suggest changing to "(see subsequent Southeastern macro-fold section)"

Line 387 – omit period at beginning of sentence

Line 759, and throughout manuscript – In some places you dash Landers-Mojave, in other areas of the text you do not (e.g., Landers Mojave Line). I assume dashed is correct. Be consistent throughout manuscript.

Line 849, 864, 872, 873, 879 – Eastern Shoreline fault or East Shoreline fault (I think Eastern, but there are two instances in the manuscript where you say East Shoreline fault at Lines 103 and 249).

**Technical Corrections –**

Line 181 – The abstract says about 0.76 Ma, but here you say before 0.76 Ma.

Line 228 – steep (shallow) ?

Line 231 – Why not just say reverse fault instead of reverse and thrust fault? Do you have constraints on it being a thrust (i.e., <30 degree dipping plane) fault? In my mind, it should be one or the other, if you're going to be explicit about stating fault type, but you cannot go wrong by simply stating reverse fault.

Line 263 – I do not think you can quantify the resolution of stitched and processed Google Earth imagery? As such, it is probably best to omit "high-resolution"

Line 268 – You do not present any restorations in your work. Perhaps "…notably to correlate bed displacements…" is a better wording?

Line 378 (and 263, 402) – Is a Google Earth image a DEM (digital elevation model) image, technically? Should "Google Earth" replace "DEM" here?

Line 393 – What do you mean by large-scale? Large-scale compared to what? Perhaps just say meso-scale, or macro-scale, or outcrop-scale…whatever scale you mean.

Line 531 – shortening strain. Shortening is the strain term, so you do not need to say strain here.

Lines 558-560 – It is unclear as written how the timing on the San Andreas fault-related structure is comparable to structure in Svalbard. Make more clear what you are comparing here.

Lines 625-626 – convergent plate boundary in the late Pleistocene? It is a full-blown transform plate boundary by then.

Line 734 – Again with convergent plate boundary – I don't think you mean plate boundary?

Line 901 – Do you actually mean convergent plate boundary (I don't think so, because it is a transform plate boundary fault system you are examining).

Reply to Comment 78 in review reply: Yes, a fault is a fracture that shows displacement, so you are correct in your reply, technically. However, you cannot expect a reader to know what you mean. Furthermore, technically faults are fractures, yes, but fractures are not faults and the presence/absence of both or one or the other can have different implications. Therefore, you need to be explicit for readers.

**Detailed comments on figures and figure captions –**

*Figure 1*

Figure 1b, in the legend the Landers-Mojave Line does not have a dash, but elsewhere in the manuscript it does. Be consistent, whichever way you choose (I think dashed is probably correct).

Line 1230–1231 – Eastern California shear zone (says "East")

*Figure 2*

For the Bishop Ash, you could also add the age on the figure (e.g., "Bishop Ash X.XX Ma")

Line 1246 – Probably better to say Google Earth image instead of "DEM"

*Figures 3, 5, 6*

I appreciate that coordinates were added to Figure 2. I still think adding coordinates to all maps would be good, but I will leave that up to the authors.

In some areas I can see that feature lines with continuous strike/trend lengths were extended to the edges of maps. However, Fig 3a and 3b is a perfect example where the mapping is not consistently/appropriately portrayed. In 3a, you show the southernmost anticlinal feature continuing for ~900 m west-east from the N-S striking fault, but in Fig 3b – which includes the southernmost portion of 3a – that same anticlinal feature ends before the western edge of the figure. I know these are the same anticline, because in 3a and 3b, you can see the north limb's 20 degree NNW dip, and on the south limb you can see the overturned 80 degree NNE dip. As shown, some of these maps give the impression that the geologic features are shorter than they actually are on the ground. A geologic map depicts reality as best it can be interpreted, whereas these maps do not depict reality, and/or are inconsistent with each other, especially when compared with each other and overall to Figure 2.

I am also concerned after close inspection to see that the location of strikes and dips vary slightly in crossover sections of Figs. 3a–c. It is very apparent these orientation measurements are generally located and not properly georeferenced to an exact point on the ground. For example, the overturned 80 degree NNE dip on the southern limb of the anticline in Figs 3a (southern part of map) and 3b (northern part of map) is in slightly different locations. Sure, the overall orientation of beds is probably represented well by that orientation symbol, but it gives me suspicion how accurately located all other orientations are.

---

## Author Response (AR2)

**Reply to anonymous reviewer**

Dear Sir, Madam,

thank you very much for your input on the manuscript, it is highly appreciated. Here is our reply to your comments. We hope the changes we implemented improve the shortcomings of the manuscript highlighted by your comments and suggestions. Please do not hesitate to contact us shall this not be the case for some comments.

**1. Comments from anonymous reviewer**

Comment 1: General Comments

This is my second review of the paper by Koehl et al. In general, it is readily apparent that the authors took ample time to take the reviewers' comments into consideration and to make adjustments as seen fit. This updated draft reads much better, and helps the reader to understand the importance of the work. I especially like the detailed Discussion section, notably the regional comparisons. Some changes are pretty sizable: e.g., the names of faults, and hence their structural significance, have changed between versions (what was termed the Banning fault is now recognized as the main SAFZ, it seems). I assume these changes reflect issues brought up by the other reviewer, and trust the new interpretations are sound.

Comment 2: There are a couple instances where the authors talk of a convergent plate boundary along the San Andreas fault. I think this is an error, and they may be referring to local contraction along the transform plate boundary?

Comment 3: The authors took care to address my biggest concerns. This includes making an updated Figure 1 (which looks great, by the way), and to add coordinates to the GoogleEarth image on Figure 2. However, the authors refrained from adding coordinates to all maps/images in Figs. 3, 5, and 6.

Comment 4: I don't feel strongly that Figs. 3, 5, and 6 all need coordinates, since they are shown clearly on Fig. 2, but if it were my paper I would certainly add coordinates, north arrows, scales, etc., to all map figures. However, this is not an issue that warrants rejecting the manuscript, and I think it is okay to leave Figs. 3, 5, and 6 without coordinates if the authors choose to.

Comment 5: One major issue I brought up was partly addressed, but still appears. I still worry about how features are mapped on Figure 2, and then subsequently represented on Figs. 3, 5, and 6. One major issue I had with the original figures was that geologic features (faults, fold axes, etc.) appeared short and discontinuous on Figs. 3, 5, and 6, whereas on Figure 2 it was apparent these features were continuous. Some instances of this mistake still persists. I feel strongly that the geology should be represented accurately, and if strike/trend lengths are continuous across and past the bounds of the figure area, then those features strike/trend lengths should go all the way to the ends of the image, not be cut short to fit within the bounds of the figure. In geologic mapping, we do not stop mapping features because they get close to the end of the map, we keep the lines going to hit the edges of the map if that is what the geology is on the ground. I suggest the authors take a careful look at all interpreted images and make sure that geologic features are mapped correctly.

Comment 6: Specific Comments –

Line 14 – "…southern California (USA),…"

Comment 7: Line 28 – "…southeast along strike…" (add "along strike")

Comment 8: Line 29 – I feel that a closing sentence is warranted to pull the reader back into why this work is important. E.g., "Our work allows for better understanding of along-strike complexity and fault zone structure of a major transform plate boundary fault."

Comment 9: Lines 40-42 – This parentheses section may be better suited in the Geologic Setting section?

Comment 10: Lines 47-48 – As noted in my original revisions, I believe shear zone should be decapitalized in Eastern California shear zone. Most recent work do not capitalize it. However, if you choose to use it make sure you are consistent.

Comment 11: Lines 52-54 – This is a great addition to the paper; brings the reader back to why this work is important at a broader scale.

Comment 12: Line 83 – Be consistent. Eastern California shear zone; eastern California shear zone (either way, I think shear zone should be decapitalized).

Comment 13: Line 85 – is axis an appropriate word here? Could it be omitted and just use trending?

Comment 14: Lines 97-101 – This sentence is pretty dense. Could break it up into two.

Comment 15: Lines 143-145 – Should there be a reference at the end of this sentence, or is this your observation?

Comment 16: Lines 154-157 – Could probably merge this single-sentence paragraph with the previous paragraph.

Comment 17: Line 249 – suggest decapitalizing "fault" in all named faults

Comment 18: Lines 249 and 258 – This is a problem from the original manuscript that persists into the present manuscript. Is it "East Shoreline fault" or "Eastern Shoreline fault"? Either way, fault should not be capitalized (as it is in Line 249), and you need to check the entire manuscript so that all names are the same (East or Eastern).

Comment 19: Line 279 – omit dash

Comment 20: Line 331 – suggest changing to "(see subsequent Southeastern macro-fold section)"

Comment 21: Line 387 – omit period at beginning of sentence

Comment 22: Line 759, and throughout manuscript – In some places you dash Landers-Mojave, in other areas of the text you do not (e.g., Landers Mojave Line). I assume dashed is correct. Be consistent throughout manuscript.

Comment 23: Line 849, 864, 872, 873, 879 – Eastern Shoreline fault or East Shoreline fault (I think Eastern, but there are two instances in the manuscript where you say East Shoreline fault at Lines 103 and 249).

Comment 24: Technical Corrections –

Line 181 – The abstract says about 0.76 Ma, but here you say before 0.76 Ma.

Comment 25: Line 228 – steep (shallow) ?

Comment 26: Line 231 – Why not just say reverse fault instead of reverse and thrust fault? Do you have constraints on it being a thrust (i.e., <30 degree dipping plane) fault? In my mind, it should be one or the other, if you're going to be explicit about stating fault type, but you cannot go wrong by simply stating reverse fault.

Comment 27: Line 263 – I do not think you can quantify the resolution of stitched and processed Google Earth imagery? As such, it is probably best to omit "high-resolution"

Comment 28: Line 268 – You do not present any restorations in your work. Perhaps "…notably to correlate bed displacements…" is a better wording?

Comment 29: Line 378 (and 263, 402) – Is a Google Earth image a DEM (digital elevation model) image, technically? Should "Google Earth" replace "DEM" here?

Comment 30: Line 393 – What do you mean by large-scale? Large-scale compared to what? Perhaps just say meso-scale, or macro-scale, or outcrop-scale…whatever scale you mean.

Comment 31: Line 531 – shortening strain. Shortening is the strain term, so you do not need to say strain here.

Comment 32: Lines 558-560 – It is unclear as written how the timing on the San Andreas fault-related structure is comparable to structure in Svalbard. Make more clear what you are comparing here.

Comment 33: Lines 625-626 – convergent plate boundary in the late Pleistocene? It is a full-blown transform plate boundary by then.

Comment 34: Line 734 – Again with convergent plate boundary – I don't think you mean plate boundary?

Comment 35: Line 901 – Do you actually mean convergent plate boundary (I don't think so, because it is a transform plate boundary fault system you are examining).

Comment 36: Reply to Comment 78 in review reply: Yes, a fault is a fracture that shows displacement, so you are correct in your reply, technically. However, you cannot expect a reader to know what you mean. Furthermore, technically faults are fractures, yes, but fractures are not faults and the presence/absence of both or one or the other can have different implications. Therefore, you need to be explicit for readers.

Comment 37: Detailed comments on figures and figure captions –

Figure 1

Figure 1b, in the legend the Landers-Mojave Line does not have a dash, but elsewhere in the manuscript it does. Be consistent, whichever way you choose (I think dashed is probably correct).

Comment 38: Line 1230–1231 – Eastern California shear zone (says "East")

Comment 39: Figure 2

For the Bishop Ash, you could also add the age on the figure (e.g., "Bishop Ash X.XX Ma")

Comment 40: Line 1246 – Probably better to say Google Earth image instead of "DEM"

Comment 41: Figures 3, 5, 6

I appreciate that coordinates were added to Figure 2. I still think adding coordinates to all maps would be good, but I will leave that up to the authors.

Comment 42: In some areas I can see that feature lines with continuous strike/trend lengths were extended to the edges of maps. However, Fig 3a and 3b is a perfect example where the mapping is not consistently/appropriately portrayed. In 3a, you show the southernmost anticlinal feature continuing for ~900 m west-east from the N-S striking fault, but in Fig 3b – which includes the

southernmost portion of 3a – that same anticlinal feature ends before the western edge of the figure. I know these are the same anticline, because in 3a and 3b, you can see the north limb's 20 degree NNW dip, and on the south limb you can see the overturned 80 degree NNE dip. As shown, some of these maps give the impression that the geologic features are shorter than they actually are on the ground. A geologic map depicts reality as best it can be interpreted, whereas these maps do not depict reality, and/or are inconsistent with each other, especially when compared with each other and overall to Figure 2.

Comment 43: I am also concerned after close inspection to see that the location of strikes and dips vary slightly in crossover sections of Figs. 3a–c. It is very apparent these orientation measurements are generally located and not properly georeferenced to an exact point on the ground. For example, the overturned 80 degree NNE dip on the southern limb of the anticline in Figs 3a (southern part of map) and 3b (northern part of map) is in slightly different locations. Sure, the overall orientation of beds is probably represented well by that orientation symbol, but it gives me suspicion how accurately located all other orientations are.

**2. Author's reply**

Comment 1: agreed.

Comment 2: agreed.

Comment 3: agreed.

Comment 4: coordinates are not absolutely necessary in these figures.

Comment 5: agreed.

Comment 6: agreed.

Comment 7: agreed.

Comment 8: agreed.

Comment 9: agreed.

Comment 10: agreed.

Comment 11: agreed.

Comment 12: agreed.

Comment 13: agreed.

Comment 14: agreed.

Comment 15: this is our observation.

Comment 16: agreed.

Comment 17: agreed.

Comment 18: agreed.

Comment 19: agreed.

Comment 20: agreed.

Comment 21: agreed.

Comment 22: agreed.

Comment 23: agreed. See also response to comment 18.

Comment 24: the abstract refers to the Indio Hills area, whereas the sentence line 185 refers to the Mecca Hills.

Comment 25: agreed, the sentence is not clear enough.

Comment 26: agreed.

Comment 27: agreed.

Comment 28: agreed.

Comment 29: agreed.

Comment 30: agreed.

Comment 31: agreed.

Comment 32: agreed. This phrase is unnecessary.

Comment 33: agreed. See also response to comment 2.

Comment 34: agreed. See also response to comment 2.

Comment 35: agreed. See also response to comment 2.

Comment 36: agreed.

Comment 37: agreed. See also response to comment 22.

Comment 38: agreed. See also response to comment 10.

Comment 39: we do not feel that it is necessary to overcrowd the figure with extra information that can be found in several places in the text.

Comment 40: agreed. See also response to comment 29.

Comment 41: see response to comment 4.

Comment 42: agreed. See response to comment 5.

Comment 43: agreed, the strike and dip symbols are not georeferenced. However, the location and geometries fold and fault structures are so well expressed on Google Earth images that

georeferencing is not necessary to plot structural measurements. In addition, slight mismatches of the location of these measurements on macro-scale folds do not impact the structure geometries at all on the presented figures.

**3. Changes implemented**

Comment 1: none commanded by the reviewer's comment.

Comment 2: replaced "convergent" by "transform" lines 625–626, 734, and 901.

Comment 3: none commanded by the reviewer's comment.

Comment 4: none.

Comment 5: adjusted extent of structures according to the reviewer's suggestion in figures 2, 3, 5, and 6.

Comment 6: added " (USA)" line 14.

Comment 7: added " along strike" line 28.

Comment 8: added "The present work contributes to better understand the structure and tectonic history of a major fault system along a transform plate boundary." lines 29–31.

Comment 9: moved sentence in parenthesis from lines 42–44 to lines 83–85.

Comment 10: adjust all occurrences to "Eastern California shear zone" lines 49–50, 87, 202, and 1234–1235.

Comment 11: none commanded by the reviewer's comment.

Comment 12: see response to comment 10.

Comment 13: replaced "along a NNW–SSE-trending axis" by "in a NNW–SSE-trend" lines 89–90.

Comment 14: split the sentence into two line 102.

Comment 15: none.

Comment 16: merged single-sentenced paragraph lines 158–161 to previous paragraph.

Comment 17: decapitalized "Fault" lines 253, 414, 687, 758, and 1318 and "Fault Zone" line 35.

Comment 18: adjust "Eastern Shoreline fault" to "East Shoreline fault" lines 262, 853, 868, 876, 877, 883, 1235, and 1328.

Comment 19: remove the strikethrough font line 283.

Comment 20: changed phrase between brackets into "see Southeastern macro-fold section" lines 335–336.

Comment 21: deleted period and space at the beginning of the sentence line 391.

Comment 22: added en-dash lines 87, 88, 686, and 1237.

Comment 23: Janecke et al. (2018) use "East Shoreline fault", so adjusted the name accordingly.

Comment 24: none.

Comment 25: added "portion of the" and deleted parenthesis lines 232–233.

Comment 26: deleted "and thrust" line 235.

Comment 27: deleted "high-resolution" line 268.

Comment 28: replaced "notably for restoring bed offsets" by "notably to correlate bed displacement" lines 273–274.

Comment 29: deleted "DEM" line 268, replaced "DEM" by "Google Earth images" lines 384, and 408, and replaced "DEM" by "Satellite and aerial" lines 935–936 and 1252.

Comment 30: replaced "Large" by "Macro" line 399.

Comment 31: deleted "strain" line 537.

Comment 32: deleted ", i.e., comparable to other settings (e.g., western Svalbard; Bergh et al., 1997; Braathen et al, 1999)" lines 565–566 and Bergh et al. (1997) and Braathen et al. (1999) from the reference list.

Comment 33: see response to comment 2.

Comment 34: see response to comment 2.

Comment 35: see response to comment 2.

Comment 36: added "Note that faults are also included as fractures in the stereonets." lines 1257–1258.

Comment 37: see also response to comment 22.

Comment 38: see also response to comment 10.

Comment 39: none.

Comment 40: see also response to comment 29.

Comment 41: see response to comment 4.

Comment 42: see response to comment 5.

Comment 43: none.

---

## Author Response (AR3)

**Reply to Anonymous referee**

**Dear Sir, Madam,**

thank you very much for your input on the manuscript, it is highly appreciated. Here is our reply to your comments. We hope the changes we implemented improve the shortcomings of the manuscript highlighted by your comments and suggestions. Please do not hesitate to contact us shall this not be the case for some comments.

**1. Comments from Anonymous referee**

Comment 1: I suggest that this paper be accepted/reconsidered after major revisions. In general, I enjoyed this paper: it is a good, well-written paper with a structurally interesting dataset from a major transform plate boundary fault zone. The dataset is collected from a transpressional uplift within the San Andreas fault zone, then compared to other similar features along strike. As such, the paper stands to be a good contribution for those trying to understand the internal structure, along-strike complexity, and tectonic evolution of transform plate boundary fault zones, and more specifically the along-strike complexity of the southeastern terminus of the San Andreas transform plate boundary fault.

Comment 2: The overwhelming majority of my comments are minor, albeit numerous. However, there are a few major points concerning the figures that need to be addressed should the manuscript be accepted for publication. These few major points concerning the figures may take some time to complete, and are my only reason for listing the revision as major, not minor. These include: Figure 1 needs to be redone to include a regional map with all the features discussed in the text plotted on that map and, in general, showing the study area in the regional context (southern California, southwestern USA). An updated figure could take the form of a two-panel figure, where Fig. 1a is the regional map showing major features discussed in text, and Fig. 1b is the close-up map that is currently presented as the sole Fig. 1. At present, the reader has no regional context for the features discussed in-text, and some features and faults are not shown on any map, making their comparison and importance to the study area difficult and unclear. Comment 3: All maps in the figures (Figs. 2, 3, 5, and 6) should have coordinates of some sort, whether as points or a grid. Additionally, I suggest that un-interpreted images of all of the map

areas should be added to the supplemental material (an un-interpreted Fig. 6 is already in the SM).

Comment 4: Folds and faults mapped on Fig. 2 appear continuous across some parts of the Northwestern, Central, and Southeastern domains. However, in Figs. 3 and 6, the folds and faults appear short and discontinuous. These figures should be updated to reflect the full extent of the structure(s) within the figure's frame to be consistent with their geology on the ground and as shown on Fig. 2. Should these changes be addressed, I think the paper will make a good contribution. Good luck, and I hope to see this in print in the near future.

Comment 5: **Specific Comments -** Title: Should a broader geographic description be applied to the title, given this is European journal but the study area is in the USA? Perhaps "Tectonic evolution of the Indio Hills segment of the San Andreas fault in southern California, southwestern USA"

Comment 6: Line 46-47 - What about this continuation in to the ECSZ? The sentence needs more description about the significance of the Indio Hills fault with the ECSZ.

Comment 7: Line 68 - I am curious about the use of the term "culmination" - I am only familiar with this term in fold-thrust systems. As defined at

https://link.springer.com/content/pdf/bbm%3A978-94-011-3066-0%2F1.pdf : "Culmination: An anticline or dome with four way closure generated by movement of the thrust sheet over underlying ramps." I understand you have transpressional folding/thrusting going on in your study area, so the term could be used, but does the Indio Hills exhibit folding over underlying thrust ramps? Or are you simply referring to a variety of distinct tectonic elements all observed together in one place? If the latter, I think a different term is warranted. If you choose to keep the term culmination, I think you need to explicitly define it, either here or in your Tectonic Culminations section below. Perhaps it is best to simply call it the Indio Hills uplift here on Line 68, as you do in the Fig. 1 caption, and leave the use of culmination (if you keep it) for the section below.

Comment 8: Line 69 - You state the Indio Hills are a transpressional uplift, but consider it analogous to a rift feature (which would suggest transtension)? See next comment. Comment 9: Line 68-70 - I think what you mean is that the Indio Hills and Mecca Hills are analogous in that they are both inverted basins? If that is correct, be more explicit here. For example, you could say: "The Indio Hills uplift is an inverted Miocene–Pliocene sedimentary basin lying upon Mesozoic granitic basement rocks. Further to the southeast, the Mecca Hills are also shown to be an inverted Miocene–Pliocene sedimentary basin (Keller et al., 1982; Damte, 1997; McNabb et al., 2017; Bergh et al., 2019)."

Comment 10: Lines 84-86 – You state "We consider" but then list references. Are you interpreting that these units are The Mecca Formation, or did the cited authors interpret these units to be the Mecca Formation in the Indio Hills. The former is slightly problematic, as it is an interpretation before the data section (but understandably a necessary one to make for your study).

Comment 11: Line 103-104 - How would sediment accumulation rates define the age of a formation? More than likely, the dates of those stratigraphic members were used to calculate the sediment accumulation rates. Did the lower and upper members of the Palm Springs Formation show increased rates of sediment accumulation during these intervals? If so, specify that. Comment 12: Lines 114-117 - See above comment on use of the term culmination. You only use the term four times in the paper, here three times and once in the former section. I suspect the term should be changed, given the formal definition I pasted in the Line 68 comment above, but if you choose to keep the term then define what you mean by "culmination" either here or at Line 68.

Comment 13: Line 119-124 - Your broad-scope description of tectonic elements here shows the necessity of adding a regional map to your Figure 1. At present, the reader has no context for the Eastern California shear zone (which is a much broader region than you show it in Fig. 1), the San Bernardino and San Jacinto faults, and San Gorgonio Pass. These features need added to a map with the location of the study area clearly shown so the reader can see their relationship and importance to the work presented here.

Comment 14: Line 155-156 – Delete the word "off-fault" – damage zones typically encompass principal slip surfaces but are technically part of the fault zone, too, so it seems kind of like a misnomer to say off-fault

Comment 15: Lines 160-163 – Is there a reference for this statement?

Comment 16: Line 219 - Are you saying that the open upright fold geometry is the result of (via) the kink/chevron styles? If so, no change is really necessary, but perhaps it could be described more clearly? If not, and instead you are describing a sequence of changing fold patterns, then I'd replace "via" with "to"

Comment 17: Line 241 – Here again with "via". Do you mean the something is the result of the kink/chevron geometry, or are you saying it is spatially changing from symmetric style, to then changing to a kink/chevron style, to then changing to isoclinal? If so, I'd suggest replacing "via" with "to"

Comment 18: Line 273 – What do you mean by monocline-like? It seems the fold would either be a monocline or an anticline, not a mix of the two. According to your Fig. 3C the fold closest to the Indio Hills fault very much looks to be an asymmetric anticline, with 20NE dip on the northeast limb and 45SW dip on the southwest limb, in which case I would delete "monocline-like" from the sentence.

Comment 19: Line 299-305 – What kind of folds are these? Anticline? Syncline? Both? Note that hinge lines are not mapped on Figure 5 like they are on Figure 2.

Comment 20: Lines 307-319; Lines 310-311 – If you are discussing faults and fractures in the basement, then that is not a fold-related fault (unless the basement is folded). Perhaps the section should be renamed "Major and minor faults, fractures, and fold-related faults"

Comment 21: Line 388 - Cite Figure 2 stereonet at the end of the sentence. Also, these fracture sets look to be ~90° to one another; I'd expect conjugates to be ~60° (40-70°) to one another. It might be best to delete the "possibly representing conjugate sets" from the sentence, as I don't think these are conjugates. This shouldn't pose a problem, as you don't discuss these features any further in the manuscript.

Comment 22: Line 416 – "…indicates a younger phase of deformation." Saying younger slip event makes it sound like only one slip event caused the present-day observed deformation pattern.

Comment 23: Line 419 – You could delete "strain" after shortening; since shortening is a strain term it is a little redundant.

Comment 24: Lines 433-440 – I think it would help the reader here to remind them stratigraphically which unit overlies/underlies which unit, or which unit is older and which unit is younger. E.g., "the Mecca Formation and overlying Palm Springs Formation." or something to that effect.

Comment 25: Line 435 – would the fault be below the contact between the PS and MH formations, or would the fault be at/near the contact of the PS and MH formations?

Comment 26: Line 456-457 – dip-slip fault-parallel fold: wouldn't this just be a fault-propagation fold? I suppose it could also be a fault-bend fold by that description, but I get the impression it is fault-propagated.

Comment 27: Line 509 - stress, or strain?

Comment 28: Line 516 – I'd be more satisfied if these features were rigorously measured and restored back to a discrete bedding orientation through stereonet analysis. As presented in Supplement S6, you are "restoring" apparent dips at the outcrop face to an approximate horizontal based on the apparent dip of bedding in the picture/outcrop face. I absolutely agree with what you are saying and interpreting, but wonder if you should not refer to this as a restoration, per se, but rather that these features "appear to define a low-angle fold and thrust system (Supplement S6)."

Comment 29: Lines 558-559 – Concerning use of axial surfaces (i.e., axial planes), wouldn't a surface/plane be E–W-striking (not trending)? Perhaps it is better to just say E–W-trending folds. Comment 30: Line 667 – Transpressional plate regime. Are you suggesting that the plate is entirely under transpression in this area (in which case, which plate – or both?), or are you saying San Andreas fault zone transpression? Depending which you mean could be important, as just to the east ~100 km some of us are arguing for late Miocene–Pliocene (and possibly ongoing) transtension in the lower Colorado River corridor. This is all the more complicated in that the ECSZ does seem to be overwhelmingly transpressive. Looking through your cited reference (Bergh et al., 2019), I think you mean San Andreas fault zone transpression – if so, please modify the "transpressional plate regime" part of the sentence to instead reflect SAFZ transpression. If you indeed mean transpression across the plate(s), I think you need to be more specific of the extent of this transpressional plate regime, and possibly even reconcile your claims by looking into recent literature for Pacific-North America plate boundary transtension inboard of the SAFZ just next door to the east (e.g., Singleton et al., 2019; Thacker et al., 2020; Dorsey et al., 2021), albeit ca. 3 to 1 Ma earlier than you propose the Indio Hills to have formed.

Comment 31: Lines 713-722; Point 4 in Conclusions (Lines 738-740) – I am a bit confused about how the Indio Hills and Durmid Hills are shown as initially different in Fig. 8a, but in this paragraph you suggest that the two areas might be similar in that the Indio Hills might be an early phase of a ladder structure like the Durmid Hills. In Fig. 8a you clearly show an inherent difference between the two areas: Indio Hills has E-W folds, Durmid Hills has NE-SW left-lateral

faults - am I to assume the E-W folds had already formed, or did E-W folds not form, which would again suggest an inherent difference between the two areas? I am also confused how these two areas are potentially similar when the proposed timing of fault activation for the oblique dextral-reverse fault in both locations is opposite: The Indio Hills fault (what became the oblique dextral-reverse fault) formed before the Banning fault, while the Eastern Shoreline fault (what became the oblique dextral-reverse fault) formed after the main San Andreas fault, according to your figure.

Comment 32: **Technical Corrections** – Line 33 – A geographic description is required. For example: "...San Andreas fault zone (SAFZ: Fig. 1; California, southwestern USA), …" Comment 33: Line 34 – add "the" ("...deformation compared to the Mecca Hills…") Comment 34: Lines 41-42 – Note that "Eastern California shear zone" is commonly written with "shear zone" not capitalized. Change here, and throughout the manuscript to be "Eastern California shear zone"

Comment 35: Line 60 – delete "transform" and remove "s" from movements so it reads "…North American plates and movement along the SAFZ…" Also, should it be North America plate or North American plate?

Comment 36: Line 65 (end of paragraph) – I think a final sentence is needed here that brings it all back into perspective. Perhaps something akin to: "This recent work provides the opportunity to explore the understudied Indio Hills segment in order to compare its structural development with other along-strike uplifted features on a major transform plate boundary fault zone."

Comment 37: Line 128 – Gorgonio is misspelled

Comment 38: Line 127-130 – Suggest breaking this one sentence into two different sentences.

Comment 39: Line 134 – Eastern California shear zone (decapitalize shear zone) – change here and throughout the manuscript and figure captions.

Comment 40: Line 137 – Delete "attitude and" so the sentence reads "Farther southeast, however, the geometry of the..."

Comment 41: Line 138 - Add an "s" to remains

Comment 42: Line 140-142 - Suggest separating these into two sentences: "The transpressional character of the Indio Hills uplift was suggested by Parrish (1983) and Sylvester and Smith (1987). Recent work, however, has not been conducted, and detailed structural analyses have not been published from this segment of the SAFZ."

Comment 43: Line 143 – perhaps change focusing to "that focused"

Comment 44: Line 149 – Be explicit here with who you are referring to. I think you mean Keller et al. (1982). If so, I suggest replacing "Their" with the reference.

Comment 45: Line 173 - I think you mean main San Andreas fault strand, based on the abbreviation, but that is not totally clear as written. Suggest saying "main San Andreas fault (mSAF) strand..."

Comment 46: Line 178 and throughout the manuscript and figures – Make sure to decapitalize "fault" after all formal names. E.g., East Shoreline fault, Banning fault, etc., even for San Andreas fault.

Comment 47: Lines 204-205 – I think this sentence needs reworked: "The study area comprises three major fold systems that are oblique to the SAFZ. These fold systems are E–W trending, moderately west-plunging, and contain multiple smaller-scale parasitic folds (Fig. 2)."

Comment 48: Lines 243–244 – This is more of an editorial preference by EGU, but I don't think

forelimb and backlimb need dashed? If not, change throughout the manuscript. If so, ignore.

Comment 49: Line 267 - I think you mean southeastern here, not southwestern

Comment 50: Line 314 – Change offset to displacement.

Comment 51: Lines 327-329 – Add "for a damage zone of a": "The granite there is highly fractured and cut by vein and joint networks (see description below), as is expected for a damage zone of a major brittle fault."

Comment 52: Line 377 – minor-scale (needs a dash I think)

Comment 53: Line 386 – in other places I think you refer to it as a leucogranite. Be consistent, whether you choose simply granite or leucogranite.

Comment 54: Lines 396-397 – Suggested rewording: "The folds are arranged in a right-stepping pattern, and are increasingly asymmetric and sigmoidal (Z-shaped) to the northeast as they approach the Indio Hills fault." Change as you see fit, but at present the sentence is difficult to understand.

Comment 55: Line 429 – as inferred for other parts of the SAFZ

Comment 56: Line 430 - remove en dash (-) in front of to

Comment 57: Line 506 – perhaps just say slip here, not "the last slip event"

Comment 58: Lines 560-562 – Your sentence is in present tense ("this is observed") but you refer to the Banning fault as you interpret it to have been at a former time. Perhaps say "what was then a precursory Banning fault."

Comment 59: Lines 601-602 – Should be Eastern California shear zone (says East, not Eastern, and shear zone needs decapitalized)

Comment 60: Lines 607-608 - These two faults do not appear to be on Figure 1

Comment 61: Line 610 - delete comma after "enhanced"

Comment 62: Lines 633-634 – Earlier in the paper (and in Fig. 8) you define main San Andreas fault as mSAF, whereas here you say main SAFZ. Is there a reason for the difference (e.g., one refers to a discrete/singular fault plane, whereas the other refers to the main fault zone)? Should mSAF just be changed to main SAFZ, or vice versa? Also do this at Lines 40, 117, 608, 637, 654, 690, 695, 698, and in various figures.

Comment 63: Line 639 – the Indio Hills fault (missing "the")

Comment 64: Lines 634 and 649 – On line 634 you reference Fig. 8c before referencing 8a and 8b, and on line 649 you reference Fig. 8c before referencing Fig. 8b. You do reference Fig. 8 in its entirety at line 627 – this is more of an editorial decision by EGU if subfigures can be referenced out of sequence.

Comment 65: Line 652 – missing a reference

Comment 66: Line 680 – Eastern Shoreline fault (combine Shore and line)

Comment 67: Line 689 - Here I think you mean Eastern Shoreline fault

Comment 68: Line 692 – see comment above about main SAFZ and mSAF. Here you say main

SAF, which you defined earlier in the paper as mSAF - should this one be mSAF or main SAFZ?

Comment 69: Line 695 and 697 - Eastern Shoreline fault

Comment 70: Line 736 - delete "in"

Comment 71: Detailed comments on figures – Figure 1 – Figure 1 needs a regional scope. At the very least, a regional map showing California and the study area should be squeezed onto to Figure 1. However, I'd suggest a more detailed regional map showing structural relationships in the area and the numerous features mentioned in the text that are not on any of the maps (e.g., San Gorgonio Pass). For example, from Figure 1, the reader at present would have no context to the extent of the Eastern California shear zone. This can be done as a two-panel figure, where

Fig. 1a is a regional map showing major features discussed in the text and the field area, and Fig.1b can be the present Fig. 1 map.

Comment 72: Line 982 – Brawley Seismic Zone needs defined as BSZ in the caption. Comment 73: Lines 985-986 – As in the manuscript, decapitalize shear zone in "Eastern California shear zone" in this caption and in all figure captions. It is okay, of course, for the abbreviation to be ECSZ.

Comment 74: All fault names here, in all figure captions, and throughout the manuscript should not have "fault" capitalized as part of the name. E.g., Banning Fault should be Banning fault, etc. Comment 75: Figure 2 – I hate to be a stickler here because these are GoogleEarth images, but all maps (Figs. 2, 3, 5, 6) should technically have at least a few coordinates, whether as a grid of lat/long or UTM, or a few lat/long coordinate points.

Comment 76: Note that your typed words have the spell check wiggle line underneath them. Make sure your final image does not have these.

Comment 77: On your stereonets labels, I suggest adding that the first two are bedding: "SAFZoblique bedding planes" and "SAFZ-parallel bedding planes"

Comment 78: In the text I think you say faults and fractures, but here you only say fractures. If both were measured, both should be specified here: "Sediments faults and fractures"; "Basement faults and fractures"

Comment 79: What program did you use to make the stereonets? Allmendinger's? You should probably cite the program, unless it is a script you wrote.

Comment 80: Figures 2, 3, 5, and 6 - In the supplemental file you have an un-interpreted figure 6; it would be good to also put un-interpreted images of figures 2, 3, and 5 in the supplemental file as well.

Comment 81: Figures 3, 5, and 6 – Your mapped features (fold hinges and faults) commonly end before the end of the figure's frame, whereas on Figure 2 many of these same features are shown to be continuous across the frame of the figure. I would suggest mapping the features along their full extent and ending them at the end of the figure frame, instead of cutting them short within the figure frame. As currently drawn, it gives the impression to the reader that these folds and faults are short and discontinuous only within the frame of the figure, but Figure 2 shows clearly that many of these features are continuous from one domain into the other. For example, the southeastern corner of Fig. 3b is also the northwestern corner of Fig. 3c - from southwest to

northeast there is an anticline then syncline then overturned anticline then overturned syncline/syncline – I think that these are the same folds in both figures, but as currently drawn Figs. 3b and 3c give the impression these are different folds.

Comment 82: Figure 3 – What are the yellow dots? I think these are photograph locations; if so state this in the Figure 3 caption.

Comment 83: In each panel (a, b, and c), you could place the domain name right above the scale bar. For example: in Fig. 3a, label "Northwestern" above the scale bar, "Central" in 3b, and "Southeastern" in 3c. This would make it easier for the reader.

Comment 84: Figure 4 – Line 1019 – The stereonet represents the cm-scale folds, correct? If so, add "cm-scale" to the caption so it is clear to the reader that these are the small cm-scale folds that cannot be seen in the photos.

Comment 85: Figure 5 – Label "Banning fault" on the main figure.

Comment 86: The fold hinge should be mapped on this figure like it is in Fig. 2.

Comment 87: Figure 7 – Line 1040 – You say Tentative model here; tentative on what? Perhaps just say "Model illustrating…"

Comment 88: Figure 8 – I make this point at Lines 633-634, but in your figure, how does SAFZ differ from mSAF? Is one a discrete fault that is considered the main strand (mSAF) and the other is a zone of deformation (SAFZ)? Is using mSAF necessary?

**2. Author's reply**

Comment 1: agreed. Comment 2: agreed. Comment 3: agreed. Comment 4: agreed. See response to comment 81. Comment 5: agreed. Comment 6: agreed. Comment 7: agreed. Yes, the Indio Hills exhibit folding over underlying thrust ramps, as proposed for the SAEZ-parallel anticline near the Indio Hills fault (see also Supplement S3a for an example

for the SAFZ-parallel anticline near the Indio Hills fault (see also Supplement S3a for an example in the field). We agree though that it is certainly more appropriate to use the term "uplift" instead of "culmination" to avoid confusion.

Comment 8: agreed. See response to comment 9.

Comment 9: agreed.

Comment 10: agreed.

Comment 11: agreed.

Comment 12: agreed. See response to comment 7.

Comment 13: agreed. See response to comment 2.

Comment 14: agreed.

Comment 15: agreed.

Comment 16: agreed.

Comment 17: agreed.

Comment 18: agreed.

Comment 19: agreed.

Comment 20: agreed.

Comment 21: agreed.

Comment 22: agreed.

Comment 23: agreed.

Comment 24: agreed.

Comment 25: the fault would be below the contact, not at/near the contact.

Comment 26: agreed.

Comment 27: agreed.

Comment 28: agreed.

Comment 29: agreed.

Comment 30: agreed.

Comment 31: the Durmid Hills and Indio Hills uplifts are located on either sides of the main San Andreas fault and the E–W-trending macro-folds in the Durmid Hills seem to have formed slightly (but perhaps not significantly) after those in the Indio Hills (see new Table 1 for the timing of the main geological events in the Coachella Valley as suggested by the other reviewer). These uncertainties around the timing of geological events in the various uplifted areas along the main San Andreas fault are, no doubt, related to the sparsity of geochronological ages of structures along the fault. As depicted by the similar timing of uplift in all three uplifted areas, it is probable that all areas evolved (almost) synchronously, but even if it were the case, it is not yet possible to argue

for such a scenario. More geochronological data and absolute ages are needed in this part of California.

Comment 32: agreed.

Comment 33: deformation is meant in a general sense.

Comment 34: agreed.

Comment 35: agreed. However, "North American plate" is the correct term.

Comment 36: agreed.

Comment 37: agreed.

Comment 38: agreed.

Comment 39: agreed. See response to comment 34.

Comment 40: agreed.

Comment 41: agreed.

Comment 42: agreed.

Comment 43: the sentence was deleted and the paragraph reworked according to the other reviewer's comments.

Comment 44: agreed. However, inserted reference earlier than suggested by the anonymous reviewer's comment.

Comment 45: agreed.

Comment 46: agreed.

Comment 47: agreed.

Comment 48: agreed. However, the authors of the present manuscript will wait for proof-reading comments by the editorial team to make the suggested correction in case it is not required by the journal.

Comment 49: agreed.

Comment 50: agreed.

Comment 51: agreed.

Comment 52: agreed.

Comment 53: agreed.

Comment 54: agreed.

Comment 55: agreed.

Comment 56: agreed.

Comment 57: agreed.

Comment 58: agreed.

Comment 59: agreed. Also see response to comment 34.

Comment 60: agreed.

Comment 61: agreed.

Comment 62: disagreed. "main SAFZ" should be changed to "main San Andreas fault".

Comment 63: agreed.

Comment 64: agreed.

Comment 65: agreed.

Comment 66: agreed.

Comment 67: agreed. Also did this throughout the manuscript.

Comment 68: agreed.

Comment 69: agreed. See response to comment 67.

Comment 70: agreed.

Comment 71: agreed. See response to comment 2.

Comment 72: agreed.

Comment 73: agreed. See response to comment 34.

Comment 74: agreed.

Comment 75: agreed. See response to comment 3.

Comment 76: agreed.

Comment 77: agreed.

Comment 78: disagreed. The term "fracture" is general and applies both to "faults" and "fractures".

Comment 79: agreed.

Comment 80: agreed. See response to comment 3.

Comment 81: agreed.

Comment 82: agreed.

Comment 83: since the three macro-folds are shown in order from northwestern to southeastern, it is unnecessary to specify the name of the "domain" on each part of Figure 3. In addition, this could give the impression to the reader that, e.g., only the southeastern macro-fold may be observed on Figure 3c, which is not the case since the central macro-fold is also shown there.

Comment 84: agreed.

Comment 85: agreed. Comment 86: agreed. Comment 87: agreed. Comment 88: agreed.

**3. Changes implemented**

Comment 1: none commanded by the reviewer's comment.

Comment 2: designed new figure 1a and b.

Comment 3: added coordinates to Figure 2, on which all Google Earth images are located. Also added uninterpreted version of all Google Earth images to the supplements and reorganized the supplement numbers in the manuscript.

Comment 4: see response to comment 81.

Comment 5: added ", southwestern USA" in the title.

Comment 6: added "and its role as possible transfer fault" lines 51–52.

Comment 7: replaced "culmination" by "uplift" lines 74, 76, 198, 229, 680, 720 and 980, and by "tectonic uplifts" line 840.

Comment 8: see response to comment 9.

Comment 9: changed "culmination" into "uplift" line 122. Changed "analogous" into "an analog" and "rift" into "inverted" line 124.

Comment 10: replaced "We consider" by "Previous mapping in the area (Dibblee, 1954; Lancaster et al., 2012) considered" lines 139–140, and deleted reference to Dibblee (1954) line 142.

Comment 11: deleted ") are consistent with sediment-accumulation rate estimates (" lines 165–166.

Comment 12: see response to comment 7.

Comment 13: see response to comment 2.

Comment 14: deleted "off-fault" line 249.

Comment 15: added reference to Sylvester and Smith (1976, 1979, 1987) and Bergh et al. (2019) lines 256–257.

Comment 16: replaced "via" by "to" line 319.

Comment 17: replaced "via" by "to" line 345.

Comment 18: deleted "to monocline-like" lines 379–380.

Comment 19: replaced "folds" by "synclines" line 407, and added "*(synclines)*" line 1321 and the hinge line of the folds in Fig. 5.

Comment 20: deleted "fold-related" line 416.

Comment 21: added "(see stereoplot in Fig. 2)" line 506. Deleted ", possibly representing, conjugate sets" lines 505–506.

Comment 22: replaced "a younger slip event" by "younger deformation along this fault" lines 548–549.

Comment 23: deleted "strain" line 552.

Comment 24: replaced "Palm Spring and Mecca foramtions" by "Mecca Formation and overlying Palm Spring Formation" lines 570–571.

Comment 25: none.

Comment 26: deleted "dip-slip" and replaced "parallel" by "propagation" lines 591–592.

Comment 27: replaced "stress" by "strain" line 673.

Comment 28: replaced "restored" by "rotated" line 497, and "restoring" by "rotating" lines 500 and 686.

Comment 29: replaced "trending" by "oriented" line 744.

Comment 30: deleted "in a changing transpressional plate regime" line 884.

Comment 31: added "The *en echelon* folds formed at a comparable time, i.e., < 0.76 Ma in the Indio Hills and at ca. 0.5 Ma in the Durmid Hills (Table 1)." lines 919–920.

Comment 32: added "in California, southwestern USA" lines 34-35.

Comment 33: none.

Comment 34: changed "Eastern California Shear Zone" into "Eastern California shear zone" throughout the manuscript.

Comment 35: replaced "transform movements" by "movement" line 67.

Comment 36: added "These recent works call for further characterization of the understudied Indio Hills segment in order to compare its structural development with other uplifted features along a major transform plate boundary fault zone." lines 72–75.

Comment 37: corrected into "San Gorgonio Pass" line 216.

Comment 38: split the sentence into two line 217.

Comment 39: see response to comment 34.

Comment 40: deleted "attitude and" line 226.

Comment 41: changed "remain" into "remains" line 227.

Comment 42: split the sentence into two and replaced ", but" by "However" line 231.

Comment 43: the sentence was deleted and the paragraph reworked according to the other reviewer's comments.

Comment 44: replaced "their study" by "Keller et al. (1982) lines 240-241.

Comment 45: added "San Andreas" and deleted "strand" line 273.

Comment 46: decapitalized "fault" throughout the manuscript.

Comment 47: added a comma before and after "SAFZ-oblique" line 307 and changed "with" into "having" line 308.

Comment 48: none for the moment. Awaiting comments by the editorial team.

Comment 49: changed "southwestern" into "southeastern" line 379.

Comment 50: replaced "offset" by "displacement" line 429.

Comment 51: replaced "near" by "in the damage zone of" line 446.

Comment 52: added an hyphen between "minor" and "scale" line 501.

Comment 53: deleted "leuco-" line 444.

Comment 54: changed sentence into "In map view (Fig. 2), the folds are right-stepping, and each

fold set is increasingly asymmetric (Z-shaped) and sigmoidal towards the Indio Hills fault in the northeast." lines 523–525.

Comment 55: added "other" line 570.

Comment 56: deleted en dash line 571.

Comment 57: replaced "slip event" by "episode of movement" line 675.

Comment 58: deleted "precursory" line 753.

Comment 59: replaced "East" by "Eastern" line 810. Also see response to comment 34.

Comment 60: added the Camp Rock and Calico faults to Figure 1.

Comment 61: deleted comma after "enhanced" line 825.

Comment 62: replaced "SAFZ" by "San Andreas fault" lines 853-854.

Comment 63: replaced "in" by "along the" line 860.

Comment 64: added "a-c" line 844.

Comment 65: replaced missing figure reference by "Figs 2 & 3c and Supplement S3a" line 875.

Comment 66: combined "Shore" and "line" line 904.

Comment 67: replaced "Shore" by "Shoreline" lines 920, 928, and 929.

Comment 68: replaced "main SAF" by "main San Andreas fault" line 930.

Comment 69: see response to comment 67.

Comment 70: deleted "in" line 976.

Comment 71: see response to comment 2.

Comment 72: added "; BSZ: Brawley seismic zone" line 1285.

Comment 73: see response to comment 34.

Comment 74: uncapitalized "Fault" throughout the manuscript.

Comment 75: see response to comment 3.

Comment 76: adjusted text in figures 2, 7, and 8.

Comment 77: added "(bedding)" twice in figure 2.

Comment 78: none.

Comment 79: added "via the Orient software (Vollmer, 2015)" line 1304 and Vollmer (2015) to the reference list.

Comment 80: see response to comment 3.

Comment 81: adjusted the hinge line of structures in Figures 3a-c, 5, and 6.

Comment 82: added "The yellow dots show the location of field photographs." line 1319.

Comment 83: none.

Comment 84: added ", centimeter-scale" line 1329.

Comment 85: added "main San Andreas fault" to Figure 5.

Comment 86: re-drew the fold hinge in Figure 6 as it appears in Figure 2.

Comment 87: deleted "Tentative" line 1351.

Comment 88: changed "SAFZ" into "mSAF" in Figure 8.

**Additional revisions by the author of the present manuscript**

-Added "(we refrain from using the name "Indio strand" given to this fault by Gold et al., 2015 to avoid confusion with the Indio Hills fault)" lines 39–41.

-Moved "Atwater and Stock, 1998;" before "Spotila et al., 2007;" line 64.

-Changed "uppermost members" into "upper member" line 118.

-Replaced "marks the" by "is a" line 201.

-Deleted "-" line 480.

-Deleted en-dash line 506.

-Added "(probably soutwest-dipping)" line 753.

-Replaced "show" by "suggest" line 786.

-Added a comma line 831.

-Corrected "Janecke et al., 2019" into "Janecke et al., 2018" line 842.

-Deleted "and main SAFZ" line 863.

-Moved "basement-seated" from line 899 to line 891.

-Deleted "The Indio Hills fault acted as a SW-dipping, normal fault in Miocene time, i.e., prior to inversion as an oblique-slip, right-lateral-reverse fault during mid (–late?) Pleistocene times" lines 899–901.

-Moved ", whereas the main San Andreas fault initiated probably as a dominantly right-slip fault during the later stages of uplift in the late Pleistocene." from lines 903–904 to lines 894–896.

-Deleted "portion of the SAFZ" line 912.

-Deleted "in Durmid Hills" line 1235.

**Reply to Jonathan Matti**

Dear Dr. Matti,

thank you very much for your input on the manuscript, it is highly appreciated. Here is our reply to your comments. We hope the changes we implemented improve the shortcomings of the manuscript highlighted by your comments and suggestions. Please do not hesitate to contact us shall this not be the case for some comments.

**1. Comments from Dr. Matti**

Manuscript se-2022-9 consists of three parts:

- a detailed structural analysis of macro- and micro-folds and associated faults that deform a sequence of Pliocene-lower Pleistocene sedimentary rocks exposed in the tectonically uplifted Indio Hills;
- a comparison of the Indio Hills structural geology with that of two similar uplifted and inverted late Cenozoic basin fills occurring farther SE within the San Andreas Fault zone (SAFZ);
- integration of the structural data into a synthesis that interprets coeval uplift of the various inverted basins in the context of Quaternary dextral-oblique transpressive tectonics within the southern San Andreas Fault system writ large.

The manuscript explores these three themes with mixed success:

- The discussion of fold and fault structures in the Indio Hills is robust and comprehensive, including appropriate analytical data and exceptional aerial and outcrop photographs that nicely illustrate structural features and relationships. One concern I have is that the structural terminology and technical language used in the manuscript are pitched toward the structural specialist—not toward general geologists like myself. I address this point below.
- The manuscript's comparison of the Indio Hills structural setting with that farther to the southeast within the SAFZ is moderately successful. The report depends heavily on results of other published investigations, and provides only cursory discussion of structural correlations and comparisons among the three inverted basins. The report would benefit from expanded discussion of these correlations, including one or more new map-type figures that better summarize geologic structures SE of the Indio Hills (otherwise, the reader has to chase the

other publications down in order to evaluate manuscript se-2022-9's proposed structural comparisons and correlations).

By comparison with the preceding two themes, the manuscript's regional synthesis in my
opinion is the weakest link in the three themes. In my review I raise some technical questions
and issues that I believe need to be addressed more completely—and in some cases explained
or corrected. These are not deal-breakers, but should be addressed by the authors.

I do not know whether Copernicus Publications provides an extensive review by a science editor, but I think that manuscript se-2022-9 needs a heavy editorial hand—either by Copernicus staff or by the authors themselves based on peer-review feedback. In part, problems with the narrative structure may stem from the fact that English may not be the first language of two of the three authors. But in addition, I sense that the narrative is too cursory and includes logic jumps that need to be explained more fully. My marginal comments on the manuscript identify many specific instances where I think the narrative can be improved both content-wise and in terms of organization.

All of this said, I enjoyed reading the manuscript. First, it adds to the body of detailed structural analysis so critical to documenting and understanding the geologic history of the southern San Andreas Fault zone and associated depositional basins; and second, it provides a testable regional synthesis for dextral and contractional events within the SAFZ writ large—including possible interactions with the Eastern California Shear Zone and the sequential development of discrete SAFZ strands in the Salton Trough.

My recommendation: The manuscript needs work, but it should be published by Copernicus Solid Earth.

\_\_\_\_\_

My review consists of two parts:

- General comments contained in this memo
- Detailed comments, questions, and suggested edits integrated into the .pdf version of Manuscript se-2022-9.

NOTE: For my review I separated the manuscript into four discrete documents: (1) the text without references, (2) references alone, (3) figures alone, and (4) supplemental material.
* * *
I enjoyed reviewing this manuscript, although I have questions and comments that may (or may not) improve the paper. I trust that the authors will receive my comments and critique in the spirit with which they are offered: to refine and clarify an important contribution our understanding of the tectonic evolution of the southern San Andreas Fault system.

Good luck with forward progress of the manuscript.

\_\_\_\_\_

Behr, W.M., Rood, D.H., Fletcher, K.E., Guzman, N., Finkel, R., Hanks, T.C., Hudnut, K.W., Kendrick, K.J., Platt, J.P., Sharp, W.D., Weldon, R.J., Yule, J.D., 2010, Uncertainties in slip-rate estimates for the Mission Creek strand of the southern San Andreas Fault at Biskra Palms Oasis, Southern California: GSA Bulletin v. 122, no. 9-10, p. 1360-1377.

Cohen, K.M., Finney, S.C., Gibbard, P.L., and Fan, J.-X., 2013 (updated 5/2021), The ICS International Chronostratigraphic Chart, version 2021-5: Episodes, v. 36, p. 199-204, accessed 7/16/21 at http://www.stratigraphy.org/ICSchart/ChronostratChart2021-05.pdf.

Dokka, R.K., and Travis, C.J., 1990a, Late Cenozoic strike-slip faulting in the Mojave Desert, California: Tectonics, v. 9, p. 311-340, accessed 3/8/2022 at

https://agupubs.onlinelibrary.wiley.com/doi/abs/10.1029/TC009i002p00311.

Dokka, R.K., and Travis, C.J., 1990b, Role of the eastern California shear zone in

accommodating Pacific-North American plate motion: Geophysical Research Letters, v. 17, p.

1323-1326, accessed 3/8/2022 at

https://agupubs.onlinelibrary.wiley.com/doi/abs/10.1029/GL017i009p01323.

Fattaruso, L.A., Cooke, M.L., Dorsey, R.J., Housen, B.A., 2016, Response of deformation

patterns to reorganization of the southern San Andreas fault system since ca. 1.5 Ma:

Tectonophysics, v. 693, Part B, p. 474-488. doi: 10.1016/j.tecto.2016.05.035.

Fuis, G.S., Scheirer, D.S., Langenheim, V.E., and Monica D. Kohler, M.D., 2012, A new perspective on the geometry of the San Andreas Fault in southern California and its relationship to lithospheric structure: Bulletin of the Seismological Society of America, v. 102, no. 1, p. 236–251, doi: 10.1785/0120110041.

Fuis, G.S., Bauer, K., Goldman, MR., Ryberg, T., Langenheim, V.E., Scheirer, D.S., Rymer,M.J., Stock, J.M., Hole, J.A., Catchings, R.D., Graves, R.W., and Aagaard, B., 2017, Subsurfacegeometry of the San Andreas Fault in southern California: Results from the Salton Seismic

Imaging Project (SSIP) and strong ground motion expectations: Bulletin of the Seismological Society of America, Vol. 107, No. 4, p. 1642–1662, doi: 10.1785/0120160309 Gibbard, P.L., Head, M.J., Walker, M.J.C., and the Subcommission on Quaternary Stratigraphy, 2010, Formal ratification of the Quaternary System/Period and the Pleistocene Series/Epoch with a base at 2.58 Ma: Journal of Quaternary Science, v. 25, no. 2, p. 96–102. https://doi.org/10.1002/jqs.1338.

Gold, P.O., Behr, W.M., Rood, D., Sharp, W.D., Rockwell, T.K., Kendrick, K., and Salin, A., 2015, Holocene geologic slip rate for the Banning strand of the southern San Andreas Fault, southern California: Journal of Geophysical Research, Solid Earth, v. 120, 25 p., accessed at https://doi.org/10.1002/2015JB012004.

Guns, K.A., Bennett, R.A., Spinler, J.C., and McGill, S.F., 2020, New geodetic constraints on southern San Andreas fault-slip rates, San Gorgonio Pass, California: Geosphere, v. 17, p. 39–68, accessed 3/8/2022 at https://doi.org/10.1130/GES02239.1.

Kendrick, K.J., Matti, J.C., and Mahan, S.A., 2015, Late Quaternary slip history of the Mill Creek strand of the San Andreas fault in San Gorgonio Pass, southern California: The role of a subsidiary left-lateral fault in strand switching: Geological Society of America Bulletin, v. 127, p. 825-849, accessed at https://doi.org/10.1130/B31101.1

Lancaster, J.T., Hayhurst, C.A., and Bedrossian, T.L., 2012, Preliminary geologic map of Quaternary surficial deposits in southern California: Palm Springs 30' x 60' quadrangle, in Bedrossian, T.L., Roffers, P., Hayhurst, C.A., Lancaster, J.T., and Short, W.R., Geologic compilation of Quaternary surficial deposits in southern California December 2012 https://www.conservation.ca.gov/cgs/fwgp/Pages/sr217.aspx#palmsprings.

Matti, J.C., and Morton, D.M., 1993, Paleogeographic evolution of the San Andreas fault in southern California: a reconstruction based on a new cross-fault correlation, in Powell, R.E., Weldon, R.J., and Matti, J.C., eds., The San Andreas fault system: displacement, palinspastic reconstruction, and geologic evolution: Geological Society of America Memoir 178, p. 107-159. Matti, J.C., Morton, D.M. and Cox, B.F., 1992, The San Andreas fault system in the vicinity of the central Transverse Ranges province, southern California: U.S. Geological Survey Open-File Report 92-354, 40 p., scale 1:250,000. https://pubs.er.usgs.gov/publication/ofr92354 Matti, J.C., Kendrick, K.J., Yule, J.D., and Heermance, R.K., 2019, The Mission Creek Fault in the San Gorgonio Pass region—A long-abandoned strand of the San Andreas Fault, or a major

player in the latest Quaternary San Andreas strain budget? Geological Society of America

Cordilleran Section Meeting, Paper 329432

https://gsa.confex.com/gsa/2019CD/meetingapp.cgi/Paper/329432

McCaffrey, R., 2005, Block kinematics of the Pacific–North America plate boundary in the southwestern United States from inversion of GPS, seismological, and geologic data: Journal of Geophysical Research Solid Earth, v. 110, p. B07401, doi: 10.1029/2004JB003307, accessed 3/8/2022 at https://agupubs.onlinelibrary.wiley.com/doi/10.1029/2004JB003307.

Nur, A., Hagai, R., and Beroza, G., 1993a, Landers-Mojave earthquake line: a new fault system?: GSA Today, v. 3, p. 253, 256-258, accessed 3/9/2022 at

https://www.geosociety.org/gsatoday/archive/3/10/pdf/i1052-5173-3-10-sci.pdf.

Nur, A., Hagai, R. and Beroza, G. C., 1993b, The Nature of the Landers-Mojave Earthquake

Line: Science, v. 261, p. 201–203, https://www.science.org/doi/10.1126/science.261.5118.201.

Passchier, C.W., and Platt, J.P., 2017, Shear zone junctions: Of zippers and freeways: Journal of

Structural Geology, v. 95, p. 188-202, accessed 3/9/2022 at

https://doi.org/10.1016/j.jsg.2016.10.010.

Pillans, B., and Gibbard, P., 2012, The Quaternary Period, in Gradstein, F.M., Ogg, J.G.,

Schmitz, M.D., and Ogg, G.M., eds., The Geologic Time Scale: Elsevier, p. 979–1010, accessed 7/2/2021 at https://doi.org/10.1016/B978-0-444-59425-9.00030-5.

Platt, J.P., and Passchier, C.W., 2016, Zipper junctions: A new approach to the intersections of conjugate strike-slip faults: Geology, v. 44, p. 795-798, accessed 3/9/2022 at https://doi.org/10.1130/G38058.1.

Powell, R.E., Matti, J.C., and Cossette, P.M., 2015, Geology of the Joshua Tree National Park geodatabase: U.S. Geological Survey Open-File Report 2015–1175, GIS database, accessed 3/9/2022 at https://pubs.er.usgs.gov/publication/ofr20151175.

Rymer, M.J., 2000, Triggered surface slips in the Coachella Valley area associated with the 1992 Joshua Tree and Landers, California, earthquakes: Bulletin of the Seismological Society of America, v. 90, p. 832-848, DOI: 10.1785/0119980130, accessed 3/8/2022 at

https://pubs.geoscienceworld.org/ssa/bssa/article/90/4/832/120523/Triggered-Surface-Slips-in-the-Coachella-Valley.

Spotila, J.A., and Garvue, M.M., 2021, Kinematics and evolution of the southern Eastern California shear zone, based on analysis of fault strike, distribution, activity, roughness, and secondary deformation: Tectonics, v. 40, 32 p., e2021TC006859, accessed 3/9/2022 at https://doi.org/10.1029/2021TC006859.

Treiman, J.A., 1992a, Eureka Peak and related faults, San Bernardino and Riverside Counties,
California: California Geological Survey [formerly California Division of Mines and Geology]
Fault Evaluation Report FER-230, scale 1:24,000, accessed 3/11/2022 at
https://maps.conservation.ca.gov/geologichazards/DataViewer/index.html.
Treiman, J.A., 1992b, Eureka Peak and Burnt Mountain faults, two "new" faults in Yucca Valley,
San Bernardino County, California, in Ebersold, D.B., ed., Landers earthquake of June 28, 1992,
San Bernardino County, California: Los Angeles, Association of Engineering Geologists,
southern California section, Annual Field Trip Guidebook, p. 19-22.
Walker, M., Head, M.J., Lowe, J., Berkelhammer, M., Björck, S., Cheng, H., Cwynar, L.C.,
Fisher, D., Gkinis, V., Long, A., Rewi, N., Rasmussen, S.O., and Weiss, H., 2019, Subdividing
the Holocene Series/Epoch: formalization of stages/ages and subseries/subepochs, and
designation of GSSPs and auxiliary stratotypes: Journal of Quaternary Science, v. 34, no. 3, p.
173-186, accessed 7/13/2021 at https://doi.org/10.1002/jqs.3097.

Yule, D., Matti, J., Kendrick, K., and Heermance, R., 2019, Evidence for inactivity since ~100 ka on the northern route of the San Andreas fault, southern California: Geological Society of America Cordilleran Section Meeting, Paper 16-8

https://gsa.confex.com/gsa/2019CD/meetingapp.cgi/Paper/329441

Matti, J.C., and Yule, J.D., 2020, Does a "structural knot" in the Quaternary San Andreas Fault exist in San Gorgonio Pass? Traditional and recent views provide conflicting answers: Geological Society of America Abstracts with Programs, v. 52, no. 4. doi: 10.1130/abs/2020CD-347317

Comment 1: General comment #1: Who is your audience?—In my opinion it is not clear who manuscript se-2022-9 is trying to reach: the specialist in structural geology? Or the regional geologist who primarily is interested in reconstructing the tectonic history of the SAFZ and related faults over the last 6 ma?

I assert this because the structural analysis of fold and faults in the Indio Hills and their kinematic interpretation (theme 1, above) is laden with specialized structural terms with which the average geologist will not be familiar. This easily can be solved by the author's sensitivity to those

geologists that are interested in the paper but become irritated when the technical language stands in the way of understanding local and regional structures.

This easily can be addressed—not by dumbing down and diluting the structural contributions but rather by using techniques like the following example:

Instead of "Farther southeast along strike, the Indio Hills and Banning faults merged along a dextral freeway junction (Platt and Passchier, 2016) that may have enhanced...." (manuscript lines 610-611), consider the following:

"Farther southeast along strike, the Indio Hills and Banning faults presumably merge along a dextral freeway junction—a type of fault intersection where the faults have similar shear sense in all three branches (see Platt and Passchier, 2016; Passchier and Platt, 2017). This configuration may have enhanced......". (BTW, you may want to add the Passchier and Platt [2017] citation to your list of references).

I recommend you use this type of narrative format to speak both to the structural geologist (probably familiar with the term already) and to the regional geologist like me (inquiring minds want to know).

Specialized structural terms are scattered throughout the manuscript. Here are a few that could be explained:

fore-limb (of folds)

back-limb (of folds)

ladder structure

shear-folding (as opposed to other fold drivers)

Comment 2: General comment #2: Discussion of faults in the greater Coachella Valley region—Manuscript se-2022-9 discusses faults of the greater San Andreas system (writ large) in three separate segments of the report: lines 39-47, lines 116-124, and lines 320-336. Not only are these lines scattered throughout various parts of the report (thus making it hard for the reader to keep track of which faults are doing what and when), but the scattered text contain assertions and interpretations that the reader has to remember and appreciate from isolated sections, and then relate within a total picture of tectonic history stretching over 6-7 million years. This is difficult to do without a well-organized and complete section at the front of the manuscript that summarizes regionally-important faults throughout the greater Coachella Valley region. Absent this introductory summary, the reader reaches no sense of structural complexity within the SAFZ in southern California—both in terms of discrete faults strands throughout the region and how they evolved through time and space.

Why is a coherent introductory regional statement needed?

The manuscript ostensibly focuses on structural relations from the latitude of the southern Indio Hills south. However, the report integrates certain regional faults, concepts, and nomenclature not only into its concluding tectonic synthesis but also into its use of fault names locally. This especially is apparent with how the authors use the name "Banning Fault" (General Comment 3 below) and with how they integrate late Quaternary strain history in the Indio Hills with modern strain patterns in the Eastern California Shear Zone (General Comment 5 below). Absent a coherent introductory summary of regional fault relations, the reader can't help but believe that the manuscript's findings in the southern Indio Hills (and similar domains to the southeast) resolve all issues related to strain distribution in the southern SAFZ throughout the last 6-7 Ma.

Recommendation:

To address this, I recommend that all discussion of regional faults be moved into a single section under "Geologic Setting", following an outline like this (or something like it):

Geologic Setting

Regional faults (including what is known about fault ages; see figure 2 in Kendrick and others, 2015)

Regional stratigraphy (already discussed in the manuscript)

Regional Tectonic Culminations (already discussed in the manuscript)

This new section hopefully will incorporate (and resolve) issues and questions identified in General Comments 3, 4, and 5 (below).

Comment 3: General comment #3: Use of the term "Banning Fault"—The manuscript applies this fault name from the San Gorgonio Pass region southeast beyond the southern Indio Hills (see figures 1 and [especially] 2, and lines 39-42, 633-634). This runs counter to the way most workers interpret faults and fault names.

The problem: Because the manuscript lacks a coherent discussion of fault nomenclature, distribution, and movement history in the greater Coachella Valley region, the reader reaches no sense of structural complexity within the southern California SAFZ—both in terms of discrete faults strands throughout the Coachella Valley region, how they evolved through time and space,

and how they interacted together. Although the manuscript ostensibly focuses on structural relations from the latitude of the southern Indio Hills south, it nevertheless brings certain concepts (and attendant nomenclature) southward from the northern Coachella Valley where structural relations are more complex than implied in the manuscript. The reader can't help but believe that the findings in manuscript se-2022-9 resolve all remaining issues related to strain distribution in the southern SAFZ throughout the last 6-7 Ma.

Manuscript se-2022-9's use of the term "Banning Fault" inadvertently (but unfortunately) contributes to this problem

**Recommendation:**

If manuscript se-2022-9 retains its current nomenclatural approach regarding the Banning Fault, at a minimum the report needs to address how its usage differs from that of other workers (discussed below). It would be better if the authors evaluated regional tectonic implications of extending the name "Banning Fault" as far south as they do—especially because northwest of their study area the fault has been shown to have a very limited time during which if functioned as a discrete strand of the SAFZ, and this time frame is incompatible with that the authors propose for the "Banning Fault" in the southern Indio Hills.

In short: Manuscript se-2022-9 needs to acknowledge that the timing they propose for movement on the "Banning Fault"—and the important structural role it plays in their analysis of how the tectonic culmination evolved—is not compatible with what is known about slip on the Banning strand of the SAF in the northern Coachella Valley.

Nomenclatural and fault-reconstruction precedents:

Matti and others (1992; Matti and Morton, 1993) have addressed nomenclature problems for strands of the SAFZ in the Coachella Valley region. They applied the name "Banning Fault" southward from eastern San Gorgonio Pass to the fault's junction with the Mission Creek Fault midway along the Indio Hills. In addition, they alluded to "Coachella Valley segments" of the various faults, anticipating future nomenclatural refinements that have emerged over the last decade or so.

Recent investigators follow this precedent regarding the spatial extent of the "Banning Fault". Behr and others (2010), Fuis and others (2012, 2017), Fattaruso and others (2014, 2016), Gold and others (2015), Kendrick and others (2015), Beyer and others (2018), and most other workers do not apply the term "Banning Fault" southeast of its junction with the Mission Creek strand. Beyond that juncture, some workers apply the name "Indio strand" to the SAF (Behr and others, 2010; Gold and others, 2015; see fig. 1 of both reports). Other workers apply the name "Coachella Valley segment of the SAF" to the fault southeast of the juncture (see Fattaruso and others, 2014, 2016).

You are not obliged to use the more common usage of "Banning Fault". However, it is incumbent on you to address the nomenclature issue.

If you choose to revise your nomenclatural approach, then you need to come up with a name that you can apply to the "San Andreas Fault" strand southeast of the Banning/Mission Creek junction. Given that—in the Indio Hills region—the Mission Creek Fault was the major SAF strand during the period 4 Ma to Holocene (the critical period during which you require a bounding dextral fault on the SW side of the Indio Hills culmination), I personally would apply the name "Mission Creek Fault" in place of your universal application of the name "Banning Fault" to this structure.

Finally, almost all workers agree that the Banning strand of the SAF between San Gorgonio Pass and its juncture with the Mission Creek strand evolved during the late Quaternary time (last 200 ka???) as the result of a left step from the Mission Creek to a newly evolving strand to the west (i.e., the Banning Fault) (Matti and others, 1985; 1992; Matti and Morton, 1993) (especially see the important paper by Gold and others, 2015, that explores exhumation and uplift rates in the northwestern Indio Hills—a story that sounds a bit like your own, only younger?).

Comment 4: General comment #4: Indio Hills Fault—Manuscript se-2022-9 proposes that the Indio Hills Fault plays a significant structural role in both (1) evolution of the local tectonic culmination and (2) the evolution of the SAFZ. The authors identify three phases of movement history for the Indio Fault:

- An initial role as a southwest-dipping normal fault (late Miocene);
- An intermediate role as a dextral strike-slip fault;
- A penultimate and current role as an active transpressive dextral-oblique thrust fault. I found it difficult to understand its polyphase role (normal fault followed by dextral-slip fault followed by oblique reverse-thrust fault)—especially the timing of activity during each tectonic phase. Comments and observations and interpretations about this structure are scattered throughout the manuscript, so it was hard for me to keep these three phases in mind and to appreciate when each was active.

New to my awareness is the manuscript's proposal that the Indio Hills Fault initially was a late Miocene normal-slip fault (1, above). This assertion needs to be supported with evidence. The authors at some places in the report point to previous workers who propose a southwest-dipping "basin-and-range" type of structure that had to exist early in the evolution of the San Andreas Fault system in the Salton Trough, but I am not aware that the Indio Hills structure was part of that "basin-and-range"-type system. Please explain and elaborate, including where this regional "basin-and-range"-type system can be recognized NW of the Indio Hills

Comment 5: General comment #5: Landers-Mojave Line connection with Indio Hills

**Fault**—I have problems with how the manuscript projects the Indio Hills Fault into a seismic trend that Nur and others (1993a, b) identified as the "Landers-Mojave Line". That "concept" was defined to represent a seismicity belt that was observed following the 1992 Landers earthquake in the Mojave Desert (note that a recent paper by Spotila and Garvue (2021) challenge some of the assertions by Nur and others, 1993a, b). Manuscript se-2022-9 asserts that the Indio Hills Fault can be connected structurally with the Landers-Mojave Line via faults in the Little San Bernardino Mountains (LSBM).

My concerns include:

• The manuscript (fig. 1) connects the Indio Hills Fault northwestward to a presumed fault at the south edge of the LSBM. Although this fault is depicted in some publications, its distribution, structural role, and age have not been documented. Therefore any reference to this fault in manuscript se-2022-9 needs to acknowledge this reality.

I recommend that you cite the recent digital geologic-map database of Joshua Tree National Park by Powell and others (2015) for a more recent and detailed rendering of geologic units and faults. The report can be viewed only in a GIS (ArcMap, for example), but once loaded into a GIS platform the files reveal much more about JTNP geology than was known previously.

• The unnamed fault is depicted by Rymer (2000, fig. 1) who plots it east of his West Deception Canyon Fault. Although Rymer (his figs. 1 and 2) shows the epicenter of the 1992 Joshua Tree earthquake located a few km north of the unnamed fault, he did not report any ground rupture on it. Instead, Rymer documented ground rupture on the West Wide Canyon Fault (see his figure 2). This structure is well to the NW of where manuscript se-2022-9 speculates that the "active" Indio Hills Fault would intersect the LSBM and connect with the "Landers-Mojave Line".

- In lines 600-603 the authors reference Dokka and Travis (1993a, b) in support of the hypothesis that the Indio Hills Fault [strand of the San Andreas Fault] "connects" with the Eastern California Shear zone (ECSZ) and the "Landers-Mojave Line of Nur and others (1993). Reference to Dokka and Travis as supportive evidence for this hypothesis is not appropriate because those authors (1990a, b) do not show the ECSZ extending south of the left-lateral Pinto Mountain Fault (see figs. 2, 14, 15, and 18 of Dokka and Travis, 1990a, and figs. 2 and 3 of Dokka and Travis, 1990b). In fact, Dokka and Travis, 1990b, p. 1325 clearly explore the notion that connection between the ECSZ and the San Andreas Fault (in this case, the Indio Hills strand) is based on slip budgets for the North American plate margin—the physical and kinematic basis for this connection is not obvious.
- Occurrence of (a) ground rupture triggered by the 1992 Joshua Tree earthquake and (b) ground rupture on the Eureka Peak fault south of the Pinto Mountain Fault during the 1992 Landers earthquake (Treiman, 1992a, b) is tempting evidence that strain probably is transferred kinematically between the southern San Andreas Fault and the ECSZ. Note, however, that ground rupture associated with the Joshua Tree event was not coextensive with the Eureka Peak Fault. Thus, it is unlikely that transfer of strain between the ECSZ and the SAFZ occurs along a single fault trace (the authors don't claim that it does, but their figure 1 implies as much. Better to clarify).
- The California Geological Survey classifies the Indio Hills Fault as a late Pleistocene and older feature, with no evidence for Holocene displacement. For current interpretation of fault activity, see California's interactive geologic-hazards map at https://maps.conservation.ca.gov/geologichazards/DataViewer/index.html and also the Quaternary Fault and Fold Database at https://www.usgs.gov/programs/earthquake-hazards/faults). These data call into question the notion that the Indio Hills Fault is a Holocene extension of the Holocene and Recent ECSZ and Nur's Landers-Mojave Line.
   The authors probably will protest that the scope and purpose of manuscript se-2022-9 is much broader and regional in scope to address details of the kind that I provide here in General Comment 5. I agree. I provide my analysis mainly to remind the authors that any model that incorporates latest Quaternary activity on the NE-bounding structure of their Indio Hills tectonic culmination (in this case, the Indio Hills Fault) needs to be compatible with what is known about the distribution and geologic history of fault elements that might (or might not) connect their

tectonic model for the SAFZ with the ECSZ—or with SAFZ structures northwest of the Indio Hills culmination.

Comment 6: General comment #6: Ages for fault activity—In general, I found it quite difficult to determine the sequencing and ages for faults the manuscript discusses and integrates into their concluding time-space model. This is very frustrating because the timing of fault movements (1) relative to overall history of SAF history in the greater Coachella Valley region and (2) relative to when and how the Indio Hills tectonic culmination evolved is a critical part of the author's tectonic model.

**Recommendations:**

- Develop a new section called "Summary of fault ages", and consolidate all the disparate observations about age of faulting currently scattered throughout the report.
- I recommend developing a new diagram like figure 2 of Kendrick and others (2015).
- Make certain that the manuscript's use of "Pliocene", Pleistocene", and "Holocene" conform to current international standards (see Pillans and Gibbard, 2012; Cohen and others, 2013; Gibbard and others, 2013; Walker and others, 2019; and other references on this subject). The boundary between Pliocene and Pleistocene now is ~ 2.6 Ma.

Comment 7: General comment #7: "Possibly", "or", and "may have"—The manuscript frequently has sentences like the following:

"Structural feature X formed by process Y, and (or) it [possibly, may have] formed by process Z".

As a reader, I asked myself "are the authors not committing themselves to structural feature X formed by process Y—their first choice"? Adding caveats like "or may have" makes sentences containing this kind of grammatical structure [presentation] sound like the authors are not sure about their assertions, and are covering themselves.

Recommendation:

• Examine the narrative, find those kind of grammatical instances, and design a more appropriate way of expressing the level of confidence the authors have in conclusions X and Y—in other words, include [discuss] the error bars that prevent complete confidence.

In short: the authors need to choose more definitively among the suite of interpretive possibilities, and not just cover their hypotheses with "or alternatively it could be a different way"! As Gozer challenged the Ghostbusters: "Choose the form of the Destructor".

**Comment 8: General comment #8: Identity, position, and age of "SW master bounding**

**fault"**—It may just be me, but I had trouble understanding what the SW-bounding master fault wasthroughout the evolution of the Indio Hills uplift, where was it positioned throughout this evolution, when did the uplift start, how long did it last, and is it still active? Comments here not only are relevant to the narrative but also to figures—especially figure 7.

Regarding the what: In general comment #3 I questioned your application of the name "Banning Fault" to the SAF strand on the SW flank of the Indio Hills uplift. My comment there pointed out that (according to current understanding) the Banning strand of the SAF in the northwestern Coachella Valley became active only in the last few hundred thousand years (late Pleistocene and slightly older). That "fact" calls into question whether the "Banning Fault" could have been the SW-bounding "master fault" during (say) 500 ka? 1.0 Ma? 1.5 Ma? 2.0 Ma? So, together with my concerns in Comment #3 I question whether you should use the term "Banning Fault" for whatever SAF strand may have formed the "master fault" bounding the SW side of the long-evolving Indio Hills uplift.

But if current thinking regarding the age of the Banning strand of the SAF in the northwestern Coachella Valley is correct, then during the last few hundred thousand years that strand was feeding slip SE along the Pacific margin of the Indio Hills uplift, so at that time application of the name "Banning" to that SW-bounding master fault may have been appropriate (as implied in fig. 7c) (but only during that slip episode).

Bottom line: Multiple SAF strands northwest of the Indio Hills have been active throughout the last 6-8 Ma, those strands have evolved sequentially (Kendrick and others, 2015, fig. 2), each of those strands has a different name, and each of those strands sequentially has fed dextral slip southeast toward the Indio Hills uplift. So the SAF bounding the SW margin of the Indio Hills has had a "changing name" throughout the total 6-8 Ma of southern SAF evolution in the Coachella Valley.

This is why application by Behr and others (2010) and Gold and others (2015) of the term "Indio strand" or "segment" to the SAF southeast of the junction between the Mission Creek and Banning Faults is so appealing: throughout time, all the messy SAF strands NW of the Indio Hills have sequentially evolved northwestward of the Indio strand—presumably NW of the current junction between the Mission Creek and Banning strands of the SAF.

Regarding the where: Your figures 7a and 7b position a queried "Banning Fault" west of the trace of the "Banning" shown in fig. 7c. Why do you do this? What is the basis for the location difference?

Regarding the when and for how long: In my General Comment #6 I recommended a new section that consolidates all fault-age information currently scattered throughout the report—or not addressed clearly. I also recommended a new figure like figure 2 of Kendrick and others (2015). Such a figure easily could add a "range-bar" for the Indio Hills tectonic culmination, thereby resolving my questions about the when and how long.

Regarding the still active?: In my General Comment #5 I questioned your correlation of the Indio Hills Fault with the "Landers-Mojave Line" of Nur and others (1993a, b). Depending on how you address my comments there, the Indio Hills fault may (or may not) still be active—and the tectonic culmination may (or may not) still be actively growing.

Relevant to the question of "is it still growing"—I can't remember whether your manuscript discusses the evidence for reverse-dextral slip on the SW-bounding SAF strand (whatever its name). Do you have fault-plane evidence or other evidence that the SW-bounding fault has generated up-on-the-NE displacement (other than the fact that the landscape is higher to the NE than the SW)? Is it possible that the Indio Hills uplift tilted SW away from a high landscape adjacent to the Indio Hills Fault toward a low landscape to the SW? In other words: has uplift on both NE- and SW-bounding master faults been equal? I think this is an important question to address.

Recommendation:

• The manuscript needs to expand and clarify questions about the what, where, and when aspects of tectonic culmination of the Indio Hills. I recommend a new section, or at least addressing these questions in a single part of the narrative.

Comment 9: General comment #9: Character of folds as they approach the Banning Fault— In lines 336 and 508 (and elsewhere) you discuss the geometry of folds as they "approach" the Banning Fault. But these folds presumably are older than a few hundred thousand years, and their axes never reach the position of the "Banning Fault" as you depict it in figures 7a and 7b (and folds closest to the "modern" Banning Fault" in figure 7c area fault-parallel and are not relevant to folds that are oblique to the master faults. Therefore, for the latter, how can you comment on the structural style, morphology, and configuration of the fold sets depicted in figs 7a and 7b? They do not reach the queried and discontinuous "Banning Fault" trace that you shoe more valleyward in figures 7a and 7b. Please clarify.

Comment 10: General comment #10: Update references—Lines 140-150 need to cite Gold and others (2015), and do a more thorough job of describing what Keller and others (1982). At the end of this memo I include many references that you should at least consider for inclusion and evaluation for your report.

Comment 11: Lines 37–38: You absolutely need to have a section that discusses more thoroughly the distribution and nomenclature of major faults in the greater Coachella Valley/Mojave Desert/Peninsular Ranges region. Your readers for the most part are going to need such a regional summary----and the few lines here don't do the job.

Comment 12: Line 39: Careful with this statement. In the literature, the name "Banning Fault" typically is applied from San Gorgonio Pass SE to its junction with the Mission Creek Fault. The literature is mixed about what name to apply to the singular dextral fault SE of this juncture various names are applied to the singular fault, but no author (to my knowledge) uses the name "Banning Fault" here

Comment 13: Lines 39–40: No. I know of no investigator who applies the name "Banning Fault" to a structure in the Mecca Hills and Durmid Hills. Cite source (other than Janecke----see my next comment.

Comment 14: Line 40: No, Janecke (2018) uses the name "San Andreas Fault main trace". She does not apply the name Banning Fault" to any structure in the Durmid Hills. In a couple of places she refers to "the Banning fault, the most active strand of the mSAF nearby"----by which she means that "nearby to the Durmid Hills the Banning Fault is the main SAF trace." In no way was she extending the name "Banning Fault" to the Durmid Hills.

Comment 15: Line 41: No. Allen (1957, fig. 1) does not apply any name to the SAF southeast of its juncture with the Mission Creek fault.

Comment 16: Line 42: This statement is a hypothesis, not an axiom. I comment on this later in the manuscript. Also, careful with the names, as my comments on lines 38-42 indicate.

Comment 17: Lines 46–47: You really need to refer to this as a POSSIBILITY----not as a proved and documented fact.

Comment 18: Lines 56–58: Actually, this is not accurate. The "Coachella Valley segment of the SAFZ" is "expressed by multiple dextral fault strands of the zone----the uplifts themselves (and

their inverted sedimentary fills) are a product of the dextral-oblique transpressional structural evolution you propose in this report. Modify.

Comment 19: Line 71: Palm Spring Formation (singular "Spring"

Comment 20: Lines 79–80: Useful information. However, given that the Indio Hills Fault intervenes between the leuco-granitic rocks and the majority of the Indio Hills stratigraphy, a comment here is needed to the effect that you interpret the leucocratic rocks as "basement" for the inverted Indio Hills basin. Or not, eh?

Comment 21: Line 84: Are the granitic clasts similar to the "disconnected" leuco-granites? Comment 22: Lines 93–100: The narrative in lines 93-100 is a little hard to follow, and the implications are not completely clear. I recommend *first* discussing the stratigraphy and lithology and member relations of the PS Formation in the Indio Hills, then compare and contrast with the PS Formation in the Mecca Hills. Intermingling observations in the two areas makes it difficult for the reader to follow.

Comment 23: Line 98: In contrast, in the Indio Hills the nature.....

Comment 24: Lines 94–112: Upon reading lines 94-112, I see that they incorporate two subjects: (1) the physical stratigraphy of the Palm Spring Formation in the Indio Hills in comparison with that in the Mecca Hills, and (2) evidence for the age of the PS Formation in the Mecca Hills-----with implications for the formation's age in the Indio Hills. Given the importance of linkage between stratigraphy, unconformities, and unit ages, I would rewrite lines 89-112 to more clearly separate (1) stratigraphy and unconformities (or not) in the two areas, and (2) age relations for the PS Formation based on Mecca Hills data with extrapolation to Indio Hills. The narrative should emphasize Indio Hills data, with contrasts and similarities with Mecca Hills.

Comment 25: Lines 104–106: This applies to the Mecca Hills, not your study area----another example of intermixing content about the two areas. Moreover, these are interpretive statements that follow from the geochronologic data----not indicators of age (see my previous two comments). I would place this at the end of the paragraph as a conclusion derived from lines 93-112.

Comment 26: Line 107: In the Mecca Hills only? Or also in the Indio Hills? Add this for specificity, OK?

Comment 27: Line 114: After I read through this section, in my opinion the heading should be "Tectonic culminations" similarities and differences regionally"

Comment 28: Lines 119–120: To be fair, Matti and others (1985, 1992; Matti and Morton, 1993) proposed the left step at San Gorgonio Pass, Dair and Cooke (2009) simply used the USGS geology. See references in comment for line 128

Comment 29: Line 120: You need to define what the term "proto-SAFZ" means. You have not used it previously in the manuscript, and its introduction out of the blue catches the reader by surprise. Also, if you are referring to the old-school use of "proto-SAF" as an "early SAF", that term largely has fallen out of use. Please provide meaning and literature source of usage. Comment 30: Line 121: Meaning of "low-topographic relief SAFZ segment" is unclear. What segment?

Comment 31: Line 121: What? are these

Comment 32: Line 122: uplifted San Bernardino "Mountains" (yes?)

Comment 33: Line 120–124: Lines 120-124 apparently are intended to provide regional structural relations surrounding the Indio Hills. As such, the narrative both (1) has some unclear structural statements and (2) some errors. First, "proto-SAF" needs to be explained. Second, the "low-relief SAFZ segment" is completely unclear, and is not identified on figures 1 and 2. Third, the left-slip faults are not identified: are these the sinistral faults of the eastern Transverse Ranges? If so, cite Powell (1993) as a useful reference for their description and interpretation. Fourth, I don't understand how the "left-slip splay faults" (unspecified) merge into the San Jacinto Fault strands. Fifth, inclusion of the West Salton Detachment in the same single sentence that begins with "north of the Coachella Valley" implies that the WSD is north of the study area and not southwest.

Comment 34: Line 128: Matti and others (1992) is more accessible to readers. Also see Matti and Morton, 1993).

Matti, J.C., Morton, D.M. and Cox, B.F., 1992, The San Andreas fault system in the vicinity of the central Transverse Ranges province, southern California: U.S. Geological Survey Open-File Report 92-354, 40 p., scale 1:250,000. https://pubs.er.usgs.gov/publication/ofr92354 Matti, J.C., and Morton, D.M., 1993, Paleogeographic evolution of the San Andreas fault in southern California: a reconstruction based on a new cross-fault correlation, in Powell, R.E., Weldon, R.J., and Matti, J.C., eds., The San Andreas fault system: displacement, palinspastic reconstruction, and geologic evolution: Geological Society of America Memoir 178, p. 107-159.
Comment 35: Line 129: Rarely is the gouge actually exposed where the fault "line" traverses the desert. Better to use "trace" in place of "main fault gouge"

Comment 36: Line 131: There is something wrong with this reference. In your references cited you identify Parrish (1983) as:

"Parrish, J.G., Geological compilation of Quaternary surficial deposits in southern California, Palm Springs 30' x 60' quadrangle, CGS Special Report 217, Plate 24, 1983"

I don't know what the 1983 report is that you actually want to cite (other than the Palm Springs sheet) or where you got "1983" from, but the "J.G. Parrish" that you cite actually was the California State Geologist when the Palm Springs 30' x 60' sheet (surficial geology) was published by CGS in 2012. The CGS Special report 217 that you refer to is the umbrella that covers all of the individual 30' x 60' surficial maps, including the Palm Springs sheet. I provide a correct bibliographic citation for the PS sheet in your references cited section, and in my general comments memo.

Comment 37: Line 131: Assuming you want to refer to the 2012 Palm Springs sheet (yes?), throughout the manuscript you should search and replace "Parrish, 1983" with "Lancaster et al, 2012".

Comment 38: Line 137: "Southeast" of what? The LSBM? Your study area? clarify Comment 39: Line 143: Need to cite Gold and others (2015) and summarize what they contributed to uplift of the NW Indio Hills.

Comment 40: Line 145: As I recall, Blisniuk et al 2021 explore local uplift as a driver for channel avulsion, but I don't see a discussion of regional uplift as a driver for the Indio Hills as a "tectonic culmination". Eliminate reference, or clarify.

Comment 41: Line 146: Antecedent unclear. Refers to Keller and others, 1982? See my proposed edit.

Comment 42: Lines 159–160: How does your "steep SAF" compare to the seismic-profiling implications of a moderately east-dipping SAF (Fuis and others that you cite in your regional studies section)

Comment 43: Line 171: Is the "elongate ridge" aligned with the Brawley Seismic Zone, or is the SAF itself? Moreover, is "aligned" the correct term, or should it be "projects toward"? Comment 44: Line 176: It is worth citing Janecke et al 2018 again because their study is so detailed and exhaustive, and is the most recent? Comment 45: Lines 194–195: Is this based on field investigation or imagery analysis>

Comment 46: Line 234: "When" is a time term.

Comment 47: Lines 239: It is a little unclear why you use the phrase "SAFZ-parallel" anticline here instead of "Indio Hills-parallel anticline". Explain

Comment 48: Line 242: Antecedent unclear. What does the plural pronoun "they" refer to previously in the sentence?

Comment 49: Lines 253–254: Does this imply detachment between strongly and weakly folded sedimentary materials?

Comment 50: Line 285: Not needed, unless you specify applicable ambiguities

Comment 51: Line 294: grammar unclear (do you mean subparallel?)

Comment 52: Line 303: Antecedent unclear: what does pronoun "its" refer back to? If the "couple of major folds", then pronoun needs to be "their".

Comment 53: Line 310: Are these "brittle faults" the same as the "meter-wide" fold-related

faults? Or a different category? The adjective "brittle" at the beginning of this sentence is puzzling

Comment 54: Line 322: If this is your criterion for fault dip (reverse, thrust), then in figure 2 why don't you extend the thrust barbs NW along the entire Indio Hills?

Comment 55: Line 325: You don't need this, as nearly all workers view the two faults as merging to the SE.

Comment 56: Line 328: Do you know whether these features formed due to movement on the Indio Hills Fault versus some older unrelated deformation that affected crystalline rocks on this part of the greater Little San Bernardino Mountain front?

Comment 57: Line 331: Have you actually observed the fault plane of the Banning? If not, I would begin the paragraph by stating "Like the Indio Hills Fault, fault-plane dip and strike of the Banning Fault must be inferred indirectly".

Comment 58: Lines 331–332: Couldn't this truncation be produced by up on the east dip-slip displacement on the Banning?

Comment 59: Line 337: As with your section describing folds, I would use a different heading format for this fault section, specifically:

Faults in folded sedimentary strata:

Minor strike-slip faults

Reverse faults

Comment 60: Line 338: If you are going to use NW-SE previously in this sentence I would use NE here.

Comment 61: Line 340: By definition, a mudstone consists of silt and clay-sized grains. So if you observed both "mudstones" and "siltstones", list both of them as separate discrete lithologies Comment 62: Line 378: But I thought this section dealt with reverse faults?

Comment 63: Line 386: See my comment above questioning correlation of these structures with late Cenozoic SAF displacements

Comment 64: Lines 396–397: This sentence is difficult to wade through. See my edit for whether it changes your meaning.

Comment 65: Line 400: Show "right-lateral" only once in this sentence, either several words earlier or here, but not twice?

Comment 66: Line 400: Can you provide evidence as to why this is "simple shear" (i.e., it has to be because.....).

Comment 67: Line 408: This has to be the Banning and Indio Hills faults, yes?

Comment 68: Lines 412–414: There is a logic jump here. If I understand your model, the folds tighten and become more complex toward the Indio Hills Fault. Thus, would it not be the fold structural geometry (rather than the deposition of the Palm Spring Formation) that would support "folds propagating outward from the Indio Hills Fault"? The formation deposition (all of the PSF) determines the age of folding, but does it determine fold- propagation style?

Comment 69: Line 418–419: How about structural stilting to the SW in the "hangingwall" of the Indio Hills fault?

Comment 70: Line 424: What is this? Reference to such a thing in the literature? Or is it simply a generic term for convergence in a fault zone?

Comment 71: Line 427: I recommend a paragraph break here.

Comment 72: Line 430: I recommend that you introduce figure 7 at this early point so that readers can better follow your structural logic.

Comment 73: Line 450: label this fold on figure 3C

Comment 74: Line 461: Yes, but doesn't he focus on extensional folds, rather than contractional folds as here in the Indio Hills?

Comment 75: Lines 472–473: Not clear to me how the fault-parallel anticline under discussion could "act" like a SW-dipping thrust fault. Either the structure was a fault or it was not. I would allow the fold's vergence to express its tectonic-transport role.

Comment 76: Line 488: I think you need an introductory paragraph here that sets up the important interpretive synthesis of the section that follows. Line 488 is not a very helpful topic sentence for either its paragraph----you just plunge into assertions about the history without providing any leadup or context. I recommend a paragraph that states something like:

"In this section we use the geometry and kinematics of folds and faults in the southern Indio Hills to reconstruct the story these structures tell us about the geologic and tectonic history not only of the inverted late Cenozoic basin but also about strike-slip and dip-slip faults that bound the basin. Essential elements of this story are (1), (2), (3), and (4)----etc."

Comment 77: Lines 488-489: See comment at the end of line 493

Comment 78: Line 490: As best I can tell, figure 3Sb does not illustrate "normal faults".

Comment 79: Line 491: See my comment on S6

Comment 80: Line 493: It probably is just me, but I don't think you have made a convincing case for the Indio Hills fault originally being a normal fault. If other reviewers agree, then I think you need to do a better job documenting the "normal-fault" interpretation.

Comment 81: Line 497: I may have touched on this in an earlier comment, but I think that you need to build a more thorough case for a "precursory Indio Hills fault". Simply asserting its existence, as you do here, without reminding us (or informing us) that an early history of the Indio Hills fault is required by your structural analysis leaves me asking the question "what is this precursory episode, and why do these guys propose it? What did they tell me earlier in the manuscript?"

Comment 82: Line 497: It only "gradually changed" if you make a better case for a normal-fault origin for the Indio Hills Fault prior to "changing" structural style.

Comment 83: Lines 503–504: Margin of what? The tectonic culmination of the Indio Hills block? The original depositional basin? Clarify

Comment 84: Line 505: Do you mean: "while during this period the Banning Fault was generating significant dextral slip"???? Clarify

Comment 85: Line 506: "Slip event" generally is viewed as an earthquake event in a moment of time. Better here to use ""recent slip style"

Comment 86: Line 509: This must be "off-fault deformation", yes?

Comment 87: Line 510: Why "early"? Wouldn't off-fault deformation (your "distributed stress") have continued throughout the entire history of dextral slip on the Banning Fault? (both early stage and late stage?)

Comment 88: Line 511: "post-dating slip on the Indio Hills Fault", yes? Otherwise, late-stage has no contextual meaning. "Late" relative to what? Geologic time? (i.e., recency in terms of the geologic time scale).

Comment 89: Line 513: In line 511 you already indicated the footprint of fold-fault interaction. Comment 90: Lines 514–515: Why is the occurrence of "minor fault-related folds" on steeply north-dipping beds important? How about moderately-dipping or shallowly-dipping beds? Do minor folds not occur on these strata? Or if they do, are those occurrences irrelevant to your story? The caveat of "steeply dipping" catches my eye and causes me to stumble.

Comment 91: Line 516: Supplement S6 doesn't appear to address restoration of "steep dips" to "horizontal" orientation. I must be missing something.

Comment 92: Lines 516–521: The giant run-on sentence in lines 516-521 needs to be reshaped into more than one sentence.

Comment 93: Line 522–523: This is confusing: line 517 indicates that you cannot discriminate whether "minor folds and faults" "pre-date (or were coeval with) the macro-folding event. But here in lines 522-523 you indicate that "minor right-slip faults evolved synchronously" with the en echelon fold limbs. Clarify. In line 517 maybe you mean "all minor faults other than right-slip faults"?????

Comment 94: Line 531: Line 529 begins with "out-of-the syncline contractional faults, yet here you indicate a specific "anticline". Text is hard to follow.

Also, as indicated in a comment elsewhere, label "the related anticline" in figure 3c.

Comment 95: Line 537: What are these? You have not discussed these elsewhere in the manuscript, and figure 4 caption does not refer to them.

Comment 96: Line 539: You absolutely have to choose between these two different scenarios---state what you conclude, not just what two contrasting choices exist without telling us what you favor.

Comment 97: Line 546: This section needs an introductory paragraph indicating what you will do with the section. I recommend something like:

"In this section we will use detailed structural analysis of folds and faults in the southern Indio Hills to (1) outline the structural history of the tectonic culmination itself, (2) compare and contrast this structural evolution with that of nearby culmination (Mecca Hills, Durmid Hills), and (3) integrate these local structural histories into a structural synthesis for the southern San Andreas Fault zone from X my to Y my. Finally, we evaluate this proposed structural synthesis in terms of what is known about strain budgets within the southern San Andreas Fault system writ large."

Something like this is needed in order to allow the reader to follow how you stitch together all the disparate structural details you have documented so far in the report.

It really would help if you began this section with a bullet outline----or even better, a table---outlining the sequence of depositional/tectonic events together with their age----especially for the complicated and sequential structural evolution of the Indio Hills Fault.

Comment 98: Line 549: I don't understand this phrase: If "inversion" is defined to be destruction and tectonic uplift of the depositional basin----and if the basinal episode involved accumulation of the Miocene Mecca Formation and the Pliocene-Pleistocene Palm Spring Formation----then how could "inversion" occur in Miocene time as line 549 asserts?

Comment 99: Lines 549–550: See my earlier comments about the Indio Hills fault having an early history of dip-slip displacement.

Comment 100: Line 552: This statement is puzzling: *First*, If the granitic basement is exposed only on the east side of the Indio Hills Fault (your figure 2), *second*, if that fault intervenes between the granitic rocks and the Indio Hills and the inverted sedimentary basin of the southern Indio Hills, and *third* if the Indio Hills Fault has a significant normal-slip, dextral-slip, and thrustslip history, then how and why do you include erosion of the isolated granitic rocks and overlap by the Miocene Mecca Formation in your discussion of the structural history of the tectonic culmination? The depositional overlap you refer to occurred *beyond* (outside) the depositional basin that subsequently was "inverted", yes? Clarify

Comment 101: Line 552: Did you present evidence for this statement earlier in the manuscript? I don't remember reading this. You need to present evidence for this age range.

Comment 102: Line 558: Which strand of the SAF? The Indio Hills Fault? The nascent Banning Fault? Clarify

Comment 103: Line 559: Why do you say "probably"? Either the fold axes trended at a high angle to the bounding faults or they didn't. Fix.

Comment 104: Line 559: You refer to "bounding master faults". But you are discussing ONLY the Indio Hills Fault. What is the *other* "bounding fault"????? In figure 7 you do not identify a coeval bounding fault (the Banning Fault is questionably identified, but in line 511 you identify the Banning as a "late stage" structure. You need to clarify this issue of "bounding faults" (plural).

Comment 105: Line 560: See my comment on line 400

Comment 106: Line 562: Precursory to what? Precursory to slip on the Indio Hills Fault? Clarify. Comment 107: Liens 564–565: You can't just identify two very different structural styles, without favoring one or the other. Choose one style and discount the other, then explain and justify your choices.

Comment 108: Line 566: Again: what is your choice? There is a significant difference between minor-fold deformation"prior to or together with" macro-fold development, eh?

Comment 109: Line 567: But your statement in line 566 indicates that you don't know when the minor folds formed, hence how can you use these structures to assign "minor partitioning"? And I presume you mean "strain partitioning"?

Comment 110: Line 568: "Partly partitioned" shortening already is going on based on line 566, so in line 568 how can there be a *gradual change to* "partly partitioned" shortening?

Comment 111: Line 570: What is the nature of the "oblique-slip"? Is there a dextral-slip component? Do you ever actually document dextral slip on the Indio Hills Fault? Please clarify. Comment 112: Line 571: By this do you mean that----prior to this time----the Banning was NOT a major player, although it would become so later in the structural evolution of the culmination? Comment 113: Line 572: By this do you mean that formerly "open" folds became "more tightly closed" as they were squeezed by "increased shear folding"?

Comment 114: Line 573: What is this fold style? You have not discussed it previously in your structural analysis.

Comment 115: Line 578: Because youthful dextral slip on the later truncated culmination folds and displaced their truncated counterparts NW away from the culmination----yes? It is important to inform the reader that you actually have no idea about the structural configuration of the macro folds SW of where you actually observe them. Comment 116: Lines 576–578: I do not understand how the interpretation in lines 579-581 follows from lines 576-578

Comment 117: Line 582: Age of "late stage" needs to be specified. In other words, if basin inversion and tectonic uplift occurred in stages, you need to specify the age ranges of terms like "early-stage" and "late-stage".

Comment 118: Line 582: Does this mean "spatially involved"? "kinematically evolved"? Clarify Comment 119: Line 587: label this structure on figures 3c and 7c.

Comment 120: Line 588: margin-parallel to what? The tectonic culmination? Clarify

Comment 121: Lines 592–593: Not clear to me what the interpretive boundary is between when "although overlapping and synchronous formation....may have occurred" (that is, if some unspecified percentage of structures may have overlapped timing wise, when does that percentage become a deal-breaker for your structural paradigm?)

Comment 122: Line 594: Antecedent unclear: what is the "latter"?

Comment 123: Line 594: Does this mean "we don't have enough field data to confirm X, Y, and Z"? Or does it mean "we have not observed a sufficient number of specific fold and fault types to confirm "progressive evolution from distributed to partly distributed deformation" (see line 591)? Please clarify

Comment 124: Line 602: See my comments earlier in this report. In the Little San Bernardino Mountains this fault zone does not exist at the surface. Revise and explain

Comment 125: Line 606: No. You can only relate recent earthquakes in the vicinity of the "Landers-Mojave Line" in the ECSZ if you can show that the Indio Hills Fault is active as an earthquake generator. You have not discussed this possibility, and no researcher to my knowledge has shown that the Indio Hills Fault is an active player.

Comment 126: Line 609: To compare alleged active faulting on the Indio Hills Fault with farflung faults like the Calico and Camp Rock is inappropriate. Better to compare with nearby faults investigated by Rymer (2000) in the Little San Bernardino Mountains, or with the 1992 Landers and Joshua Tree events themselves. Revise.

Comment 127: Line 609: See my comment earlier about regional extent of the term "Banning Fault".

Comment 128: Line 610: You absolutely need to explain what this is. You can't just point us to Platt and Passchier and find out for ourselves.

By the way, reference to P&P (2016) implies that those two authors investigated the SAFZ southeast of the Indio Hills. Did they? I am not going to go look. You need to assert that you are adopting the dextral freeway junction (sensu P&P, 2016) who invented the term based on work elsewhere.

Comment 129: Line 610: Somthing wrong with this grammar (phrase)

Comment 130: Line 611: What does (late) mean? In comparison with "early"? You really need a table or figure that shows the age and movement history of the Banning and Indio Hills Faults. See my previous comment.

Comment 131: Lines 615–616: Have you stated this age and discussed it earlier in your manuscript? I cannot find such a discussion. You need to document ages for the Banning and Indio Hills faults, with basis for age calls. Given the complexity of fault relations, the diverse tectonic interpretations in the literature, and uncertainty about fault names throughout the region, you absolutely need to clarify and document age relations for specific faults.

Comment 132: Line 625: Well, almost certainly the genesis of the Indio Hills tectonic culmination (resulting in basin inversion) was gradual and progressive. Is it better to state that you have documented what is intuitive to most investigators in this region? Compare with Gold and others (2015) and Blisniuk and others (2021). Do they elaborate on progressive uplift and exhumation?

Comment 133: Lines 633–634: I don't know what you mean by "the SAFZ main strand". According to previous statements in this manuscript, the Banning Fault is the "main strand" SE of the Indio Hills (but see my comment to that effect earlier in this manuscript). So what does the Banning Fault merge into as it projects southeastward? See how Behr and others (2010) deal with this using an "Indio strand".

Comment 134: Line 639: Antecedent unclear. What does "them" refer to?

Comment 135: Lines 639–640: Do you mean "Indio Hills" footprint? Or "Indio Hills Fault"?. Clarify. If you refer to the fault, then how do you know it has thin versus thick gouge if you can't see the fault in any outcrops? (Everywhere concealed by unconsolidated surficial deposits). Comment 136: Lines 641–642: Also, the relatively massively structured Mecca Formation intervenes between the Indio Hills Fault and the highly deformed Palm Spring Formation. Would that make a difference? Comment 137: Liens 645–646: This is new information for me. McNabb and others (2017) do not refer to this (contrary to your statement in line 648), nor do Fattaruso and others (2014, 2016). For benefit to readers like myself, it would be helpful if you were to review the proposals for----and basis for----large normal-slip faults bounding the east margins of Miocene-Pliocene sedimentary basins that evolved in the Indio-Mecca Hills footprint of the ancient Salton Trough. I have read the summary on p. 4 of Bergh and others (2019), and I see that some workers posit that basin fills like those in the Mecca and Indio Hills accumulated in a Miocene-Pliocene depocenter formed (in the hangingwall?) adjacent to a large Basin-and-Range style "normal fault". Is this structure documented, or is model-dependent? Does this large "normal fault" crop out anywhere, or has it been so transformed by younger activity that its normal-slip origin can only be inferred? A better summary by you would be helpful to regional geologists like myself. Comment 138: Line 648: In McNabb et al I do not see any reference to a normal fault on the NE side of the Mecca Hills, that later was reactivated as an oblique reverse fault

Comment 139: Line 649: I see on p. 5 of Bergh and others (2019) that Sylvester and company view the Painted Canyon Fault as an "oblique reverse strike-slip fault"; but might this fault originally have had a normal-slip origin. Please help us here.

Comment 140: Lines 699–700: Do you actually observe the width of the Banning Fault damage zone in the southern Indio Hills? I don't remember you discussing this in your manuscript. Where you might observe the "damage zone" isn't it typically obscured by young surficial deposits and wind-blown sand?

Comment 141: Line 725: I don't remember you defining this or referring to it elsewhere in the manuscript. What is the distribution of this structure in figure 1? Is this the same as the Banning Fault + Indio Hills Fault? Most workers refer to the combined traces of the SAFZ in the Coachella Valley as the "Coachella Valley segment of the SA" following Matti and others, 1992 (see

Comment 142: Line 731–732: If the Indio Hills and Banning faults are not coeval, then what was the southwest bounding fault that worked together with the Indio Hills Fault to produce the tectonic culmination of the Indio Hills?

Comment 143: Line 738: Your conclusions should emphasize that the Indio Hills Fault has a polyphase history: (1) It started it life as a Miocene normal fault according to your interpretation; (2) it evolved into a reverse/thrust fault that initiated inversion of the Pliocene sedimentary basin;

(3) it now (?) is a dextral strand of the SAFZ according to conclusion (4). This polyphase history should go near the top of the conclusions section in my opinion.

Comment 144: Lines 738–739: As I repeat throughout the manuscript, this is a hypothesis only. You must state it as such.

Comment 145: Line 881: Insert "Lancaster and coauthors, 2012", here in place of Parrish, 1983 as discussed below and in my memo

Comment 146: Line 928: This reference is incorrect (I should know, as I should have been lead author). The actual citation should be:

Lancaster, J.T., Hayhurst, C.A., and Bedrossian, T.L., 2012, Preliminary geologic map of Quaternary surficial deposits in southern California: Palm Springs 30' x 60' quadrangle, in

Bedrossian, T.L., Roffers, P., Hayhurst, C.A., Lancaster, J.T., and Short, W.R., Geologic

compilation of Quaternary surficial deposits in southern California December 2012

https://www.conservation.ca.gov/cgs/fwgp/Pages/sr217.aspx#palmsprings.

Comment 147: Line 930: You may want to cite Passchier and Platt (2016) in the Journal of Structural Geology in addition to Platt and Passchier

Comment 148: Line 978: Most workers would apply the name "Mission Creek Fault" to what you term the "Banning Fault". The point is: which fault is the dominant throughgoing structure at the latitude of your study area?

Comment 149: Line 1002: On this figure, it would be useful to show the geologic contact between the Mecca Formation and the Palm Spring Formation.

Comment 150: Line 1002: Label this fold so that readers of lines 586-587 can identify it and associate it with your kinematic model.

Comment 151: Line 1002: You apparently refer to this "fault-parallel anticline in lines 450 and 454. If so, I would label it here.

Comment 152: Line 1041: What is this time interval, and how long was it? Episode in Pliocene? Pleistocene? Clarify

Comment 153: Line 1055: Literature source for extending the Garnet Hill strand this far SE

Comment 154: Line 1062: cite reference (Janecke et al, 2018)????

Comment 155: Line 1063: Singular "Hill"

Comment 156: Supplment S6: Not clear from the photographs that these are "normal" faults. Please explain how you make this interpretation.

**2. Author's reply**

Comment 1: agreed, we trying to reach both specialists in structural geology and regional geologists. See also response to comments 33, 47, 128 and 147. However, the term "ladder" was defined upon first occurrence in the text lines 272–274: "as defined by Davis (1999) and Schulz and Balasko (2003), where overlapping, E–W- to NW–SE-striking step-over faults rotated along multiple connecting cross faults".

Comment 2: agreed.

Comment 3: agreed.

Comment 4: agreed.

Comment 5: we agree that reference to Dokka et al. (1990a, 1990b) to support connection of the Eastern California Shear Zone and Landers Mojave Line with the Indio Hills fault is not enough. Therefore we add reference to Nur et al. (1993a, 1993b) who connect the 1992 Joshua Tree earthquake with the series of earthquakes that occurred along the Landers Mojave Line between 1947–1992 (see their figure 1). We also agree that the manuscript should clarify that it is unlikely that transfer of strain between the ECSZ and the SAFZ occurs along a single fault trace. However, we do not understand the comment of the reviewer claiming that the epicenter of the 1992 Joshua Tree earthquake is located a few km north of the West Deception Canyon fault. Looking at Rymer (2000, his figures 1 and 2), it is clear that the earthquake's epicenter is located exactly along the trace of the West Deception Canyon fault. Rymer (2000) even mentions that although he observed rupture along the East Wide Canon fault, that it is clear from the location of the epicenter (east of this fault) and from this fault's western dip and very small amount of rupture-related offset (ca. 6 mm) that it cannot have triggered the 1992 Joshua Tree earthquake. Rymer (1992) further specifies that structural fieldwork and interpretation of aerial images suggest that the West Deception Canyon fault is connected to the Eureka Peak fault, which moved during the June 1992 earthquake. Comment 6: agreed.

Comment 7: the "grammatical structures" mentioned in the comment are the way chosen to express uncertainty. In some cases, the data allow us to "favor" one or the other possibility (e.g., lines 473, 559, and 691), whereas in other cases they do not. In the instances the data do not allow us to favor one alternative or the other(s), it would be inappropriate and unethical to arbitrarily choose any of the alternatives. We prefer to remain open and admit that we do not "know" everything for certain, leaving ample room to future workers in the area to further validate, reject, or rework the proposed model.

Comment 8: agreed. See response to comments 5, 12, and 125. Uplift along the Indio Hills fault seems to have been superior to uplift along the main San Andreas fault as suggested by the intensely folded geometries of sedimentary strata and higher topographic relief in the vicinity of the Indio Hills fault. This needs to be mentioned in the manuscript. However, the present manuscript does not address present-day growth or not of the Indio Hills.

Comment 9: agreed.

Comment 10: agreed. Also see response to comments 39 and 40.

Comment 11: agreed. See response to comment 2.

Comment 12: agreed. Instead of calling the main splay of the San Andreas fault the "Banning Fault", we change it to the "main San Andreas fault" (mSAF), in accordance to the recent study by Janecke et al. (2018) in the area.

Comment 13: agreed. See response to comment 12.

Comment 14: agreed. See response to comment 12.

Comment 15: agreed. See response to comment 12.

Comment 16: agreed. Also see response to comment 12.

Comment 17: agreed.

Comment 18: agreed.

Comment 19: agreed.

Comment 20: agreed.

Comment 21: agreed. See response to comment 20.

Comment 22: agreed.

Comment 23: see response to comment 22.

Comment 24: agreed.

Comment 25: agreed. Also see response to comment 6.

Comment 26: agreed.

Comment 27: partly agreed.

Comment 28: agreed.

Comment 29: agreed.

Comment 30: agreed.

Comment 31: agreed. See response to comment 30.

Comment 32: agreed. See response to comment 30.

Comment 33: agreed. See response to comment 30.

Comment 34: agreed. See response to comment 28.

Comment 35: agreed.

Comment 36: agreed.

Comment 37: see response to comment 36.

Comment 38: agreed.

Comment 39: agreed.

Comment 40: agreed.

Comment 41: agreed.

Comment 42: the study of Fuis et al. (2017) suggests that the SAFZ dips steeply at shallow depth both the in Indio Hills and Mecca Hils, but more moderate dips are probable at higher depth.

Comment 43: agreed.

Comment 44: agreed.

Comment 45: agreed. The study is based on both field study and imagery analysis.

Comment 46: agreed.

Comment 47: the term "Indio Hills-parallel" does not bear any structural meaning, whereas "SAFZ-parallel" suggests that the anticline is the product of dip-slip to oblique-slip thrusting rather than strike-slip fault movements as inferred for the SAFZ-oblique macro-folds.

Comment 48: agreed.

Comment 49: this does not necessarily imply detachment between strongly and weakly folded strata, but local detachment/décollement may occur as suggested by the next sentence (see also possible detachment folds in figure 4 in the present manuscript). We concede that these local detachment folds and décollements should be discussed, but not in the Result section since it is too interpretative. We instead mention this interpretation in the Discussion section lines 583–584 and 613.

Comment 50: agreed.

Comment 51: agreed.

Comment 52: agreed.

Comment 53: agreed.

Comment 54: agreed.

Comment 55: agreed.

Comment 56: the fractures in the granitic basement have not been dated and may therefore be older than the deformation of sedimentary strata in the Indio Hills. However, the strike of brittle fault sets in the granite basement matches that of fractures in sedimentary strata (see stereoplots in Figure 2), which suggests that the fractures formed due to similarly oriented stress. The authors of the present manuscript suggest to mention this explanation in the manuscript if also judged convenient by the referee.

Comment 57: agreed.

Comment 58: yes indeed, the truncation of the vertical folds and *en echelon* SAF-oblique folds may result from dip-slip reverse movement along the main San Andreas fault. However, no direct evidence was encountered to support reverse dip-slip movement along this fault, and the geometry of vertical (shear) folds in the direct vicinity of the fault (Figure 5) and the anticlockwise rotation of the axis of the three *en echelon* macro-folds towards the fault (Figure 2) suggest a significant component of lateral movement along the main San Andreas fault at some point.

Comment 59: agreed. However, this would require an additional level of sub-title/sub-section, which is not allowed by the journal's standards.

Comment 60: disagreed because of the journal's standard.

Comment 61: agreed.

Comment 62: agreed.

Comment 63: see response to comment 56.

Comment 64: agreed.

Comment 65: agreed.

- Comment 66: agreed.
- Comment 67: agreed.
- Comment 68: agreed.
- Comment 69: agreed.
- Comment 70: agreed.
- Comment 71: agreed.
- Comment 72: agreed.
- Comment 73: agreed.

Comment 74: agreed. Comment 75: agreed. Comment 76: agreed. Comment 77: see response to comment 80. Comment 78: agreed. Comment 79: see response to comments 80 and 156. Comment 80: agreed. Comment 81: see response to comment 80. Comment 82: agreed. Comment 83: agreed. Comment 83: agreed. Comment 84: agreed. Comment 85: partly agreed. Comment 85: partly agreed. Comment 86: yes.

Comment 87: the truncating relationship of the main San Andreas fault and the en echelon macrofolds in the study area suggest that deformation was distributed at first, but more localized during the activation of the main San Andreas fault in the late stage of deformation since no major reworkeing of the macro-folds is observed along this fault (apart from some shear folding near the fault; see figure 5 in the present manuscript).

Comment 88: agreed. The term "late" was meant as "late for the study area". However, we concede that its meaning is not appropriate since there is probably ongoing slip along the Mission Creek and main San Andreas faults at present (Gold et al., 2015), and possibly along the Indio Hills fault if it is part of the Landers Mojave line (e.g., Nur et al., 1993a, 1993b). Clarification of the meaning is therefore needed in the sentence.

Comment 89: agreed. However, this paragraph analyzes the fold-fault relationships for a different purpose, i.e., to show that strain partitioning already occurred in the early phase of deformation of the area despite the occurrence of distributed deformation (e.g., SAFZ-oblique macro-folds).

Comment 90: we want to draw the attention of the reader on the fact that, in their current position, it is not possible to explain the formation of the observed small folds and faults with simple transpression and/or contraction. The unorthodox geometries of the observed folds and faults is however simply due to the steep dip of the strata in which they were observed. If restoring the strata

to horizontal (i.e., prior to deformation), the small folds and faults turn into structures with geometries typical of contractional folds and thrusts.

Comment 91: agreed. The sentence is not clear. Supplement S6 addresses restoration of the steep of the sedimentary strata to horizontal (i.e., prior to deformation).

Comment 92: agreed.

Comment 93: precisely, agreed.

Comment 94: agreed.

Comment 95: agreed.

Comment 96: if the reviewer's comment is meant for distributed versus partitioned deformation, we specify that we favor synchronous distributed and partitioned deformation. If the comment is targeting the progressive versus synchronous bit, the end-member we favor is specified line 677. Comment 97: agreed.

Comment 98: completely agreed.

Comment 99: agreed. See response to comment 80.

Comment 100: agreed. This is a mix up from the multiple re-writing phases of the manuscript. The meaning of the sentence needs to be adjusted. The Mecca Formation is not found in the footwall of the fault (e.g., map by Lancaster et al., 2012).

Comment 101: this statement is based on the previous study by McNabb et al. (2017) in the Mecca Hills, not on the present study.

Comment 102: agreed.

Comment 103: agreed.

Comment 104: agreed. See response to comment 102.

Comment 105: agreed.

Comment 106: agreed.

Comment 107: disagreed. The present dataset and observations do not allow to favor one or the other. It is therefore important to mention both possibilities to account for uncertainties, which are a very important part of the research itself from a science ethics perspective.

Comment 108: disagreed. There is not much difference between "prior to" and "together with" in the present scenario/model, and, again, the dataset does not allow to favor one or the other alternative.

Comment 109: yes, we meant "strain partitioning". It is true that the minor folds may have formed either prior to or together with the *en echelon* macro-folds. However, this is irrelevant with regards to the present sentence because we argued that distributed deformation occurred in the early stages of deformation during formation of the macro-folds. Thus, if the minor folds indicate minor strain partitioning, it suggests that both distributed and partitioned deformation occurred at the same time. Comment 110: agreed.

Comment 111: agreed. We connect SAFZ-oblique *en echelon* macro-folds in the study area to right-lateral slip along the Indio Hills fault.

Comment 112: absolutely. Agreed that the sentence need rewording rto better reflect its intended meaning.

Comment 113: yes, agreed.

Comment 114: agreed, the term is confusing.

Comment 115: the macro-folds were not observed southwest of the main San Andreas fault. But it is irrelevant to mention this in the present section. We agree, however, to mention it in the result section where it is more appropriate.

Comment 116: agreed. It does not. The sentence needs rewording.

Comment 117: agreed.

Comment 118: agreed. We mean kinematically evolved.

Comment 119: it is probably not wise to overcrowd the figures with excessive labelling.

However, we agree that we should be more specific to help the reader follow.

Comment 120: agreed.

Comment 121: see response to comment 105.

Comment 122: agreed.

Comment 123: see response to comment 105.

Comment 124: see response to comment 5.

Comment 125: the work of Nur et al. (1993a, 1993b) demonstrates that the Landers Mojave Line is a through-going fault that crosscuts the Pinto Mountain fault. They add that "segments of this fault were identified in the field before 1992 (M. Rymer, personal communication)", but that "it was not recognized as a through-going, coherent and seismogenic fault" then. Rymer had been working with the Indio Hills fault at that time and the location of this fault matches that of the southern segment of the Landers Mojave Line near the Joshua Tree earthquake. We therefore

argue that the Indio Hills fault is a segment of this through-going active fault. Nevertheless, we concede that it is necessary to add up to this paragraph for clarification.

Comment 126: partly agreed. The contribution by Rymer (2000) is appropriate to mention here. Rymer shows that the 1992 Joshua Tree earthquake occurred along the NNW–SSE-striking, west-dipping West Deception Canyon fault in the Little San Bernardino Mountains, and that this fault merges to the south with the Indio Hills fault (see his figure 1). It is therefore relevant and paramount to discuss the role of the Indio Hills fault with regards to the various fault systems it seems to connect, including NNW–SSE-striking faults of the Landers Mojave Line (e.g., West Deception Canyon fault), NW–SE-striking faults of the Eastern California Shear Zone (e.g., Calico and Camprock faults), and the main San Andreas fault.

Comment 127: agreed. See response to comment 12.

Comment 128: agreed.

Comment 129: agreed.

Comment 130: agreed.

Comment 131: agreed. See response to comment 117.

Comment 132: partly agreed. Gold et al. (2015) and Blisniuk et al. (2021) investigate recent (Holocene) slip rates for the Mission Creek and Banning faults, whereas we study older deformation history in the southeastern Indio Hills (Pleistocene). However, we concede that a few sentences are needed in order to compare our findings to those of Keller et al. (1982) in the northwestern Indio Hills.

Comment 133: agreed.

Comment 134: agreed.

Comment 135: partly agreed. However, gouge and anastomosing geometries should probably be observed along minor faults that strike parallel to, show comparable kinematics, and are located in the vicinity of the Indio Hills fault (e.g., S3a). This is not the case.

Comment 136: yes, it could, but the conglomerates of the Mecca Formation are only weakly folded. Comment 137: disagreed. McNabb et al. (2017) inferred "SW-side down slip" along the Painted Canyon fault based on the presence of the Mecca Formation conglomerate in the hanging wall and its absence in the footwall of the fault, and on sedimentary facies (conglomerate fining up upwards; see first paragraph in their discussion pp. 81 and pp., their figs. 15 and 16, and the second paragraph of the conclusion pp. 84). Also see response to comment 80. Comment 138: see response to comment 137.

Comment 139: see response to comment 137.

Comment 140: agreed.

Comment 141: agreed. See response to comment 12.

Comment 142: we did not imply that the main San Andreas and Indio Hills faults were not coeval.

They were coeval at least in the later stages of our model (i.e., in the late Pleistocene). See also response to comment 117.

Comment 143: agreed.

Comment 144: see response to comments 125 and 126.

Comment 145: agreed.

Comment 146: agreed. See response to comment 145.

Comment 147: agreed. See response to comment 128.

Comment 148: agreed.

Comment 149: the transition between the two units is gradual and is not very well constrained. In addition, the transition between the two units is not critical to the present contribution. The reader may have a look at the map by Lancaster et al. (2012) to get an approximative idea of the location of the transition.

Comment 150: agreed. See response to comment 73.

Comment 151: partly agreed. See response to comment 119.

Comment 152: agreed.

Comment 153: agreed.

Comment 154: the wide arrows indicating main shortening direction are not from Janecke et al. (2018), but from the present contribution for the Indio Hills, and from Bergh et al. (2019) for the Mecca Hills.

Comment 155: agreed.

Comment 156: a simple rotation of sedimentary and truncating micro faults by 52 degrees counterclockwise to restore the sedimentary bed to the horizontal (i.e., prior to macro-folding deformation) shows that some micro faults display reverse offsets of bedding surfaces and most likely formed as micro thrusts (see Supplement S6a–b), whereas other micro faults display normal offsets of bedding surfaces and form micro graben structures with associated syn-kinematic growth strata (see Supplement S6c).

**3. Changes implemented**

Comment 1: see also response to comments 33, 47, 128 and 147. Also added "(stretched long limb in an overturned fold)" line 319, "(the shortened, inverted limb indicating the direction of tectonic transport in an overturned fold)" lines 339–340, and "folds involving shearing along a plane that is parallel to subparallel to the fold's axial plane; Groshong, 1975; Meere et al., 2013;" lines 302–304 and Groshong (1975) and Meere et al. (2013) to the reference list.

Comment 2: rewrote lines 40–47 and moved some information to the new section about "Regional faults" as suggested by the referee. Moved first paragraph of the section about Tectonic culminations to the new section about "Regional faults". Wrote section about Regional faults as follows: "*Regional faults*

[revised manuscript text omitted]

Comment 3: see response to comment 12.

Comment 4: see response to comments 80, 117, 130, 131, and 143.

Comment 5: added reference to Nur et al. (1993a, 1993b) line 698. Added "be one of several faults to" lines 808–809. Changed "transfers" into "contributes to transfer" line 966. Comment 6: developed a new summary table with the ages of relevant geological events in the Coachella Valley and nearby and added reference to the table lines 79, 114, 128, 139, 206, . In addition, adjusted ages throughout the manuscript after latest findings as suggested, including "Miocene–Pliocene" into "Pliocene–Pleistocene" lines 13, 61, and 71, "2.2–" into "or later than" line 18, "2.2" into "2.6" line 18, "Miocene–Pliocene" into "mid to upper Pliocene" line 88, "is an angular unconformity that signals" into "is marked by two angular unconformities that signal" line 103, "3.7–2.6" into "3.0–2.3", and "mid–late Pliocene" into "latest Pliocene–early Pleistocene" line 107, "2.8–1.0" into "2.6–0.76" and "late Pliocene" lines 152 and 156 into "Pleistocene", "4.0–3.7" into "3.7–3.0" line 675, "3.7–2.8" into "3.0–2.3" line 676, "2.8–1.0" into "2.6–0.76" line 75, "1" into "3.0–2.3" line 676, "2.8–1.0" into "2.6–0.76" and "late 840.

Also added reference to McNabb et al. (2017) line 112. Added "In contrast to other uplift areas in Coachella Valley, the Ocotillo Formation has not been mapped in the Indio Hills in the present study. However, based on the occurrence of the Bishop Ash at the northwestern edge of the study area and on the occurrence of the volcanic deposit within the uppermost Palm Spring Formation or at the base of the overlying Ocotillo Formation in the Mecca Hills, it is likely that the Ocotillo Formation occurs just northwest of the area mapped (Fig. 2). In addition, it is deposited on the flank northeast of the Indio Hills fault, and southwest of the main San Andreas fault (Figs. 1 and 2), indicating that this unit was either not deposited or eroded in the area that recorded the most uplift in Indio Hills." lines 116-123. Deleted "Mecca Hills and Durmid Hills" lines 124-125 and replaced by "the Coachella Valley". Added ". The volcanic deposit is found within" line 127, "(which is unconformably overlain by the Ocotillo Formation) in the hanging wall of the Painted Canyon fault away from the fault, and within the base of the Ocotillo Formation in the hanging wall of the Painted Canyon fault near the fault (Ocotillo and uppermost Palm Spring formations interfingering near the fault) and in the footwall of the fault" lines 128–132, and ". The unconformable contact between the Palm Spring and Ocotillo formations away from the Painted Canyon fault towards the southwest and their interfingering relationship near the fault suggest that uplift had already initiated prior to deposition of the Ocotillo Formation (i.e., before 0.76 Ma, in the mid Pleistocene), possibly during the formation of the lower unconformity between the lower and upper members of the Palm Spring Formation (McNabb et al., 2017). In addition, the involvement of the Bishop Ash in deformation suggest that deformation continued past 0.76 Ma (in the late Pleistocene)." lines 132–139. Deleted "Janecke et al., 2018" line 132. Deleted "In contrast to other uplift areas in Coachella Valley, the Ocotillo Formation has not been mapped in the Indio Hills, but rather is deposited on the flank northeast of the Indio Hills fault, and southwest of the main San AndreasBanning fault (Figs. 1 and 2), indicating that the Ocotillo Formation was either not deposited, or eroded in the area of uplift." lines 140–143. Added "because of the involvement in folding of the Bishop Ash and of adjacent strata possibly of the Ocotillo Formation (i.e., maximum age of 0.76 Ma – earliest late Pleistocene; Fig. 2 and Table 1)" lines 483–485 and "Should the whole Ocotillo Formation be folded in the Indio Hills, the maximum age constraints could be narrowed to < 0.6-0.5 Ma based on magnetostratigraphic ages for the upper part of the Ocotillo Formation (Kirby et al., 2007)." lines 486-489. Added " in the late Pleistocene" line 538. Added "in the (earliest?) late Pleistocene (Table 1)" lines 546–547. Added "-stage (i.e., late Pleistocene)" lines 563–564. Added "in the mid-Miocene–Pliocene (ca. 15–3.0 Ma)" lines 570–571, "in the (earliest?) late Pleistocene to present-day (< 0.76 Ma)" line 572, and "in the late Pleistocene to present-day (< 0.76 Ma; Table 1)" lines 573–574. Added "in the late Pleistocene" lines 606–607, 615, and 617. Added "in the – earliest? – late Pleistocene" line 618. Added "(see phases 1 and 2 in Table 1)" line 654. Added "(< 0.76 Ma)" line 659. Added "late" line 671. Added "(Miocene?-) Pliocene" line 672. Replaced "they were overlain by" by "strata of" line 674. Added "were deposited in the Pliocene" line 675, "members of the" line 676 and reference to Chang et al. (1987) and Boley et al. (1994) lines 677–678. Deleted "by " and " succeeding," line 675. Added "(earliest?) late" line 679. Added "after the latter was deposited (< 0.76 Ma), i.e., probably in earliest late Pleistocene time (Table 1)" lines 682–683. Deleted "Late-stage" line 711. Added "and phase 3 in Table 1" line 717. Added "(overlapping of phases 1 and 2 in Table 1)" line 727 and "; phase 3 in Table 1" line 728. Added "late" line 736 and "to present-day" lines 736–737. Added "in the late Pleistocene" and "and Table 1" line 761. Added "-Pliocene" line 803. Added "(late)" line 805. Added "(i.e., earliest to mid Pleistocene) with partial and local erosion of the Palm Spring Formation (see lower and upper unconformities in McNabb et al., 2017)" lines 840-842. Added "(see unconformity between the uppermost Palm Spring Formation and base of the Ocotillo Formation southwest of the Painted Canyon fault in

McNabb et al., 2017)" lines 842–844. Added "whole" line 844. Deleted "A comparable time frame and ongoing activity are expected for the Indio Hills." lines 846–847 and replaced by "Fault activity and tectonic uplift of the Mecca Hills therefore most likely initiated earlier (earliest Pleistocene) than in the Indio Hills (earliest late Pleistocene; Table 1), where the transition from the lower to the upper member of the Palm Spring Formation is gradual and does not show any major unconformity." lines 847–850. Added "(ca. 1 Ma – early/mid Pleistocene)" line 870. Added "(i.e., probably in the earliest or middle part of the late Pleistocene)" lines 874–875. Added "mid" line 892. Added "in the late Pleistocene" line 896. Added "The initiation of right-lateral-reverse slip along major SAFZ-parallel faults and the main San Andreas fault in the Coachella Valley is younger towards the northwest (Pliocene in the Durmid Hills, early Pleistocene in the Mecca Hills and late Pleistocene in the Indio Hills). The onset of transpressional uplift, however, appears to be coeval in all tectonic culminations (late to latest) Pleistocene." Lines 918–922.

**Comment 7: none.**

Comment 8: see response to comments 5, 12, and 125. Also adjusted the position of the main San Andreas fault in figure 7a and b to match that in figure 7c. Added "Possibly as a consequence of a longer period of activity, and as suggested by relatively higher topographic relief and more intensely folded geometries of sedimentary strata in the vicinity and along the Indio Hills fault than along the main San Andreas fault, it is probable that the former accommodated significantly larger amounts of uplift than the latter. This implies a southwest-tilted geometry for the Indio Hills culmination." lines 587–591.

Comment 9: deleted "where they approach the main San Andreas fault" lines 456–457, and added this to the Discussion chapter. Deleted "main San Andreas and", added "rather" and "a single", and replaced "faults" by "fault" twice lines 540–541. Deleted "active" and added "more active" line 541. Deleted "(i.e., near the main San Andreas fault)" line 588. Replaced "main San Andreas fault" by "the southwest", replaced "strain" by "off-fault deformation", and added "main San Andreas" lines 681–683. Replaced "with the main San Andreas fault" by "farther southwest" lines 776–777.

Comment 10: added Gold et al. (2015) to the reference list. Added "The study also showed that drainage systems were offset recently (at ca. 0.03-0.02 Ma) and indicate relatively high slip rates along the Mission Creek fault in the order of 23–35 cm.y-1, i.e., comparable to the more recent c.

23 cm.y-1 estimate by Blisniuk et al. (2021)." lines 238–240. Also see response to comments 39 and 40.

Comment 11: see response to comment 2.

Comment 12: replaced "Banning" by "main San Andreas" lines 17, 23, 25, 39, 42, 96, 116, 131, 132, 145, 210, 220, 223, 233, 239, 244, 248, 284, 317, 323, 331, 337, 340, 346, 351, 354, 356, 358, 392, 422, 436, 445, 475, 517, 539, 540, 543, 546, 602, 611, 618, 626, 630, 635, 641, 653, 655, 658, 660, 665, 678, 723, 743, 754, 760, 773, 777, 783, 786, 1043, 1074, 1093, and 1103. Deleted "thought to correspond to the main SAFZ in" line 40. Added ", which merge into the main San Andreas fault" lines 154–155.

Comment 13: see response to comment 12.

Comment 14: see response to comment 12.

Comment 15: see response to comment 12.

Comment 16: replaced "marges" by "may merge" line 41–42. Also see response to comment 12.

Comment 17: added "potential" line 46.

Comment 18: rewrote the sentence lines 56–58 as "The Coachella Valley segment of the SAFZ in southern California is expressed asmultiple, right-lateral fault strands, which uplifted blocks in the Indio Hills, Mecca Hills, and Durmid Hills".

Comment 19: corrected "Palm Springs Formation" into "Palm Spring Formation" lines 73 and 471. Comment 20: deleted "basement" lines 81 and 86, and added "and that at least part of the clasts are from the leuco-granitic rocks, which must correspond to basement rocks of the inverted Indio Hills basin" lines 88–90.

Comment 21: see response to comment 20.

Comment 22: added "gradual" line 99 and "By contrast, " line 101. Deleted "In the Indio Hills, however, the nature of the transition between the lower and upper member of the Palm Spring Formation and the presence of an angular unconformity is unknown." lines 103–105. Moved "respectively" earlier in the sentence for clarity line 107. Split the last sentence into a new paragraph lines 117–120 because dealing with a different stratigraphic unit (Ocotillo Formation). Comment 23: see response to comment 22.

Comment 24: separated the narrative about age relationships into a discrete paragraph (lines 106–116).

Comment 25: deleted "Inversion of the Mecca basin started and lasted beyond the early/mid Pleistocene (< 0.76 Ma)." lines 111–112. Also see response to comment 6.

Comment 26: replaced "Mecca Hills and Durmid Hills" by "the Coachella Valley" lines 112–113. Comment 27: rewrote the section title into "Major tectonic culminations in the Coachella Valley" line 124.

Comment 28: added reference to Matti et al. (1985, 1992) and Matti and Morton (1993) lines 129–130 and Matti et al. (1992) and Matti and Morton (1993) to the reference list.

Comment 29: deleted "proto-SAFZ" lines 130-131 and replaced by "sedimentary".

Comment 30: rewrote sentence lines 130–136 into "Northwest and west of the Coachella Valley, the Miocene–Pliocene sedimentary strata are structurally bounded by the San Bernardino and San Jacinto fault strands of the SAFZ (Bilham and Williams, 1985; Matti et al., 1985; Morton and Matti, 1993; Spotila et al., 2007). To the southwest, Miocene–Pliocene strata are bounded by the West Salton detachment fault (Dorsey et al., 2011).". Also added Morton and Matti (1993) to the reference list.

Comment 31: see response to comment 30.

Comment 32: see response to comment 30.

Comment 33: see response to comment 30.

Comment 34: see response to comment 28.

Comment 35: replaced "fault gouge" by "trace" line 141.

Comment 36: deleted reference to Parrish (1983) throughout the manuscript (lines 35, 94, 144 and

154) and from the reference list. Also added reference to Dibblee and Mich (2008) line 94.

Comment 37: see response to comment 36.

Comment 38: replaced "Farther southeast" by "Southeast of the Indio Hills" line 150.

Comment 39: changed phrase lines 150–151 to "Gold et al. (2015) explore tectonogeomorphic evidence for dextral-oblique uplift and Keller et al. (1982) focus on landscape evolution". Also added Gold et al. (2015) to the reference list.

Comment 40: moved reference to Blisniuk et al. (2021) together with Keller et al. (1982) they both studied landscape evolution in the Indio Hills.

Comment 41: replaced "Besides studying" by "In addition to investigating" line 153 and "their study" by "Keller et al. (1982)" lines 154–155.

Comment 42: added "(shallow)" and "(Fuis et al., 2012, 2017)" line 178.

Comment 43: rewrote lines 189–192 into "The Durmid Hills are an elongate ridge that parallels the main strand of the SAFZ at the south edge of the Salton Sea in Imperial Valley (Fig. 1). To the south, this deformation zone and the SAFZ project towards the Brawley seismic zone, an oblique, transtensional rift area with particularly high seismicity".

Comment 44: deleted reference to Janecke et al. (2018) line 194 and added it lines 196 and 198.

Comment 45: added "both in the field and via imagery analysis" lines 215–216.

Comment 46: changed "when approaching" into "as they approach" line 234.

Comment 47: none.

Comment 48: replaced "they" by "the folds of the central macro-fold" line 254.

Comment 49: none.

Comment 50: deleted "with some confidence" line 300.

Comment 51: deleted parenthesis.

Comment 52: changed "The fold geometry is" by "Fold geometries are" line 329. Replaced "Its" by "The" line 330 and added "of these folds" line 331.

Comment 53: moved the following sentence "Brittle faults exist both in granitic basement and in sedimentary rocks of the Mecca and Palm Spring formations." to the beginning of the paragraph lines 336–337.

Comment 54: extended the barbs to the northwest along the Indio Hills fault in Figure 2.

Comment 55: deleted "Rymer, unpublished data" line 340.

Comment 56: added "The fault sets in granitic basement rocks trend parallel to fault sets in sedimentary strata southeast of the Indio Hills fault (see stereoplots in Figure 2) and are therefore suggested to have formed due to similarly oriented stress." lines 427-430.

Comment 57: added "Like the Indio Hills Fault, fault-plane dip and strike of the main San Andreas fault must be inferred indirectly." lines 361–362.

Comment 58: none.

Comment 59: none because not allowed by the journal's standards.

Comment 60: none.

Comment 61: changed "mud-silt-stone" into "mudstone-siltstone" line 356.

Comment 62: added "and thrust" to the title of the sub-section line 400 to better reflect the content.

Comment 63: see response to comment 56.

Comment 64: changed sentence to "In map view (Fig. 2), the folds are right-stepping, and each fold set is increasingly asymmetric (Z-shaped) and sigmoidal towards the Indio Hills fault in the northeast" lines 414–416.

Comment 65: deleted ", right-lateral" line 421.

Comment 66: deleted "due to distributed simple shear" lines 443-444.

Comment 67: added "(main San Andreas and Indio Hills faults)" line 430.

Comment 68: added "folds propagating outward from the Indio Hills fault is supported by the increased structural complexity of the fold geometries towards the Indio Hills fault." lines 457–459, and deleted ", thus favoring folds propagating outward from the Indio Hills fault" lines 460–461.

Comment 69: added ", and/or to structural tilting in the hanging wall of the Indio Hills fault" lines 469–470.

Comment 70: replaced "convergent tectonic" by "transpressional uplift" line 564, and added "(i.e., a contractional uplift formed synchronously with successively with simple shear transpression to balance internal forces in a crustal-scale critical taper; Dahlen, 1990)" lines 564–566 and Dahlen (1990) to the reference list.

Comment 71: added a paragraph break line 449.

Comment 72: added reference to Figure 7 line 478.

Comment 73: labelled the parasitic folds in Figure 3C.

Comment 74: replaced reference to Schlische (1995) by Suppe and Medwedeff (1990) line and in the reference list.

Comment 75: split the sentence into two for clarity. Replaced ", which" by ". The fault" line 522.

Comment 76: added "In this section we use the geometry and kinematics of folds and faults in the southern Indio Hills to reconstruct the tectonic history of the area, not only of the inverted late Cenozoic basin but also about strike-slip and dip-slip faults that bound the basin. Essential tectonic events include (1) extensional normal faulting along the Indio Hills fault, (2) reactivation of the Indio Hills fault as a right-lateral to right-lateral-reverse fault, and (3) right-lateral movement along the main San Andreas fault." Lines 539–544

Comment 77: see response to comment 80.

Comment 78: modified referce to Supplement S3b to Supplement S3d line 541.

Comment 79: see response to comments 80 and 156.

Comment 80: added ", by the deposition and preservation of sedimentary strata of the Palm Spring and Mecca formations southwest of the Indio Hills, whereas they were eroded or never deposited northeast of the fault, and by fining up upwards of the stratigraphic units from conglomerates in the Mecca Formation to coarse-grained sandstone in the lower parts of the Palm Spring Formation" lines 548–550. Added "In addition, the flat geometry of micro thrust faults (e.g., Supplements S3b– c) suggests that they were intensely rotated during macro-folding. Restoration of all micro faults in their initial position prior to macro-folding shows that some of these faults exhibit normal kinematics with associated syn-tectonic growth strata (Supplements S3d and S6)." lines 550-554. Added "basin geometry and formation similar to that of the Mecca Hills, where down-SW slip along the Painted Canyon fault was inferred in the (Miocene?-) Pliocene (McNabb et al., 2017), and of the transtensional Ridge Basin though with opposite vergence (Crowell, 1982; Ehman et al., 2000) with a" lines 556–558 and Crowell (1982) and Ehman et al. (2000) to the reference list. Added "during basin inversion" line 559. Also added "Formation of the Indio Hills fault probably occurred in mid-Miocene times during extension related to the opening of the Gulf of California (Stock and Hodges, 1989; Stock and Lee, 1994) as proposed for the Salton Trough (Dorsey et al., 2011 and references therein)." And Stock and Hodges (1989) and Stock and Lee (1994) to the reference list.

Comment 81: see response to comment 80.

Comment 82: deleted ", which gradually changed to a dominantly right-lateral-reverse fault" lines 562–563.

Comment 83: replaced "margin" by "convergent plate boundary" line 569.

Comment 84: added "during this period" line 572.

Comment 85: replaced "slip event" by "episode of movement along" line 573 and replaced "is clearly younger than the episode of" by "clearly postdates" line 574.

Comment 86: none commanded by the referee's comment.

Comment 87: none.

Comment 88: added "(i.e., after the initial transpressional slip events along the Indio Hills fault)." lines 580–581.

Comment 89: none.

Comment 90: none.

Comment 91: split the sentence lines 584–587 into two and changed the second sentence to ". However, when restoring the sedimentary strata to horizontal (Supplement S6), the fault-related folds define a low-angle fold-and-thrust system".

Comment 92: split the sentence lines 587–593 into two and added ". This implies" line 590.

Comment 93: added "(other than right-slip faults)" line 588.

Comment 94: added commas lines 602 and 603, added "upright" line 603, and changed "suggests" into "suggest" line 604.

Comment 95: replaced "layer-parallel" by "low-angle". Added "These disharmonic folds are interpreted as intra-detachment folds." lines 307–308. Added "intra-detachment" in figure 4 caption line 1126.

Comment 96: none.

Comment 97: added "In this section we use detailed structural analysis of folds and faults in the southeastern Indio Hills to outline the structural history of the tectonic culmination itself, evaluate it in terms of what is known about strain budgets within the southern San Andreas fault system, link it to nearby structures (Eastern California Shear Zone and Landers Mojave Line), and integrate the local structural history into a structural synthesis for the southern San Andreas Fault zone in the past 4 Myr." lines 621–626. Added "Here we compare and contrast the structural evolution of the southeastern Indio Hills with that of nearby culminations (Mecca Hills and Durmid Hills)." lines 718–719.

Comment 98: replaced "Miocene" by "Pleistocene" line 629. Deleted "early" line 117. Replaced "mid" by "late" line 107. Deleted "Pliocene–" line 737. Deleted "Pliocene and" line 821.

Comment 99: see response to comment 80.

Comment 100: replaced "bounded" by "downthrew Miocene sedimentary strata against" lines 630–631. Added "in the footwall of the fault. In the hanging wall of the fault they were" lines 632–633. Comment 101: none.

Comment 102: changed "master fault(s)" into "" lines 643–644.

Comment 103: changed the phrase lines 641–643 to "The fold set evolved oblique to the main San Andreas strand of the SAFZ and formed a right-stepping pattern of E–W-oriented axial surfaces that trend".

Comment 104: see response to comment 102.

Comment 105: added "in between two active strike slip faults" line 533 and "consistently" line 534. Replaced "near the main San Andreas fault" by "in the southwest" line 537. Added ", where the macro-folds still display their initial non-plunging geometries" lines 757–758.

Comment 106: deleted "precursory" line 646.

Comment 107: none.

Comment 108: none.

Comment 109: added "strain" line 651.

Comment 110: added "from mostly distributed with minor partitioned deformation" lines 653–654. Comment 111: added "right-lateral-" line 656.

Comment 112: replaced "seems to have still played a minor role" by "did not yet play a major role" line 657.

Comment 113: rewrote "attenuation of the macro-folds toward the Indio Hills fault, increased shear folding, and clockwise rotation of fold axes to a steeper westerly plunge" into "tightening of the macro-folds toward the Indio Hills fault and clockwise rotation of fold axes to a steeper westerly plunge due to increased shear folding" lines 658–660.

Comment 114: replaced "buckle-" by "en echelon upright" line 660.

Comment 115: added "These folds were not observed northeast of the Indio Hills fault, nor southwest of the main San Andreas fault." lines 250–251.

Comment 116: added "Furthermore, " line 666.

Comment 117: added "in the Pliocene" line 634, "in the Pleistocene" line 638, ", i.e., probably in mid Pleistocene time" line 642, "in the mid Pleistocene" line 654, "in the late Pleistocene (earliest late Pleistocene 0.765 Ma Bishop Ash involved in folding; Sarna-Wojcicki et al., 2000; Zeeden et al., 2014)" lines 670–672, and "mid (–late?)" line 849.

Comment 118: added "kinematically" line 780.

Comment 119: added "see anticline closest to Indio Hills fault in" lines 678-679.

Comment 120: deleted "margin-parallel" and added "parallel to the convergent plate boundary" lines 679–680.

Comment 121: see response to comment 105.

Comment 122: replaced "latter" by "overlapping and synchronous formation of structures" line 623.

Comment 123: see response to comment 105.

Comment 124: see response to comment 5.

Comment 125: added "that a through-going NNW–SSE-striking fault crosscuts the Pinto Mountain fault (e.g., 1992 Joshua Tree earthquake near the study area) and" line 708–709, "its segment in the study area," lines 709, ", NW–SE-striking" line 712, and "in the south" lines 7013–714.

Comment 126: added ". Notably, the 1992 Joshua Tree earthquake occurred along the NNW–SSEstriking, west-dipping West Deception Canyon fault (Rymer, 2000 and references therein), which merges with the Indio Hills fault in the south (see figure 1 in Rymer, 2000). Therefore, we propose" lines 708–711, and Rymer (2000) to the reference list.

Comment 127: see response to comment 12.

Comment 128: added ", i.e., a junction of three dextral fault branches" and "sensu" line 716, replaced "that" by ", which" line 717, reference to Passchier and Platt (2017) lines 717 and 721, and Passchier and Platt (2017) to the reference list.

Comment 129: deleted comma line 717.

Comment 130: added "late Pleistocene" lines 718-719.

Comment 131: added ", i.e., late Pleistocene" lines 724–725. Also see response to comment 117. Comment 132: added "Our observations of mostly lateral movement along the main San Andreas fault (i.e., southeastern continuation of the Mission Creek fault) and the proposed Pleistocene age for deformation in the southeastern Indio Hills are consistent with work by Keller et al. (1982). A major difference between the northwestern and southeastern Indio Hills is the relatively higher amount and more intense character of deformation in between the two bounding faults in the latter with tighter macro-folding over a narrower area (Figs. 2 and 3; Keller et al., 1982;

Lancaster et al., 2012)." lines 694–700.

Comment 133: rewrote the sentence into "in the southeasternmost Indio Hills and proceed as the main San Andreas fault" lines 746–747.

Comment 134: replaced "them" by "these faults" line 753.

Comment 135: none.

Comment 136: added "Another possible explanation may be the presence of coarse-grained deposits of the Mecca Formation, which may have partitioned/decoupled deformation along the Indio Hills fault from that in overlying Palm Spring sedimentary strata." lines 758–761.

Comment 137: added reference to McNabb et al. (2017) line 184. Also see response to comment 80.

Comment 138: see response to comment 137.

Comment 139: see response to comment 137.

Comment 140: deleted "The increasing width of damage zones adjacent to SAFZ-related faults southward in Coachella Valley, and increased number of strike-slip and oblique to orthogonal cross faults in the Durmid Hills compared with Indio Hills and Mecca Hills may be due to closeness and transition to a transtensional rift setting around the Brawley seismic zone (Janecke et al., 2018)." lines 934–938. Replaced "one–two" by "several" line 545.

Comment 141: see response to comment 12.

Comment 142: see response to comment 117.

Comment 143: added "The Indio Hills fault probably initiated as a SW-dipping normal fault during the opening of the Gulf of California in the Miocene, and was later inverted as an right-lateral reverse, oblique-slip fault in the mid (–late?) Pleistocene due to transpression along the convergent plate boundary." lines 843–846.

Comment 144: see response to comments 125 and 126.

Comment 145: added Lancaster et al. (2012) to the reference list.

Comment 146: see response to comment 145.

Comment 147: see response to comment 128.

Comment 148: see response to comment 12.

Comment 149: none.

Comment 150: see response to comment 73.

Comment 151: see response to comment 119.

Comment 152: added ages of the three tectonic phases to figures 7 and 8.

Comment 153: decreased length of Garnet Hill fault in figure 8.

Comment 154: none.

Comment 155: correct "Hills" into "Hill" line 1230.

Comment 156: see response to comment 80.

**Revisions by the authors of the present manuscript based on editing comments in supplementary pdf by the referee**

-Replaced "late" by "mid/upper" line 74 and added reference to Kirby et al. (2007) lines 76–77. -Deleted "the" line 81. -Replace "disconnected" by "segmented" line 81.

-Replaced "turns into" by "is succeeded by" line 87.

-Added "southwestwards" line 94.

-Deleted "was" and "and" and moved reference to Bergh et al. (2019) line 95.

-Deleted "Absolute dating revealed an" line 102.

-Deleted "and" and replaced by "are consistent with" line 105.

-Changed "1.0" to "0.76" lines 106 and 642 for the minimum age of the Palm Spring Formation for consistency with dating of the Bishop Ash by Sarna-Wojcicki et al. (2000).

-Added "-" line 106.

-Added "We infer a similar age range for the Palm Spring Formation in the southern Indio Hills." Lines 106–107.

-Replaced "emerges from the involvement" by "include tephrochronology", and added ", which is involved in deformation" line 111.

-Moved "either" early in sentence and added "structures like" lines 142–143.

-Deleted "modern data remain scarce, and" line 146.

-Replaced "have not been published from this segment of the SAFZ" by "documenting this hypothesis for the culmination as a whole have not been conducted" lines 147–148.

-Added ", i.e., northwest of the study area" lines 152–153.

-Moved reference to Bergh et al. (2014, 2019) from line 168 to line 163.

-Replaced "exist" by "occur".

-Deleted "whereas" and moved "northeast of the SAF" earlier in the sentence, and split sentence into two lines 189–190.

-Replaced "the present study" by "our investigation of the Indio Hills" line 202.

-Deleted "in the Indio Hills" line 204.

-Replaced "when" by "for" line 207.

-Replaced "with" by "and nearby" lines 208–209.

-Replaced "with" by "having" line 216.

-Changed "an approximately two kilometers wide zone" into "a zone approximately two kilometers wide" lines 217–218.

-Changed "with" into "showing" line 223.

-Deleted "in map view" line 229.

-Replaced ", whereas" by ". In contrast," line 235.

-Deleted "in the southwest" line 236.

-Replaced "when approaching" by "as they approach" lines 245–246.

-Replaced "turn" by "transform" line 248.

-Changed the phrase lines 251–253 into "From southwest to neatheast, the central macro-fold hinge zone displays a corresponding change in geometry".

- -Replaced "with" by "having" line 261.
- -Changed "units" into" parts" line 265.

-Changed "the Mecca Formation conglomerates" into "conglomerates of the underlying Mecca Formation" lines 265–266.

-Added "(especially)" and moved "in relatively weak clayish–silty dark mudstone layers" earlier in sentence lines 267–268.

- -Replaced "On the contrary" by "By contrast" line 270.
- -Replaced "with" by "showing" line 273.

-Changed the phrase lines 276–278 to "Combined with a relatively narrow hinge zone, these attitudes define".

-Changed phrase line 285 into "conglomerates of the underlying Mecca Formation".

-Added "surfaces" line 285 and changed "is2 into "are" line 286.

- -Replaced "with" by "having" line 290.
- -Replaced "units" by "strata" line 291.
- -Changed phrase lines 297–298 into "the conglomerates of the Mecca Formation".
- -Replaced "until" by "to where" line 300.
- -Replaced "almost" by "nearly" line 304.
- -Added "strata of the" line 304.

-Replaced "with" by "showing" and "is" by "are" line 313.

-Changed phrase line 327 into ", where preserved, they display".

-Changed sentence lines 335–338 into "Along the Indio Hills fault, poor exposures make it difficult to measure fault strike and dip directly, but DEM images suggest a rectilinear geometry in map view relative to the uplifted sedimentary strata to the southwest".

-Repalced "possibly" by "probably" line 339.

-Moved "by the Banning fault" from lie 350 to 349.
-Changed "when approaching" into "where they approach" lines 351–352.

-Replaced "impact" by "developed" line 355.

-Added "-" line 361.

-Replaced "with" by "displaying" line 368.

-Replaced "operate together defining" by "appear to form" line 376.

-Replaced "formed" by "developed" line 377.

-Added "-" lines 395 and 402.

-Changed "50 meters" into "50-m" line 403.

-Added "Structural" lines 412 and 471.

-Changed phrase lines 413–414 into "We mapped and analyzed three macro-scale fold systems that occur".

-Changed the phrase lines 416–417 into "Based on these properties, we interpret the fold sets".

-Changed the sentence lines 418–422 into "Various investigators (Babcock, 1974; Miller, 1998; Titus et al., 2007; Janecke et al., 2018; Bergh et al., 2019) describe similar fold geometries in sedimentary strata from many other segments of the SAFZ and are interpreted as structures formed by right-lateral displacement between two major fault strands due to distributed simple shear.".

-Changed the phrase lines 422–423 into "However, the present fold-orientation data in the Indio Hills (Fig. 2)".

-Replaced "e.g.," by "compare with" line 432.

-Added "-" line 447.

-Added "other" line 452.

-Replaced "hidden" by "blind" line 457.

-Deleted "(Fig. 3c)" line 478.

-Added "(NE-vergent)" line 482.

-Added "(not fully understood)" line 492.

-Replaced "with" by "having" line 521.

-Added "(and decreasing right-lateral)" line 522.

-Replaced "acted" by "ultimately functioned" line 525.

-Added "simultaneously" line 528.

-Replaced "However" by "In addition" line 531.

-Replaced "when approaching" by "towards" line 532.

-Deleted "also" line 533.

-Changed the phrase lines 536–537 into "spatial, temporal and kinematic" and deleted "in the Indio Hills" line 537

-Changed phrase lines 550-551 into "escaped from the mudstone beds and propagated".

-Replaced "events with" by ", incorporating" line 573.

-Added "of the Indio Hills" line 610.

-Changed phrase lines 611–613 into "was accommodated by right-lateral-oblique, top-NE thrusting along the Indio Hills fault and major strike-slip movement".

-Added "3c and" line 616.

-Changed the phrase lines 629–631 to "The right-lateral-reverse character of the Indio Hills Fault and its role in our kinematic model for basin inversion in the southern Indio Hills".

-Replaced "East" by "Eastern" line 632.

-Replaced "simultaneously" by "coevally" line 699.

-Added "right-lateral" line 700.

-Rewrote "main SAFZ" into "main San Andreas fault" line 751.

**Additional revisions by the authors of the present manuscript**

-Added "(we refrain from using the name "Indio strand" given to this fault by Gold et al., 2015 to avoid confusion with the Indio Hills fault)" lines 39–41.

-Moved "Atwater and Stock, 1998;" before "Spotila et al., 2007;" line 64.

-Changed "uppermost members" into "upper member" line 118.

-Replaced "marks the" by "is a" line 201.

-Deleted "-" line 480.

-Deleted en-dash line 506.

-Added "(probably soutwest-dipping)" line 753.

-Replaced "show" by "suggest" line 786.

-Added a comma line 831.

-Corrected "Janecke et al., 2019" into "Janecke et al., 2018" line 842.

-Deleted "and main SAFZ" line 863.

-Moved "basement-seated" from line 899 to line 891.

[revised manuscript text omitted]